# On the Importance of the Recovery Procedure in the Semi-Analytical Solution for the Static Analysis of Curved Laminated Panels: Comparison with 3D Finite Elements

**DOI:** 10.3390/ma17030588

**Published:** 2024-01-25

**Authors:** Francesco Tornabene, Matteo Viscoti, Rossana Dimitri

**Affiliations:** Department of Innovation Engineering, School of Engineering, University of Salento, 73100 Lecce, Italy; matteo.viscoti@unisalento.it (M.V.); rossana.dimitri@unisalento.it (R.D.)

**Keywords:** equivalent single layer, generalized differential quadrature, higher order theories, navier solution, static analysis, stress and strain recovery

## Abstract

The manuscript presents an efficient semi-analytical solution with three-dimensional capabilities for the evaluation of the static response of laminated curved structures subjected to general external loads. A two-dimensional model is presented based on the Equivalent Single Layer (ESL) approach, where the displacement field components are described with a generalized formulation based on a higher-order expansion along the thickness direction. The fundamental equations are derived from the Hamiltonian principle, and the solution is found by means of Navier’s approach. Then, an efficient recovery procedure, derived from the three-dimensional elasticity equations and based on the Generalized Differential Quadrature (GDQ) method, is adopted for the derivation of the three-dimensional solution. Some examples of investigation are presented, where the numerical predictions of refined three-dimensional Finite-Element-based models are matched with a high level of accuracy. The model is validated for both straight and curved panels, taking into account different lamination schemes and load shapes. Furthermore, it is shown that the numerical solution to the elasticity problem in the recovery procedure is determining and accurately predicting the three-dimensional static response of the doubly-curved shell solid.

## 1. Introduction

Recent advancements in engineering applications require innovative strategies to accurately the static and dynamic responses of structural components of very complex shapes [1,2,3,4,5]. For this reason, advanced models are necessary to describe the geometry and the mechanical characteristics of these structures with reduced computational costs [6,7]. Three-dimensional approaches based on the elasticity equations provide highly accurate predictions of the structural response of a doubly-curved solid, but they can be computationally expensive [8,9]. On the other hand, two-dimensional formulations that consider a higher-order description of the field variable can be a valid alternative to three-dimensional models [10,11,12,13]. In a two-dimensional theory, a doubly-curved surface is considered to have equivalent mechanical properties instead of a doubly-curved three-dimensional solid [14,15,16]. The unknown field variables are described using either the Equivalent Single Layer (ESL) or the Layer-Wise (LW) approaches [17,18,19,20,21]. More specifically, in ESL theories, a higher-order expansion is established along the thickness direction by means of the so-called thickness functions. However, for moderately thick and thick panels, a LW description of the field variable may be more accurate. According to the LW methodology, the fundamental relations are written in each layer of shell, and the interaction between adjacent laminae is taken into account through compatibility conditions. Referring to two-dimensional ESL theories, when the shell laminated structure is made of advanced materials, it is very likely that a higher-order expansion of the unknown field variable is required [22,23,24], leading to the well-known Higher-Order Shear Deformation Theories (HSDTs). The higher-order expansion in hand can be performed with both polynomial and non-polynomial thickness functions [25,26,27,28]. Besides, the coupling between two-adjacent layers, which is known as zigzag effects, can be easily modeled in ESL theories with the so-called zigzag thickness functions, as shown in Refs. [29,30,31]. In this way, a piecewise inclination of the displacement field profile is provided. The approach provided for the first time in Ref. [32] has been shown to be very simple and accurate, among others. On the other hand, in Refs. [33,34,35], the refined zigzag theories are presented, where the zigzag functions are derived from the stiffness properties of the lamination scheme of the selected panel, leading to the so-called refined zigzag theories.

Among two-dimensional approaches, an efficient strategy to define the fundamental relations can be found in the well-known generalized formulation [36,37,38,39], where the through-the-thickness expansion of the configuration variable up to an arbitrary order is modeled regardless of the effective expression of the selected thickness function. In this way, several structural models with different kinematic assumptions can be derived. Furthermore, classical formulations like the First Order Shear Deformation Theory (FSDT) and the Third Order Shear Deformation Theory (TSDT) can be seen as particular cases of the generalized higher-order theory [40,41]. The adoption of a generalized configuration variable can be an efficient strategy when structures of different materials and various lamination schemes are studied with an arbitrary variation of mechanical properties, as happens in the case of Functionally Graded Materials (FGMs) [42,43,44], Carbon Nanotube (CNT) composites [45,46,47], honeycomb and lattice cells [48,49], as well as three-dimensional Variable Angle Tow (3D-VAT) composites [50,51,52]. Furthermore, the adoption of HSDTs allows one to study the effect of porosity because the presence of voids with an arbitrary distribution within the structure leads to a reduction in material stiffness [53,54].

For structures with complex shapes, materials, and boundary conditions, it is very difficult to derive a closed-form solution to the governing equations; where numerical approximations must be used to discretize the problem [55,56,57,58]. On the other hand, it is possible to derive a closed-form solution if some simplifications are taken into account [59,60,61,62,63]. In addition, some applications of practical interest can be examined with an infinite series expansion of each unknown variable [64,65,66,67]. It should be noted that exact solutions for simply supported layered structures are typically adopted to check the accuracy of the numerical solution. Among semi-analytical solutions for linear elasticity, a milestone is the research work by Pagano [68,69], where a three-dimensional closed-form solution is derived for laminated composite plates and sandwich panels. Then, two-dimensional models have been developed for laminated plates and curved sandwich shells. Starting from formulations based on the Classical Plate Theory (CPT), refined theories can be found in literature based on FSDT and TSDT. Recent developments in the field of smart materials have led to the development of new formulations regarding plates and shells with smart properties like piezoelectricity, magnetostriction, and heat transfer [70], and closed-form solutions have been derived for the validation of numerical simulations. Some preliminary works regarding laminated plates with piezoelectric [71,72] and piezomagnetic [73,74] properties must be cited.

It is interesting to note that closed-form solutions may not be used for practical applications because of the assumptions that are usually considered. However, some results of more practical interest can be obtained with semi-analytical formulations. In a semi-analytical theory, the solution is obtained with an expansion of degrees of freedom (DOFs) up to a sufficient order. The Navier method and the Levy procedure have been extensively adopted in several papers regarding linear elasticity problems for plates and shells [75,76,77,78]. More specifically, the Navier solution, based on the description of the field variable with a trigonometric series, can be adopted in the case of simply-supported laminated panels with cross-ply lamination schemes and antisymmetric angle-ply composites. In contrast, the Levy method [79,80] is suitable for panels with two simply supported edges and the other two subjected to arbitrary boundary conditions. An arbitrary load case can be analyzed with the Navier method if the external actions are expanded with a double Fourier trigonometric series. It has been shown in Ref. [81] that this methodology can also be adopted for closed panels of revolution. On the other hand, the Navier approach can be difficult to apply to structures of very complex shapes made of non-conventional materials because a significantly high number of terms of the harmonic expansion should be considered to obtain a sufficient level of accuracy. For this reason, a numerical model is more frequently adopted to derive an approximate solution under a limited number of hypotheses. Among numerical techniques, the Finite Element Method (FEM), which is extensively adopted in several applications and commercial codes, is based on a local interpolation of the unknown field variable in a discrete computational grid by means of the so-called shape functions [82,83,84]. In contrast, the class of spectral collocation methods [85,86,87] are based on a global interpolation of the solution by means of higher-order functions; therefore, a smoother solution can be derived with a reduced number of DOFs. Belonging to this class, the Generalized Differential Quadrature (GDQ) method [88,89,90,91,92] approximates the derivatives of an arbitrary function as a weighted sum of the function itself. It has been shown in several papers that the highest level of accuracy, computational stability, and efficiency is reached when the computational domain is discretized with non-uniform grids [93,94]. Moving from the GDQ numerical technique, the Generalized Integral Quadrature (GIQ) allows one to compute the integrals with a significantly reduced number of DOFs [95,96].

When a two-dimensional solution is derived, the reconstruction of the effective structural response of the panel can be difficult since it is not guaranteed, a priori, that the distribution of stress components satisfies the three-dimensional elasticity equations. This issue may lead to erroneous results, especially for moderately thick structures, where the stress components acting in the thickness direction cannot be neglected. For this reason, a correction of the stress and strain profiles should be conducted based on the three-dimensional elasticity equations. In Refs. [97,98,99], an effective stress and strain recovery procedure is provided for the evaluation of the static response of moderately thick doubly-curved shell structures made of laminated composite materials with arbitrary orientation and FGM, starting from a refined two-dimensional GDQ-based numerical solution. The recovery procedure has been demonstrated to be an effective tool for the reconstruction of the three-dimensional response of doubly-curved shell structures with advanced lamination schemes, starting from numerical solutions of refined formulations based on HSDTs. In some previous works [100,101], the recovery procedure has been applied to some two-dimensional GDQ-based numerical solutions for the prediction of the static response of doubly-curved shells with generally anisotropic materials. However, the effects of this procedure on two-dimensional semi-analytical solutions have not been checked.

In the present work, a two-dimensional model based on the ESL approach with HSDTs is presented to predict the linear static response of laminated curved panels. The geometry of the structure is described with the differential geometry basics and curvilinear principal coordinates. The generalized formulation is adopted for the description of the kinematic field, and a higher-order expression is provided along the thickness direction of the panel for each displacement field component. Furthermore, the zigzag effects are considered in the kinematic model. Following the ESL approach, the mechanical properties of each layer, modeled with a generally anisotropic constitutive relationship, are homogenized on the reference surface. The fundamental equations are derived from the Hamiltonian principle, accounting for an arbitrary distribution of the external surface loads, which are applied at the top and bottom of the laminated panel. Then, the semi-analytical Navier solution is found under some geometric and mechanical assumptions, taking into account a Fourier series expansion of the unknown field variable. Finally, a recovery procedure based on the three-dimensional elasticity equations for a doubly-curved solid is applied for the reconstruction of the three-dimensional response of the panel. Some examples of investigation are presented, where the accuracy of the semi-analytical formulation is checked for different curvatures, lamination schemes, and load cases. The numerical predictions have been performed with various kinematic field assumptions, and the results are compared to those coming from three-dimensional finite element models. Furthermore, some comparisons are conducted with the results of refined GDQ numerical models. It is shown that the recovery procedure determines when a semi-analytical procedure is adopted. In this way, it is possible to accurately predict the three-dimensional response of curved laminated structures with a reduced computational cost even though a two-dimensional formulation is adopted. On the other hand, if the stress and strain profiles are derived directly from the two-dimensional solution without any post-processing technique, some results may be obtained that are not consistent with the equilibrium of the structure under external loads.

## 2. Geometry of a Shell Structure

A doubly-curved shell is a three-dimensional solid within the Euclidean space whose position vector, denoted by $R\left(\alpha_{1},\alpha_{2},\alpha_{3}\right)$, is dependent on three parameters $\alpha_{\;i}=\alpha_{\;1},\alpha_{\;2},\alpha_{\;3}$. These parameters are defined in the closed interval $\alpha_{\;i}\in \left[\alpha_{i}^{0},\alpha_{i}^{1}\right]$ with i=1,2,3, where $\alpha_{\;i}^{0}<\alpha_{\;i}^{1}$ are the extremes of the domain at issue. Following the ESL approach, the position vector R of a doubly-curved shell element is defined in terms of a reference surface, denoted by $r\left(\alpha_{1},\alpha_{2}\right)$, located in the middle thickness $h\left(\alpha_{1},\alpha_{2}\right)$ of the panel:(1)$$R\left(\alpha_{1},\alpha_{2},\zeta \right)=r\left(\alpha_{1},\alpha_{2}\right)+\frac{h\left(\alpha_{1},\alpha_{2}\right)}{2}zn\left(\alpha_{1},\alpha_{2}\right)$$

In the previous relation, z=2ζ/h is a dimensionless coordinate for the thickness direction, oriented alongside the outward normal unit vector $n\left(\alpha_{1},\alpha_{2}\right)$. This vector can be calculated at each point of the reference surface in terms of the partial derivatives of $r\left(\alpha_{1},\alpha_{2}\right)$ with respect to the principal coordinates $\alpha_{\;1},\alpha_{\;2}$, denoted by the symbol $r_{,i}=\partial r/\partial \alpha_{i}$ with i=1,2. One gets [16]:(2)$$n\left(\alpha_{1},\alpha_{2}\right)=\frac{r_{,1}\times r_{,2}}{\left|r_{,1}\times r_{,2}\right|}$$

Following Equation (1), the geometric properties of the three-dimensional shell element can thus be defined from those of the reference surface $r\left(\alpha_{1},\alpha_{2}\right)$. The principal radii of curvature $R_{\;i}\left(\alpha_{1},\alpha_{2}\right)=R_{\;1},R_{\;2}$, referred to the $\alpha_{\;i}=\alpha_{\;1},\alpha_{\;2}$ principal direction, can thus be derived as follows:(3)$$R_{\;i}\left(\alpha_{1},\alpha_{2}\right)=-\frac{r_{,i}\cdot r_{,i}}{r_{,ii}\cdot n}\;\;\;\;\;i=1,2$$

Note that the symbol $r_{,ij}=\partial^{2}r/\left(\partial \alpha_{i}\partial \alpha_{j}\right)$ with i,j=1,2 refers to the second order derivatives of the reference surface r with respect to $\alpha_{\;i},\alpha_{\;j}=\alpha_{\;1},\alpha_{\;2}$. For the sake of completeness, the principal curvature $k_{\;ni}=1/R_{i}$ with i=1,2 can be introduced along each principal direction. The Lamè parameters $A_{\;i}\left(\alpha_{1},\alpha_{2}\right)=A_{\;1},A_{\;2}$ can be computed at each point of the physical domain according to the following relation:(4)$$A_{\;i}\left(\alpha_{1},\alpha_{2}\right)=\sqrt{r_{,i}\cdot r_{,i}}\;\;\;\;\;i=1,2$$

The Lamè parameters $A_{\;i}=A_{\;1},A_{\;2}$ are used for the computation of the infinitesimal arch lengths $ds_{i}=ds_{1},ds_{2}$ along the principal directions $\alpha_{\;i}=\alpha_{\;1},\alpha_{\;2}$ of the reference surface. The arch length $s_{\;i}$ with i=1,2 is defined so that $s_{\;i}\in \left[0,L_{i}\right]$, being $L_{\;i}$ the length of the parametric line. The following relation can thus be written:(5)$$ds_{i}=A_{\;i}d\alpha_{i}\;\;\;\;\;i=1,2$$

On the other hand, the scaling parameters $H_{\;i}=H_{\;1},H_{\;2}$ are defined in order to evaluate the scaling effects along the thickness direction that can be seen in a solid with one or more curvatures:(6)$$H_{i}\left(\alpha_{1},\alpha_{2},\zeta \right)=1+\frac{\zeta }{R_{i}}\;\;\;\;\;i=1,2$$

The three-dimensional shell solid is obtained from the superimposition of l different layers of thickness $h_{\;k}$, therefore the total thickness of the structure is thus obtained as the sum of the width of each k-th lamina, with k=1,…,l, which is located between its intrados and its extrados, denoted by $\zeta_{\;k}$ and $\zeta_{\;k+1}$, respectively [16]:(7)$$h\left(\alpha_{1},\alpha_{2}\right)=\sum_{k=1}^{l}h_{k}\left(\alpha_{1},\alpha_{2}\right)=\sum_{k=1}^{l}\left(\zeta_{k+1}-\zeta_{k}\right)$$

## 3. Kinematic Relations

In the present section, a generalized ESL approach is presented for the expansion up to the $\left(N+1\right)$-th order of the three-dimensional displacement field vector $U^{\left(k\right)}\left(\alpha_{1},\alpha_{2},\zeta \right)={\left[U_{1}^{\left(k\right)}\;\;U_{2}^{\left(k\right)}\;\;U_{3}^{\left(k\right)}\right]}^{\;T}$ along the thickness direction. To this end, three thickness functions $F^{\left(k\tau \right)\alpha_{i}}\left(\zeta \right)=F^{\left(k\tau \right)\alpha_{1}},F^{\left(k\tau \right)\alpha_{2}},F^{\left(k\tau \right)\alpha_{3}}$ are introduced for each τ-th kinematic expansion order with τ=0,…,N+1. Remembering that the thickness coordinate is defined so that $\zeta \in \left[\zeta_{k},\zeta_{k+1}\right]$ for k=1,…,l, a generalized formulation can be derived, and the vector $u^{\left(\tau \right)}\left(\alpha_{1},\alpha_{2}\right)$ of the generalized displacement field components $u_{\;1}^{\left(\tau \right)},u_{\;2}^{\left(\tau \right)},u_{\;3}^{\left(\tau \right)}$ is introduced for each τ=0,…,N+1. One gets the following relation [16]:(8)$$U^{\left(k\right)}\left(\alpha_{1},\alpha_{2},\zeta \right)=\sum_{\tau =0}^{N+1}F^{\left(k\tau \right)}\left(\zeta \right)u^{\left(\tau \right)}\left(\alpha_{1},\alpha_{2}\right)\;\Leftrightarrow \;\left[\begin{array}{c} U_{1}^{\left(k\right)}\\ U_{2}^{\left(k\right)}\\ U_{3}^{\left(k\right)} \end{array}\right]=\sum_{\tau =0}^{N+1}\left[\begin{array}{ccc} F^{\left(k\tau \right)\alpha_{\;1}} & 0 & 0\\ 0 & F^{\left(k\tau \right)\alpha_{\;2}} & 0\\ 0 & 0 & F^{\left(k\tau \right)\alpha_{\;3}} \end{array}\right]\left[\begin{array}{c} u_{\;1}^{\left(\tau \right)}\\ u_{\;2}^{\left(\tau \right)}\\ u_{\;3}^{\left(\tau \right)} \end{array}\right]$$

The ESL formulation of the previous relation allows one to derive a generalized two-dimensional theory for the mechanical elasticity problem. The selection of a particular expression of the thickness functions allows one to obtain several models for the static analysis of shell structures, including classical approaches like the CPT, FSDT, and TSDT. On the other hand, the thickness function related to τ=N+1 simulates the zig-zag effect that occurs in the interlaminar region, which consists of an abrupt variation of the profile of each displacement field component. In the present work, power thickness functions are used for each τ=0,…,N, together with Murakami’s zigzag function associated to τ=N+1:(9)$$F^{\left(k\tau \right)\alpha_{i}}\left(\zeta \right)=\left\{\begin{array}{ll} \zeta^{\;\tau } & \tau =0,\ldots ,N\\ {\left(-1\right)}^{k}z_{k}={\left(-1\right)}^{k}\left(\frac{2}{\zeta_{k+1}-\zeta_{k}}\zeta -\frac{\zeta_{k+1}+\zeta_{k}}{\zeta_{k+1}-\zeta_{k}}\right) & \tau =N+1 \end{array}\right.$$

When the thickness functions of Equation (9) are used, the notation EDZ-N can be used for the identification of the higher-order theory [16]. More specifically, “E” means that the two-dimensional theory is based on the ESL approach, whereas “D” means that an axiomatic expansion is adopted of the displacement field components, which are the configuration variables of the problem. When the zig-zag function is considered for τ=N+1, the letter “Z” is used. Finally, N denotes the maximum expansion order of the configuration variable.

At this point, the kinematic relations are derived for the ESL theory from those of the three-dimensional elasticity problem, as shown in Ref. [16]. If $\varepsilon^{\left(k\right)}\left(\alpha_{1},\alpha_{2},\zeta \right)={\left[\varepsilon_{1}^{\left(k\right)}\;\;\;\varepsilon_{2}^{\left(k\right)}\;\;\;\gamma_{12}^{\left(k\right)}\;\;\;\gamma_{13}^{\left(k\right)}\;\;\;\gamma_{23}^{\left(k\right)}\;\;\;\varepsilon_{3}^{\left(k\right)}\right]}^{T}$ is the three-dimensional strain vector of the k-th layer, the following condensed relation can be taken into account:(10)$$\varepsilon^{\left(k\right)}\left(\alpha_{1},\alpha_{2},\zeta \right)=DU^{\left(k\right)}=D_{\zeta }\left(\sum_{i=1}^{3}D_{\Omega }^{\alpha_{\;i}}\right)U^{\left(k\right)}$$

In the previous relation, D is the kinematic differential operator, which is split in those denoted by $D_{\zeta }$ and $D_{\;\Omega }^{\alpha_{i}}=D_{\;\Omega }^{\alpha_{1}},D_{\;\Omega }^{\alpha_{2}},D_{\;\Omega }^{\alpha_{3}}$, which contain the derivatives with respect to the coordinate ζ and $\alpha_{\;1},\alpha_{\;2}$, respectively. In an extended form, $D_{\zeta }$ can be expressed as:(11)$$D_{\zeta }=\left[\begin{array}{ccccccccc} \frac{1}{H_{1}} & 0 & 0 & 0 & 0 & 0 & 0 & 0 & 0\\ 0 & \frac{1}{H_{2}} & 0 & 0 & 0 & 0 & 0 & 0 & 0\\ 0 & 0 & \frac{1}{H_{1}} & \frac{1}{H_{2}} & 0 & 0 & 0 & 0 & 0\\ 0 & 0 & 0 & 0 & \frac{1}{H_{1}} & 0 & \frac{\partial }{\partial \zeta } & 0 & 0\\ 0 & 0 & 0 & 0 & 0 & \frac{1}{H_{2}} & 0 & \frac{\partial }{\partial \zeta } & 0\\ 0 & 0 & 0 & 0 & 0 & 0 & 0 & 0 & \frac{\partial }{\partial \zeta } \end{array}\right]$$

On the other hand, the kinematic operators $D_{\;\Omega }^{\alpha_{1}},D_{\;\Omega }^{\alpha_{2}},D_{\;\Omega }^{\alpha_{3}}$ of size 9×3 take the following aspect:(12)$$\begin{array}{l} D_{\Omega }^{\alpha_{1}}=\left[{\bar{D}}_{\Omega }^{\alpha_{\;1}}\;\;0\;\;\;\;0\right]\\ D_{\Omega }^{\alpha_{2}}=\left[0\;\;\;\;{\bar{D}}_{\Omega }^{\alpha_{\;2}}\;\;0\right]\\ D_{\Omega }^{\alpha_{3}}=\left[0\;\;\;\;0\;\;\;\;{\bar{D}}_{\Omega }^{\alpha_{\;3}}\right] \end{array}$$

The sub-operators ${\bar{D}}_{\;\Omega }^{\alpha_{1}},{\bar{D}}_{\;\Omega }^{\alpha_{2}},{\bar{D}}_{\;\Omega }^{\alpha_{3}}$ are written in an extended form as:(13)$${\bar{D}}_{\;\Omega }^{\alpha_{1}}=\left[\begin{array}{c} \frac{1}{A_{1}}\frac{\partial }{\partial \alpha_{\;1}}\\ \frac{1}{A_{1}A_{2}}\frac{\partial A_{2}}{\partial \alpha_{\;1}}\\ -\frac{1}{A_{1}A_{2}}\frac{\partial A_{1}}{\partial \alpha_{\;2}}\\ \frac{1}{A_{2}}\frac{\partial }{\partial \alpha_{\;2}}\\ -\frac{1}{R_{1}}\\ 0\\ 1\\ 0\\ 0 \end{array}\right],\;\;\;{\bar{D}}_{\;\Omega }^{\alpha_{2}}=\left[\begin{array}{c} -\frac{1}{A_{1}A_{2}}\frac{\partial A_{1}}{\partial \alpha_{\;2}}\\ \frac{1}{A_{2}}\frac{\partial }{\partial \alpha_{\;2}}\\ \frac{1}{A_{1}}\frac{\partial }{\partial \alpha_{\;1}}\\ -\frac{1}{A_{1}A_{2}}\frac{\partial A_{2}}{\partial \alpha_{\;1}}\\ 0\\ -\frac{1}{R_{2}}\\ 0\\ 1\\ 0 \end{array}\right],\;\;\;{\bar{D}}_{\;\Omega }^{\alpha_{3}}=\left[\begin{array}{c} \frac{1}{R_{1}}\\ \frac{1}{R_{2}}\\ 0\\ 0\\ \frac{1}{A_{1}}\frac{\partial }{\partial \alpha_{\;1}}\\ \frac{1}{A_{2}}\frac{\partial }{\partial \alpha_{\;2}}\\ 0\\ 0\\ 1 \end{array}\right]$$

Introducing Equation (8) in Equation (10), the generalized ESL kinematic relations are obtained, accounting for the effects of the curvature of the shell and the higher-order kinematic expansion of the displacement field variable [16]:(14)$$\varepsilon^{\left(k\right)}=\sum_{\tau =\;0}^{N+1}\sum_{i=1}^{3}Z{\;}^{\left(k\tau \right)\alpha_{\;i}}\varepsilon^{\left(\tau \right)\alpha_{\;i}}$$

As can be seen, the three-dimensional strain vector $\varepsilon^{\left(k\right)}\left(\alpha_{1},\alpha_{2},\zeta \right)$ has been expressed in terms of the generalized strain vector, for each τ=0,…,N+1, denoted by $\varepsilon^{\left(\tau \right)\alpha_{i}}\left(\alpha_{1},\alpha_{2}\right)={\left[\varepsilon_{1}^{\left(\tau \right)\alpha_{\;i}}\;\;\;\varepsilon_{2}^{\left(\tau \right)\alpha_{\;i}}\;\;\;\gamma_{1}^{\left(\tau \right)\alpha_{\;i}}\;\;\;\gamma_{2}^{\left(\tau \right)\alpha_{\;i}}\;\;\;\gamma_{13}^{\left(\tau \right)\alpha_{\;i}}\;\;\;\gamma_{23}^{\left(\tau \right)\alpha_{\;i}}\;\;\;\omega_{13}^{\left(\tau \right)\alpha_{\;i}}\;\;\;\omega_{23}^{\left(\tau \right)\alpha_{\;i}}\;\;\;\varepsilon_{3}^{\left(\tau \right)\alpha_{\;i}}\right]}^{T}$, whose definition is reported in the following:(15)$$\varepsilon^{\left(\tau \right)\;\alpha_{i}}=D_{\;\Omega }^{\;\alpha_{i}}u^{\left(\tau \right)}\;\mathrm{for}\;\begin{array}{l} \tau =0,\ldots ,N+1,\\ i=1,2,3 \end{array}$$

On the other hand, the ESL kinematic operator $Z{\;}^{\left(k\tau \right)\alpha_{i}}$ is defined for each τ=0,…,N+1 in terms of the generalized thickness functions of Equation (8):(16)$$Z^{\left(k\tau \right)\alpha_{i}}=D_{\zeta }F^{\left(k\tau \right)\alpha_{i}}=\left[\begin{array}{ccccccccc} \frac{F^{\left(k\tau \right)\alpha_{\;i}}}{H_{1}} & 0 & 0 & 0 & 0 & 0 & 0 & 0 & 0\\ 0 & \frac{F^{\left(k\tau \right)\alpha_{\;i}}}{H_{2}} & 0 & 0 & 0 & 0 & 0 & 0 & 0\\ 0 & 0 & \frac{F^{\left(k\tau \right)\alpha_{\;i}}}{H_{1}} & \frac{F^{\left(k\tau \right)\alpha_{\;i}}}{H_{2}} & 0 & 0 & 0 & 0 & 0\\ 0 & 0 & 0 & 0 & \frac{F^{\left(k\tau \right)\alpha_{\;i}}}{H_{1}} & 0 & \frac{\partial F^{\left(k\tau \right)\alpha_{\;i}}}{\partial \zeta } & 0 & 0\\ 0 & 0 & 0 & 0 & 0 & \frac{F^{\left(k\tau \right)\alpha_{\;i}}}{H_{2}} & 0 & \frac{\partial F^{\left(k\tau \right)\alpha_{\;i}}}{\partial \zeta } & 0\\ 0 & 0 & 0 & 0 & 0 & 0 & 0 & 0 & \frac{\partial F^{\left(k\tau \right)\alpha_{\;i}}}{\partial \zeta } \end{array}\right]$$

## 4. Constitutive Relationship

In the present section, the constitutive behavior of the three-dimensional shell solid is described within the two-dimensional ESL model employing higher-order theories. Referring to an arbitrary k-th layer of the solid, a three-dimensional linear elastic constitutive relation is considered, defined as follows [16]:(17)$$\sigma^{\left(k\right)}={\bar{E}}^{\left(k\right)}\varepsilon^{\left(k\right)}\;\;\;\;↔\;\;\;\;\left[\begin{array}{c} \sigma_{\;1}^{\left(k\right)}\\ \sigma_{\;2}^{\left(k\right)}\\ \tau_{\;12}^{\left(k\right)}\\ \tau_{\;13}^{\left(k\right)}\\ \tau_{\;23}^{\left(k\right)}\\ \sigma_{\;3}^{\left(k\right)} \end{array}\right]=\left[\begin{array}{cccccc} {\bar{E}}_{\;11}^{\left(k\right)} & {\bar{E}}_{\;12}^{\left(k\right)} & {\bar{E}}_{\;16}^{\left(k\right)} & {\bar{E}}_{\;14}^{\left(k\right)} & {\bar{E}}_{\;15}^{\left(k\right)} & {\bar{E}}_{\;13}^{\left(k\right)}\\ {\bar{E}}_{\;12}^{\left(k\right)} & {\bar{E}}_{\;22}^{\left(k\right)} & {\bar{E}}_{\;26}^{\left(k\right)} & {\bar{E}}_{\;24}^{\left(k\right)} & {\bar{E}}_{\;25}^{\left(k\right)} & {\bar{E}}_{\;23}^{\left(k\right)}\\ {\bar{E}}_{\;16}^{\left(k\right)} & {\bar{E}}_{\;26}^{\left(k\right)} & {\bar{E}}_{\;66}^{\left(k\right)} & {\bar{E}}_{\;46}^{\left(k\right)} & {\bar{E}}_{\;56}^{\left(k\right)} & {\bar{E}}_{\;36}^{\left(k\right)}\\ {\bar{E}}_{\;14}^{\left(k\right)} & {\bar{E}}_{\;24}^{\left(k\right)} & {\bar{E}}_{\;46}^{\left(k\right)} & {\bar{E}}_{\;44}^{\left(k\right)} & {\bar{E}}_{\;45}^{\left(k\right)} & {\bar{E}}_{\;34}^{\left(k\right)}\\ {\bar{E}}_{\;15}^{\left(k\right)} & {\bar{E}}_{\;25}^{\left(k\right)} & {\bar{E}}_{\;56}^{\left(k\right)} & {\bar{E}}_{\;45}^{\left(k\right)} & {\bar{E}}_{\;55}^{\left(k\right)} & {\bar{E}}_{\;35}^{\left(k\right)}\\ {\bar{E}}_{\;13}^{\left(k\right)} & {\bar{E}}_{\;23}^{\left(k\right)} & {\bar{E}}_{\;36}^{\left(k\right)} & {\bar{E}}_{\;34}^{\left(k\right)} & {\bar{E}}_{\;35}^{\left(k\right)} & {\bar{E}}_{\;33}^{\left(k\right)} \end{array}\right]\left[\begin{array}{c} \varepsilon_{\;1}^{\left(k\right)}\\ \varepsilon_{\;2}^{\left(k\right)}\\ \gamma_{\;12}^{\left(k\right)}\\ \gamma_{\;13}^{\left(k\right)}\\ \gamma_{\;23}^{\left(k\right)}\\ \varepsilon_{\;3}^{\left(k\right)} \end{array}\right]\;\;\;\;\;k=1,\ldots ,l$$

In the previous relation, ${\bar{E}}^{\left(k\right)}$ denotes the three-dimensional stiffness matrix, whose generic component ${\bar{E}}_{\;ij}^{\left(k\right)}$ with i,j=1,…,6 relates the i-th element of the stress vector, denoted by $\sigma^{\left(k\right)}\left(\alpha_{1},\alpha_{2},\zeta \right)$, to the j-th element of the strain vector $\varepsilon^{\left(k\right)}\left(\alpha_{1},\alpha_{2},\zeta \right)$. The constitutive behavior of each lamina is usually provided not in the geometric reference system, as happens in Equation (17), but in the material reference system $O^{\prime }{\hat{\alpha }}_{\;1}{\hat{\alpha }}_{\;2}{\hat{\alpha }}_{\;3}$, whose axes are oriented along the material symmetry directions. For this reason, the equation reported in the following should be considered, where ${\hat{\sigma }}^{\left(k\right)}$ and ${\hat{\varepsilon }}^{\left(k\right)}$ are the vectors of the three-dimensional stress and strain components, respectively, written with respect to $O^{\prime }{\hat{\alpha }}_{\;1}{\hat{\alpha }}_{\;2}{\hat{\alpha }}_{\;3}$ material reference system, whereas $E^{\left(k\right)}$ is the corresponding stiffness matrix:(18)$${\hat{\sigma }}^{\left(k\right)}=E^{\left(k\right)}{\hat{\varepsilon }}^{\left(k\right)}\;\;\;\;↔\;\;\;\;\left[\begin{array}{c} {\hat{\sigma }}_{\;1}^{\left(k\right)}\\ {\hat{\sigma }}_{\;2}^{\left(k\right)}\\ {\hat{\tau }}_{\;12}^{\left(k\right)}\\ {\hat{\tau }}_{\;13}^{\left(k\right)}\\ {\hat{\tau }}_{\;23}^{\left(k\right)}\\ {\hat{\sigma }}_{\;3}^{\left(k\right)} \end{array}\right]=\left[\begin{array}{cccccc} E_{\;11}^{\left(k\right)} & E_{\;12}^{\left(k\right)} & E_{\;16}^{\left(k\right)} & E_{\;14}^{\left(k\right)} & E_{\;15}^{\left(k\right)} & E_{\;13}^{\left(k\right)}\\ E_{\;12}^{\left(k\right)} & E_{\;22}^{\left(k\right)} & E_{\;26}^{\left(k\right)} & E_{\;24}^{\left(k\right)} & E_{\;25}^{\left(k\right)} & E_{\;23}^{\left(k\right)}\\ E_{\;16}^{\left(k\right)} & E_{\;26}^{\left(k\right)} & E_{\;66}^{\left(k\right)} & E_{\;46}^{\left(k\right)} & E_{\;56}^{\left(k\right)} & E_{\;36}^{\left(k\right)}\\ E_{\;14}^{\left(k\right)} & E_{\;24}^{\left(k\right)} & E_{\;46}^{\left(k\right)} & E_{\;44}^{\left(k\right)} & E_{\;45}^{\left(k\right)} & E_{\;34}^{\left(k\right)}\\ E_{\;15}^{\left(k\right)} & E_{\;25}^{\left(k\right)} & E_{\;56}^{\left(k\right)} & E_{\;45}^{\left(k\right)} & E_{\;55}^{\left(k\right)} & E_{\;35}^{\left(k\right)}\\ E_{\;13}^{\left(k\right)} & E_{\;23}^{\left(k\right)} & E_{\;36}^{\left(k\right)} & E_{\;34}^{\left(k\right)} & E_{\;35}^{\left(k\right)} & E_{\;33}^{\left(k\right)} \end{array}\right]\left[\begin{array}{c} {\hat{\varepsilon }}_{\;1}^{\left(k\right)}\\ {\hat{\varepsilon }}_{\;2}^{\left(k\right)}\\ {\hat{\gamma }}_{\;12}^{\left(k\right)}\\ {\hat{\gamma }}_{\;13}^{\left(k\right)}\\ {\hat{\gamma }}_{\;23}^{\left(k\right)}\\ {\hat{\varepsilon }}_{\;3}^{\left(k\right)} \end{array}\right]\;\;\;\;\;k=1,\ldots ,l$$

In the previous equation, $E_{\;ij}^{\left(k\right)}$ with i,j=1,…,6 are the elements of the matrix $E^{\left(k\right)}$. More specifically, they can be taken as equal to the three-dimensional stiffness components of the material of the -th layer, namely $E_{\;ij}^{\left(k\right)}=C_{\;ij}^{\left(k\right)}$. However, when the kinematic field assumption of Equation (8) neglects the stretching effect, $E_{\;ij}^{\left(k\right)}$ turn out to be the reduced elastic coefficients $Q_{\;ij}^{\left(k\right)}$, which are calculated from the three-dimensional ones according to what is exerted in Ref. [16].

Finally, Equation (18) can revert to Equation (17) if the stiffness matrix $E^{\left(k\right)}$ is rotated by means of the transformation matrix $T^{\left(k\right)}$, as defined in Ref. [16]. One gets:(19)$${\bar{E}}^{\left(k\right)}=T^{\left(k\right)}E^{\left(k\right)}{\left(T^{\left(k\right)}\right)}^{T}$$

In the present study, the material axis ${\hat{\alpha }}_{\;3}$ is taken along the thickness direction of the shell; therefore, the equivalence ${\hat{\alpha }}_{\;3}=\zeta$ is considered. As a consequence, the rotation matrix $T^{\left(k\right)}$ depends only on the angle $\vartheta^{\left(k\right)}$ between $\alpha_{\;1}$ and ${\hat{\alpha }}_{\;1}$ reference axes such that $T^{\left(k\right)}=T^{\left(k\right)}\left(\vartheta^{\left(k\right)}\right)$. Introducing the generalized higher-order expansion of the kinematic relations of Equation (14) in the three-dimensional constitutive relationship of Equation (17), the following higher-order constitutive relationship comes out:(20)$$\sigma^{\left(k\right)}={\bar{E}}^{\left(k\right)}\varepsilon^{\left(k\right)}=\sum_{\eta =0}^{N+1}\sum_{j=1}^{3}{\bar{E}}^{\left(k\right)}Z{\;}^{\left(k\eta \right)\alpha_{\;j}}\varepsilon^{\left(\eta \right)\alpha_{\;j}}\;\;\;\mathrm{for}\;\;\;k=1,\ldots ,l$$

At this point, the previous relation can be used for the evaluation of the virtual variation δ Φ of the elastic strain energy of the shell:(21)$$\delta \;\Phi =\sum_{k=1}^{l}\int_{\alpha_{\;1}}\int_{\alpha_{\;2}}\int_{\zeta_{\;k}}^{\zeta_{\;k+1}}{\left(\delta \varepsilon^{\left(k\right)}\right)}^{T}\sigma^{\left(k\right)}A_{\;1}A_{\;2}H_{\;1}H_{\;2}d\alpha_{\;1}d\alpha_{\;2}d\zeta$$

The computation of the integrals of Equation (21) in the closed interval $\left[\zeta_{k},\zeta_{k+1}\right]$ allows one to introduce the generalized stress resultants, which are collected for each τ=0,…,N+1 and i=1,2,3 in the generalized stress resultant vector $S^{\left(\tau \right)\alpha_{i}}={\left[N_{1}^{\left(\tau \right)\alpha_{\;i}}\;\;N_{2}^{\left(\tau \right)\alpha_{\;i}}\;\;N_{12}^{\left(\tau \right)\alpha_{\;i}}\;\;N_{21}^{\left(\tau \right)\alpha_{\;i}}\;\;T_{1}^{\left(\tau \right)\alpha_{\;i}}\;\;T_{2}^{\left(\tau \right)\alpha_{\;i}}\;\;P_{1}^{\left(\tau \right)\alpha_{\;i}}\;\;P_{2}^{\left(\tau \right)\alpha_{\;i}}\;\;S_{3}^{\left(\tau \right)\alpha_{\;i}}\right]}^{T}$ [16]:(22)$$\delta \;\Phi =\sum_{\tau =0}^{N+1}\sum_{i=1}^{3}\int_{\alpha_{\;1}}\int_{\alpha_{\;2}}{\left(\delta \varepsilon^{\left(\tau \right)\alpha_{\;i}}\right)}^{T}S^{\left(\tau \right)\alpha_{\;i}}A_{\;1}A_{\;2}d\alpha_{\;1}d\alpha_{\;2}$$

Finally, the higher-order ESL elastic constitutive relationship can be written in terms of $S^{\left(\tau \right)\alpha_{i}}$ and $\varepsilon^{\left(\tau \right)\alpha_{i}}$ by substituting Equation (20) in the final expression of the virtual variation of the elastic strain energy δ Φ, as reported in Equation (22). One gets:(23)$$S^{\left(\tau \right)\alpha_{i}}=\sum_{\eta =0}^{N+1}\sum_{j=1}^{3}A^{\left(\tau \eta \right)\alpha_{i}\alpha_{j}}\varepsilon^{\left(\eta \right)\alpha_{j}}\;\;\;\;\mathrm{for}\;\;\;\tau =0,\ldots ,N+1,i=1,2,3$$

In the previous equation, $A^{\left(\tau \eta \right)\alpha_{i}\alpha_{j}}$ denotes the generalized higher-order constitutive operator, which is defined for each τ,η=0,…,N+1 and i,j=1,2,3 as follows [16]:(24)$$A^{\left(\tau \eta \right)\alpha_{i}\alpha_{j}}=\sum_{k=1}^{l}\int_{\zeta_{\;k}}^{\zeta_{\;k+1}}{\left(Z^{\left(k\tau \right)\alpha_{\;i}}\right)}^{T}{\bar{E}}^{\left(k\right)}Z^{\left(k\eta \right)\alpha_{\;j}}H_{\;1}H_{\;2}d\zeta =\left[\begin{array}{cc} A^{\left(\tau \eta \right)\left[00\right]\;\alpha_{\;i}\alpha_{\;j}} & A^{\left(\tau \eta \right)\left[01\right]\;\;\alpha_{\;i}\alpha_{\;j}}\\ A^{\left(\tau \eta \right)\left[10\right]\;\;\alpha_{\;i}\alpha_{\;j}} & A^{\left(\tau \eta \right)\left[11\right]\;\;\alpha_{\;i}\alpha_{\;j}} \end{array}\right]$$

For the sake of clarity, the sub-matrices $A^{\left(\tau \eta \right)\left[00\right]\;\alpha_{i}\alpha_{j}}$, $A^{\left(\tau \eta \right)\left[01\right]\;\alpha_{i}\alpha_{j}}$, $A^{\left(\tau \eta \right)\left[10\right]\;\alpha_{i}\alpha_{j}}$, $A^{\left(\tau \eta \right)\left[11\right]\;\alpha_{i}\alpha_{j}}$ are reported below in an extended form:(25)$$A^{\left(\tau \eta \right)\left[00\right]\;\alpha_{i}\alpha_{j}}=\left[\begin{array}{cccccc} A_{\;11\left(20\right)11}^{\left(\tau \eta \right)\left[00\right]\alpha_{i}\alpha_{j}} & A_{\;12\left(11\right)12}^{\left(\tau \eta \right)\left[00\right]\alpha_{i}\alpha_{j}} & A_{\;16\left(20\right)13}^{\left(\tau \eta \right)\left[00\right]\alpha_{i}\alpha_{j}} & A_{\;16\left(11\right)14}^{\left(\tau \eta \right)\left[00\right]\alpha_{i}\alpha_{j}} & A_{\;14\left(20\right)}^{\left(\tau \eta \right)\left[00\right]\alpha_{i}\alpha_{j}} & A_{\;15\left(11\right)}^{\left(\tau \eta \right)\left[00\right]\alpha_{i}\alpha_{j}}\\ A_{\;12\left(11\right)}^{\left(\tau \eta \right)\left[00\right]\alpha_{i}\alpha_{j}} & A_{\;22\left(02\right)}^{\left(\tau \eta \right)\left[00\right]\alpha_{i}\alpha_{j}} & A_{\;26\left(11\right)}^{\left(\tau \eta \right)\left[00\right]\alpha_{i}\alpha_{j}} & A_{\;26\left(02\right)}^{\left(\tau \eta \right)\left[00\right]\alpha_{i}\alpha_{j}} & A_{\;24\left(11\right)}^{\left(\tau \eta \right)\left[00\right]\alpha_{i}\alpha_{j}} & A_{\;25\left(02\right)}^{\left(\tau \eta \right)\left[00\right]\alpha_{i}\alpha_{j}}\\ A_{\;16\left(20\right)}^{\left(\tau \eta \right)\left[00\right]\alpha_{i}\alpha_{j}} & A_{\;26\left(11\right)}^{\left(\tau \eta \right)\left[00\right]\alpha_{i}\alpha_{j}} & A_{\;66\left(20\right)}^{\left(\tau \eta \right)\left[00\right]\alpha_{i}\alpha_{j}} & A_{\;66\left(11\right)}^{\left(\tau \eta \right)\left[00\right]\alpha_{i}\alpha_{j}} & A_{\;46\left(20\right)}^{\left(\tau \eta \right)\left[00\right]\alpha_{i}\alpha_{j}} & A_{\;56\left(11\right)}^{\left(\tau \eta \right)\left[00\right]\alpha_{i}\alpha_{j}}\\ A_{\;16\left(11\right)}^{\left(\tau \eta \right)\left[00\right]\alpha_{i}\alpha_{j}} & A_{\;26\left(02\right)}^{\left(\tau \eta \right)\left[00\right]\alpha_{i}\alpha_{j}} & A_{\;66\left(11\right)}^{\left(\tau \eta \right)\left[00\right]\alpha_{i}\alpha_{j}} & A_{\;66\left(02\right)}^{\left(\tau \eta \right)\left[00\right]\alpha_{i}\alpha_{j}} & A_{\;46\left(11\right)}^{\left(\tau \eta \right)\left[00\right]\alpha_{i}\alpha_{j}} & A_{\;56\left(02\right)}^{\left(\tau \eta \right)\left[00\right]\alpha_{i}\alpha_{j}}\\ A_{\;14\left(20\right)}^{\left(\tau \eta \right)\left[00\right]\alpha_{i}\alpha_{j}} & A_{\;24\left(11\right)}^{\left(\tau \eta \right)\left[00\right]\alpha_{i}\alpha_{j}} & A_{\;46\left(20\right)}^{\left(\tau \eta \right)\left[00\right]\alpha_{i}\alpha_{j}} & A_{\;46\left(11\right)}^{\left(\tau \eta \right)\left[00\right]\alpha_{i}\alpha_{j}} & A_{\;44\left(20\right)}^{\left(\tau \eta \right)\left[00\right]\alpha_{i}\alpha_{j}} & A_{\;45\left(11\right)}^{\left(\tau \eta \right)\left[00\right]\alpha_{i}\alpha_{j}}\\ A_{\;15\left(11\right)}^{\left(\tau \eta \right)\left[00\right]\alpha_{i}\alpha_{j}} & A_{\;25\left(02\right)}^{\left(\tau \eta \right)\left[00\right]\alpha_{i}\alpha_{j}} & A_{\;56\left(11\right)}^{\left(\tau \eta \right)\left[00\right]\alpha_{i}\alpha_{j}} & A_{\;56\left(02\right)}^{\left(\tau \eta \right)\left[00\right]\alpha_{i}\alpha_{j}} & A_{\;45\left(11\right)}^{\left(\tau \eta \right)\left[00\right]\alpha_{i}\alpha_{j}} & A_{\;55\left(02\right)}^{\left(\tau \eta \right)\left[00\right]\alpha_{i}\alpha_{j}} \end{array}\right]$$
(26)$$A^{\left(\tau \eta \right)\left[01\right]\;\alpha_{i}\alpha_{j}}=\left[\begin{array}{ccc} A_{\;14\left(10\right)}^{\left(\tau \eta \right)\left[01\right]\alpha_{i}\alpha_{j}} & A_{\;15\left(10\right)}^{\left(\tau \eta \right)\left[01\right]\alpha_{i}\alpha_{j}} & A_{\;13\left(10\right)}^{\left(\tau \eta \right)\left[01\right]\alpha_{i}\alpha_{j}}\\ A_{\;24\left(01\right)}^{\left(\tau \eta \right)\left[01\right]\alpha_{i}\alpha_{j}} & A_{\;25\left(01\right)}^{\left(\tau \eta \right)\left[01\right]\alpha_{i}\alpha_{j}} & A_{\;23\left(01\right)}^{\left(\tau \eta \right)\left[01\right]\alpha_{i}\alpha_{j}}\\ A_{\;46\left(10\right)}^{\left(\tau \eta \right)\left[01\right]\alpha_{i}\alpha_{j}} & A_{\;56\left(10\right)}^{\left(\tau \eta \right)\left[01\right]\alpha_{i}\alpha_{j}} & A_{\;36\left(10\right)}^{\left(\tau \eta \right)\left[01\right]\alpha_{i}\alpha_{j}}\\ A_{\;46\left(01\right)}^{\left(\tau \eta \right)\left[01\right]\alpha_{i}\alpha_{j}} & A_{\;56\left(01\right)}^{\left(\tau \eta \right)\left[01\right]\alpha_{i}\alpha_{j}} & A_{\;36\left(01\right)}^{\left(\tau \eta \right)\left[01\right]\alpha_{i}\alpha_{j}}\\ A_{\;44\left(10\right)}^{\left(\tau \eta \right)\left[01\right]\alpha_{i}\alpha_{j}} & A_{\;45\left(10\right)}^{\left(\tau \eta \right)\left[01\right]\alpha_{i}\alpha_{j}} & A_{\;34\left(10\right)}^{\left(\tau \eta \right)\left[01\right]\alpha_{i}\alpha_{j}}\\ A_{\;45\left(01\right)}^{\left(\tau \eta \right)\left[01\right]\alpha_{i}\alpha_{j}} & A_{\;55\left(01\right)}^{\left(\tau \eta \right)\left[01\right]\alpha_{i}\alpha_{j}} & A_{\;35\left(01\right)}^{\left(\tau \eta \right)\left[01\right]\alpha_{i}\alpha_{j}} \end{array}\right]$$
(27)$$A^{\left(\tau \eta \right)\left[10\right]\;\alpha_{i}\alpha_{j}}=\left[\begin{array}{cccccc} A_{\;14\left(10\right)}^{\left(\tau \eta \right)\left[10\right]\alpha_{i}\alpha_{j}} & A_{\;24\left(01\right)}^{\left(\tau \eta \right)\left[10\right]\alpha_{i}\alpha_{j}} & A_{\;46\left(10\right)}^{\left(\tau \eta \right)\left[10\right]\alpha_{i}\alpha_{j}} & A_{\;46\left(01\right)}^{\left(\tau \eta \right)\left[10\right]\alpha_{i}\alpha_{j}} & A_{\;44\left(10\right)}^{\left(\tau \eta \right)\left[10\right]\alpha_{i}\alpha_{j}} & A_{\;45\left(01\right)}^{\left(\tau \eta \right)\left[10\right]\alpha_{i}\alpha_{j}}\\ A_{\;15\left(10\right)}^{\left(\tau \eta \right)\left[10\right]\alpha_{i}\alpha_{j}} & A_{\;25\left(01\right)}^{\left(\tau \eta \right)\left[10\right]\alpha_{i}\alpha_{j}} & A_{\;56\left(10\right)}^{\left(\tau \eta \right)\left[10\right]\alpha_{i}\alpha_{j}} & A_{\;56\left(01\right)}^{\left(\tau \eta \right)\left[10\right]\alpha_{i}\alpha_{j}} & A_{\;45\left(10\right)}^{\left(\tau \eta \right)\left[10\right]\alpha_{i}\alpha_{j}} & A_{\;55\left(01\right)}^{\left(\tau \eta \right)\left[10\right]\alpha_{i}\alpha_{j}}\\ A_{\;13\left(10\right)}^{\left(\tau \eta \right)\left[10\right]\alpha_{i}\alpha_{j}} & A_{\;23\left(01\right)}^{\left(\tau \eta \right)\left[10\right]\alpha_{i}\alpha_{j}} & A_{\;36\left(10\right)}^{\left(\tau \eta \right)\left[10\right]\alpha_{i}\alpha_{j}} & A_{\;36\left(01\right)}^{\left(\tau \eta \right)\left[10\right]\alpha_{i}\alpha_{j}} & A_{\;34\left(10\right)}^{\left(\tau \eta \right)\left[10\right]\alpha_{i}\alpha_{j}} & A_{\;35\left(01\right)}^{\left(\tau \eta \right)\left[10\right]\alpha_{i}\alpha_{j}} \end{array}\right]$$
(28)$$A^{\left(\tau \eta \right)\left[11\right]\;\alpha_{i}\alpha_{j}}=\left[\begin{array}{ccc} A_{\;44\left(00\right)}^{\left(\tau \eta \right)\left[11\right]\alpha_{i}\alpha_{j}} & A_{\;45\left(00\right)}^{\left(\tau \eta \right)\left[11\right]\alpha_{i}\alpha_{j}} & A_{\;34\left(00\right)}^{\left(\tau \eta \right)\left[11\right]\alpha_{i}\alpha_{j}}\\ A_{\;45\left(00\right)}^{\left(\tau \eta \right)\left[11\right]\alpha_{i}\alpha_{j}} & A_{\;55\left(00\right)}^{\left(\tau \eta \right)\left[11\right]\alpha_{i}\alpha_{j}} & A_{\;35\left(00\right)}^{\left(\tau \eta \right)\left[11\right]\alpha_{i}\alpha_{j}}\\ A_{\;34\left(00\right)}^{\left(\tau \eta \right)\left[11\right]\alpha_{i}\alpha_{j}} & A_{\;35\left(00\right)}^{\left(\tau \eta \right)\left[11\right]\alpha_{i}\alpha_{j}} & A_{\;33\left(00\right)}^{\left(\tau \eta \right)\left[11\right]\alpha_{i}\alpha_{j}} \end{array}\right]$$

The generalized elastic coefficients of Equations (25)–(28) can be computed with the following condensed expression, setting the definitions $\partial^{0}F^{\left(k\tau \right)\alpha_{i}}/\partial \zeta^{0}=F^{\left(k\tau \right)\alpha_{i}}$ and $\partial^{0}F^{\left(k\eta \right)\alpha_{j}}/\partial \zeta^{0}=F^{\left(k\eta \right)\alpha_{j}}$:(29)$$\begin{array}{cc} A_{nm\;\;\left(pq\right)}^{\left(\tau \eta \right)\left[fg\right]\alpha_{\;i}\alpha_{\;j}}=\sum_{k=1}^{l}\int_{\zeta_{k}}^{\zeta_{k+1}}{\bar{B}}_{nm}^{\left(k\right)}\frac{\partial^{f}F^{\left(k\eta \right)\alpha_{\;j}}}{\partial \zeta^{f}}\frac{\partial^{g}F^{\left(k\tau \right)\alpha_{\;i}}}{\partial \zeta^{g}}\frac{H_{\;1}H_{\;2}}{H_{\;1}^{p}H_{\;2}^{q}}d\zeta & \;\;\;\mathrm{for}\;\begin{array}{l} \tau ,\eta =0,\ldots ,N+1,\\ n,m=1,\ldots ,6,\\ p,q=0,1,2,\\ \alpha_{\;i},\alpha_{\;j}=\alpha_{\;1},\alpha_{\;2},\alpha_{\;3},\\ f,g=0,1 \end{array} \end{array}$$

The quantities ${\bar{B}}_{\;nm}^{\left(k\right)}$ with n,m=1,…,6 occurring in the previous expression denote the three-dimensional elastic stiffness coefficients ${\bar{E}}_{\;nm}^{\left(k\right)}$ of Equation (17). As can be seen, their value is corrected with the shear correction factor, denoted by $\kappa \left(\zeta \right)$, in order to consider the effects of shear stresses to the global deflection of the structure when lower order displacement field assumptions are considered:(30)$${\bar{B}}_{\;nm}^{\left(k\right)}=\left\{\begin{array}{c} {\bar{E}}_{\;nm}^{\left(k\right)}\\ \kappa \left(\zeta \right){\bar{E}}_{\;nm}^{\left(k\right)} \end{array}\;\;\;\;\;\;\mathrm{for}\;\;\;\;\;\begin{array}{l} n,m=1,2,3,6,\\ n,m=4,5 \end{array}\right.$$

Accordingly, when a linear distribution of the displacement field components is considered in Equation (8), the value $\kappa \left(\zeta \right)=5/6$ is assumed, whereas in the case of higher-order theories, a unitary value $\kappa \left(\zeta \right)=1$ is assigned to this quantity. The value $\kappa \left(\zeta \right)=5/6$ is here assumed, as in classical formulations for beams with rectangular cross-sections. The higher-order constitutive relation of Equation (23) can be written in terms of the kinematic field assumption of Equation (8) in order to provide a relationship between the stress resultant vector $S^{\left(\tau \right)\alpha_{i}}$ and the generalized displacement field vector $u^{\left(\tau \right)}$. Substituting the definition of the vector $\varepsilon^{\left(\tau \right)\alpha_{i}}$ of the generalized strain components of Equation (15) in the higher-order constitutive relation of Equation (23), the relation reported in the following is obtained:(31)$$S^{\left(\tau \right)\alpha_{i}}=\sum_{\eta =0}^{N+1}\sum_{j=1}^{3}A^{\left(\tau \eta \right)\alpha_{\;i}\alpha_{\;j}}D_{\Omega }^{\alpha_{\;j}}u^{\left(\eta \right)}=\sum_{\eta =0}^{N+1}\sum_{j=1}^{3}O^{\left(\tau \eta \right)\alpha_{\;i}\alpha_{\;j}}u^{\left(\eta \right)}\;\;\;\;\mathrm{for}\;\begin{array}{l} \tau =0,\ldots ,N+1,\\ i=1,2,3 \end{array}$$

The matrix $O^{\left(\tau \eta \right)\alpha_{i}\alpha_{j}}$ of size 9×3, defined for each τ,η=0,…,N+1 and $\alpha_{\;i},\alpha_{\;j}=\alpha_{\;1},\alpha_{\;2},\alpha_{\;3}$ is reported in the following in an extended form:(32)$$O^{\left(\tau \eta \right)\alpha_{i}\alpha_{j}}=\left[\begin{array}{ccc} O_{\;11}^{\left(\tau \eta \right)\alpha_{i}\alpha_{1}} & O_{\;12}^{\left(\tau \eta \right)\alpha_{i}\alpha_{2}} & O_{\;13}^{\left(\tau \eta \right)\alpha_{i}\alpha_{3}}\\ O_{\;21}^{\left(\tau \eta \right)\alpha_{i}\alpha_{1}} & O_{\;22}^{\left(\tau \eta \right)\alpha_{i}\alpha_{2}} & O_{\;23}^{\left(\tau \eta \right)\alpha_{i}\alpha_{3}}\\ O_{\;31}^{\left(\tau \eta \right)\alpha_{i}\alpha_{1}} & O_{\;32}^{\left(\tau \eta \right)\alpha_{i}\alpha_{2}} & O_{\;33}^{\left(\tau \eta \right)\alpha_{i}\alpha_{3}}\\ O_{\;41}^{\left(\tau \eta \right)\alpha_{i}\alpha_{1}} & O_{\;42}^{\left(\tau \eta \right)\alpha_{i}\alpha_{2}} & O_{\;43}^{\left(\tau \eta \right)\alpha_{i}\alpha_{3}}\\ O_{\;51}^{\left(\tau \eta \right)\alpha_{i}\alpha_{1}} & O_{\;52}^{\left(\tau \eta \right)\alpha_{i}\alpha_{2}} & O_{\;53}^{\left(\tau \eta \right)\alpha_{i}\alpha_{3}}\\ O_{\;61}^{\left(\tau \eta \right)\alpha_{i}\alpha_{1}} & O_{\;62}^{\left(\tau \eta \right)\alpha_{i}\alpha_{2}} & O_{\;63}^{\left(\tau \eta \right)\alpha_{i}\alpha_{3}}\\ O_{\;71}^{\left(\tau \eta \right)\alpha_{i}\alpha_{1}} & O_{\;72}^{\left(\tau \eta \right)\alpha_{i}\alpha_{2}} & O_{\;73}^{\left(\tau \eta \right)\alpha_{i}\alpha_{3}}\\ O_{\;81}^{\left(\tau \eta \right)\alpha_{i}\alpha_{1}} & O_{\;82}^{\left(\tau \eta \right)\alpha_{i}\alpha_{2}} & O_{\;83}^{\left(\tau \eta \right)\alpha_{i}\alpha_{3}}\\ O_{\;91}^{\left(\tau \eta \right)\alpha_{i}\alpha_{1}} & O_{\;92}^{\left(\tau \eta \right)\alpha_{i}\alpha_{2}} & O_{\;93}^{\left(\tau \eta \right)\alpha_{i}\alpha_{3}} \end{array}\right]\;\;\;\;\mathrm{for}\;\begin{array}{l} \tau ,\eta =0,\ldots ,N+1,\\ i,j=1,2,3\;\;\;\;\;\;\;\;\;\;\;\; \end{array}$$

The complete expression of the coefficients $O_{gr}^{\left(\tau \eta \right)\alpha_{i}\alpha_{j}}$ with g=1,…,9 and r=1,2,3 can be found in Ref. [16].

## 5. Governing Equations

Once the kinematic and constitutive relations have been presented, the fundamental governing equations are derived for the linear static analysis of doubly-curved shell structures. Following an energetic procedure, the equilibrium configuration of the solid is derived from the minimum potential energy principle, taking into account the elastic deformation energy of the system, denoted by Φ, and the virtual work $L_{\;e}$ of external loads. If the virtual variation of each physical quantity is denoted by δ, the following relation is considered [16]:(33)$$\delta \;\Phi -\delta \;L_{\;e}=0$$

As shown in Equation (22), the virtual variation δ Φ of the elastic strain energy is written in terms of the virtual variation of the vector $\delta \varepsilon^{\left(\tau \right)\alpha_{i}}$ of the generalized strain components. Applying the integration by parts rule, an expression of the variation δ Φ is obtained in terms of the virtual variation of the generalized displacement field components $\delta u_{\;1}^{\left(\tau \right)},\delta u_{\;2}^{\left(\tau \right)},\delta u_{\;3}^{\left(\tau \right)}$:(34)$$\begin{array}{l} \delta \;\Phi =\sum_{\tau =0}^{N+1}\int_{\alpha_{1}}\int_{\alpha_{2}}\left(-\left(\frac{\partial \left(N_{\;1}^{\left(\tau \right)\alpha_{1}}A_{\;2}\right)}{\partial \alpha_{\;1}}+\frac{\partial \left(N_{\;21}^{\left(\tau \right)\alpha_{1}}A_{\;1}\right)}{\partial \alpha_{\;2}}+N_{\;12}^{\left(\tau \right)\alpha_{1}}\frac{\partial A_{\;1}}{\partial \alpha_{\;2}}-N_{\;2}^{\left(\tau \right)\alpha_{1}}\frac{\partial A_{\;2}}{\partial \alpha_{\;1}}+\left(\frac{T_{\;1}^{\left(\tau \right)\alpha_{1}}}{R_{\;1}}-P_{\;1}^{\left(\tau \right)\alpha_{1}}\right)A_{\;1}A_{\;2}\right)\delta u_{\;1}^{\left(\tau \right)}\right.+\\ \;\;\;\;\;\;\;\;\;-\left(\frac{\partial \left(N_{\;2}^{\left(\tau \right)\alpha_{2}}A_{\;1}\right)}{\partial \alpha_{\;2}}+\frac{\partial \left(N_{\;12}^{\left(\tau \right)\alpha_{2}}A_{\;2}\right)}{\partial \alpha_{\;1}}+N_{21}^{\left(\tau \right)\alpha_{2}}\frac{\partial A_{\;2}}{\partial \alpha_{\;1}}-N_{1}^{\left(\tau \right)\alpha_{2}}\frac{\partial A_{\;1}}{\partial \alpha_{\;2}}+\left(\frac{T_{\;2}^{\left(\tau \right)\alpha_{2}}}{R_{\;2}}-P_{\;2}^{\left(\tau \right)\alpha_{2}}\right)A_{1}A_{2}\right)\delta u_{2}^{\left(\tau \right)}+\\ \left.\;\;\;\;\;\;\;\;-\left(\frac{\partial \left(T_{\;1}^{\left(\tau \right)\alpha_{3}}A_{\;2}\right)}{\partial \alpha_{\;1}}+\frac{\partial \left(T_{\;2}^{\left(\tau \right)\alpha_{3}}A_{\;1}\right)}{\partial \alpha_{\;2}}-\left(\frac{N_{1}^{\left(\tau \right)\alpha_{3}}}{R_{\;1}}+\frac{N_{2}^{\left(\tau \right)\alpha_{3}}}{R_{\;2}}\right)A_{\;1}A_{\;2}-S_{3}^{\left(\tau \right)\alpha_{3}}A_{\;1}A_{\;2}\right)\delta u_{3}^{\left(\tau \right)}\right)d\alpha_{1}d\alpha_{2}+\\ \;\;\;\;\;\;\;\;\;+\sum_{\tau =0}^{N+1}\oint_{\alpha_{1}}\left(N_{\;21}^{\left(\tau \right)\alpha_{1}}\delta u_{\;1}^{\left(\tau \right)}+N_{\;2}^{\left(\tau \right)\alpha_{2}}\delta u_{\;2}^{\left(\tau \right)}+T_{\;2}^{\left(\tau \right)\alpha_{3}}\delta u_{\;3}^{\left(\tau \right)}\right)A_{\;1}d\alpha_{\;1}+\\ \;\;\;\;\;\;\;\;\;+\;\sum_{\tau =0}^{N+1}\oint_{\alpha_{2}}\left(N_{\;1}^{\left(\tau \right)\alpha_{1}}\delta u_{\;1}^{\left(\tau \right)}+N_{\;12}^{\left(\tau \right)\alpha_{2}}\delta u_{\;2}^{\left(\tau \right)}+T_{\;1}^{\left(\tau \right)\alpha_{3}}\delta u_{\;3}^{\left(\tau \right)}\right)A_{\;2}d\alpha_{\;2} \end{array}$$

The virtual work of external actions, denoted by $\delta L_{\;e}^{\left(3D\right)}$, is computed as the sum of the virtual work of the actions $q_{\;ia}^{\left(+\right)}=q_{\;ia}^{\left(1\right)}$ and $q_{\;ia}^{\left(-\right)}=q_{\;ia}^{\left(2\right)}$ with i=1,2,3 acting at the top $\left(j=1\right)$ and the bottom $\left(j=2\right)$ of the shell, respectively. Referring to a doubly-curved three-dimensional solid, one gets:(35)$$\delta L_{\;e}^{\left(3D\right)}=\int_{\alpha_{1}}\int_{\alpha_{\;2}}\left(\sum_{j=1}^{2}\left(\sum_{i=1}^{3}q_{\;ia}^{\left(j\right)}\delta U_{\;i}^{\left(j\right)}\right)H_{\;1}^{\left(j\right)}H_{\;2}^{\left(j\right)}\right)A_{\;1}A_{\;2}d\alpha_{\;1}d\alpha_{\;2}$$

In the previous relation, $U_{\;i}^{\left(+\right)}=U_{\;i}^{\left(1\right)}$ and $U_{\;i}^{\left(-\right)}=U_{\;i}^{\left(2\right)}$ with i=1,2,3 are the virtual variations of the displacement field components at the top $\left(\zeta =h/2\right)$ and the bottom $\left(\zeta =-h/2\right)$ surfaces of the three-dimensional solid, respectively. The application of the static equivalence principle introduces a set of generalized loads for each τ=0,…,N+1 kinematic expansion order, which are collected in the vector $q^{\left(\tau \right)}\left(\alpha_{1},\alpha_{2}\right)={\left[q_{1}^{\left(\tau \right)}\;\;q_{2}^{\left(\tau \right)}\;\;q_{3}^{\left(\tau \right)}\right]}^{T}$. They are associated to the virtual variation of the vector $u^{\left(\tau \right)}\left(\alpha_{1},\alpha_{2}\right)$ of the generalized displacement field components. The following relation is thus obtained:(36)$$\delta L_{e}=\sum_{\tau =0}^{N+1}\int_{\alpha_{\;1}}\int_{\alpha_{\;2}}{\left(\delta u^{\left(\tau \right)}\right)}^{T}q^{\left(\tau \right)}A_{\;1}A_{\;2}d\alpha_{\;1}d\alpha_{\;2}=\sum_{\tau =0}^{N+1}\int_{\alpha_{\;1}}\int_{\alpha_{\;2}}\sum_{i=1}^{3}q_{\;i}^{\left(\tau \right)}\delta u_{\;i}^{\left(\tau \right)}A_{\;1}A_{\;2}d\alpha_{\;1}d\alpha_{\;2}$$

According to the static equivalence principle, the virtual work $\delta L_{\;e}^{\left(3D\right)}$ of Equation (35) turns out to be equal to the virtual work $\delta L_{e}$ of Equation (36):(37)$$\delta L_{\;e}^{\left(3D\right)}=\delta L_{e}$$

Substituting in Equation (35) the kinematic assumptions of Equation (8) and introducing them in Equation (37), the following expression is derived for the generalized loads $q_{\;i}^{\left(\tau \right)}=q_{\;1}^{\left(\tau \right)},q_{\;2}^{\left(\tau \right)},q_{\;3}^{\left(\tau \right)}$ [16]:(38)$$q_{\;i}^{\left(\tau \right)}=\sum_{j=1}^{2}q_{ia}^{\left(j\right)}F_{\tau }^{\alpha_{i}\left(j\right)}H_{1}^{\left(j\right)}H_{2}^{\left(j\right)}\;\;\;\;\;\mathrm{for}\;\;i=1,2,3$$

In the previous equation, the quantity $F_{\;\tau }^{\alpha_{i}\left(j\right)}$ with j=1,2 denotes the thickness function associated to $U_{\;i}^{\left(+\right)}$ and $U_{\;i}^{\left(-\right)}$, whereas $H_{\;1}^{\left(j\right)},H_{\;2}^{\left(j\right)}$ with j=1,2 are the scaling parameters calculated at the top $\left(\zeta =h/2\right)$ and the bottom $\left(\zeta =-h/2\right)$ surfaces of the shell.

Starting from Equation (35), it is possible to embed in the present formulation a general surface load. To this end, an arbitrary surface distribution $\bar{q}\left(\alpha_{1},\alpha_{2}\right)$ is considered, whereas the magnitude of the external load at issue is denoted by $q_{i}^{\left(j\right)}$
(39)$$q_{ia}^{\left(j\right)}\left(\alpha_{1},\alpha_{2},\pm \frac{h}{2}\right)={\left.q_{i}^{\left(j\right)}\right|}_{\zeta =\pm h/2}\bar{q}\left(\alpha_{1},\alpha_{2}\right)\;\;\;\;\;\;\;\mathrm{for}\;\;\begin{array}{l} i=1,2,3\\ j=1,2 \end{array}$$

In the case of a constant distribution of the external load, the distribution $\bar{q}\left(\alpha_{1},\alpha_{2}\right)=1$ is considered.

Introducing in Equation (33) the expression of the virtual work of external actions and remembering Equation (34), the higher-order equilibrium equations are derived on the shell reference surface. Employing a compact notation, it gives:(40)$$\sum_{i=1}^{3}D_{\Omega }^{*\alpha_{i}}S^{\left(\tau \right)\alpha_{i}}-q^{\left(\tau \right)}=0\;\;\;\;\;\mathrm{for}\;\tau =0,\ldots ,N+1$$

In the previous relation, $D_{\Omega }^{*\alpha_{i}}=D_{\Omega }^{*\alpha_{1}},D_{\Omega }^{*\alpha_{2}},D_{\Omega }^{*\alpha_{3}}$ denote the equilibrium operators, which can be expressed with a matrix notation as follows:(41)$$D_{\Omega }^{*\alpha_{1}}=\left[\begin{array}{c} {\bar{D}}_{\Omega }^{*\alpha_{\;1}}\\ 0\\ 0 \end{array}\right],\;\;\;\;\;\;D_{\Omega }^{*\alpha_{2}}=\left[\begin{array}{c} 0\\ {\bar{D}}_{\Omega }^{*\alpha_{\;2}}\\ 0 \end{array}\right],\;\;\;\;\;\;D_{\Omega }^{*\alpha_{3}}=\left[\begin{array}{c} 0\\ 0\\ {\bar{D}}_{\Omega }^{*\alpha_{\;3}} \end{array}\right]$$

An extended version of the quantities ${\bar{D}}_{\Omega }^{*\alpha_{i}}={\bar{D}}_{\Omega }^{*\alpha_{1}},{\bar{D}}_{\Omega }^{*\alpha_{2}},{\bar{D}}_{\Omega }^{*\alpha_{3}}$ is reported below:(42)$$\begin{array}{l} {\bar{D}}_{\Omega }^{*\alpha_{1}}=\left[\begin{array}{ccccccccc} {\left({\bar{D}}_{\;\Omega }^{*\alpha_{\;1}}\right)}_{\;1} & {\left({\bar{D}}_{\;\Omega }^{*\alpha_{\;1}}\right)}_{\;2} & {\left({\bar{D}}_{\;\Omega }^{*\alpha_{\;1}}\right)}_{\;3} & {\left({\bar{D}}_{\;\Omega }^{*\alpha_{\;1}}\right)}_{\;4} & {\left({\bar{D}}_{\;\Omega }^{*\alpha_{\;1}}\right)}_{\;5} & {\left({\bar{D}}_{\;\Omega }^{*\alpha_{\;1}}\right)}_{\;6} & {\left({\bar{D}}_{\;\Omega }^{*\alpha_{\;1}}\right)}_{\;7} & {\left({\bar{D}}_{\;\Omega }^{*\alpha_{\;1}}\right)}_{\;8} & {\left({\bar{D}}_{\;\Omega }^{*\alpha_{\;1}}\right)}_{\;9} \end{array}\right]\\ {\bar{D}}_{\Omega }^{*\alpha_{2}}=\left[\begin{array}{ccccccccc} {\left({\bar{D}}_{\;\Omega }^{*\alpha_{\;2}}\right)}_{\;1} & {\left({\bar{D}}_{\;\Omega }^{*\alpha_{\;2}}\right)}_{\;2} & {\left({\bar{D}}_{\;\Omega }^{*\alpha_{\;2}}\right)}_{\;3} & {\left({\bar{D}}_{\;\Omega }^{*\alpha_{\;2}}\right)}_{\;4} & {\left({\bar{D}}_{\;\Omega }^{*\alpha_{\;2}}\right)}_{\;5} & {\left({\bar{D}}_{\;\Omega }^{*\alpha_{\;2}}\right)}_{\;6} & {\left({\bar{D}}_{\;\Omega }^{*\alpha_{\;2}}\right)}_{\;7} & {\left({\bar{D}}_{\;\Omega }^{*\alpha_{\;2}}\right)}_{\;8} & {\left({\bar{D}}_{\;\Omega }^{*\alpha_{\;2}}\right)}_{\;9} \end{array}\right]\\ {\bar{D}}_{\Omega }^{*\alpha_{3}}=\left[\begin{array}{ccccccccc} {\left({\bar{D}}_{\;\Omega }^{*\alpha_{\;3}}\right)}_{\;1} & {\left({\bar{D}}_{\;\Omega }^{*\alpha_{\;3}}\right)}_{\;2} & {\left({\bar{D}}_{\;\Omega }^{*\alpha_{\;3}}\right)}_{\;3} & {\left({\bar{D}}_{\;\Omega }^{*\alpha_{\;3}}\right)}_{\;4} & {\left({\bar{D}}_{\;\Omega }^{*\alpha_{\;3}}\right)}_{\;5} & {\left({\bar{D}}_{\;\Omega }^{*\alpha_{\;3}}\right)}_{\;6} & {\left({\bar{D}}_{\;\Omega }^{*\alpha_{\;3}}\right)}_{\;7} & {\left({\bar{D}}_{\;\Omega }^{*\alpha_{\;3}}\right)}_{\;8} & {\left({\bar{D}}_{\;\Omega }^{*\alpha_{\;3}}\right)}_{\;9} \end{array}\right] \end{array}$$
where each term ${\bar{D}}_{\Omega }^{*\alpha_{i}}$ with i=1,2,3 reads as:(43)$$\begin{array}{l} {\left({\bar{D}}_{\Omega }^{*\alpha_{\;1}}\right)}_{1}={\left({\bar{D}}_{\Omega }^{*\alpha_{\;2}}\right)}_{3}={\left({\bar{D}}_{\Omega }^{*\alpha_{\;3}}\right)}_{5}=\frac{1}{A_{1}}\frac{\partial }{\partial \alpha_{1}}+\frac{1}{A_{1}A_{2}}\frac{\partial A_{2}}{\partial \alpha_{1}},\;\;{\left({\bar{D}}_{\Omega }^{*\alpha_{\;1}}\right)}_{4}={\left({\bar{D}}_{\Omega }^{*\alpha_{\;2}}\right)}_{2}={\left({\bar{D}}_{\Omega }^{*\alpha_{\;3}}\right)}_{6}=\frac{1}{A_{2}}\frac{\partial }{\partial \alpha_{2}}+\frac{1}{A_{1}A_{2}}\frac{\partial A_{1}}{\partial \alpha_{2}},\\ {\left({\bar{D}}_{\Omega }^{*\alpha_{\;1}}\right)}_{3}=-{\left({\bar{D}}_{\Omega }^{*\alpha_{\;2}}\right)}_{1}=\frac{1}{A_{1}A_{2}}\frac{\partial A_{1}}{\partial \alpha_{2}},\;\;{\left({\bar{D}}_{\Omega }^{*\alpha_{\;1}}\right)}_{2}=-{\left({\bar{D}}_{\Omega }^{*\alpha_{\;2}}\right)}_{4}=-\frac{1}{A_{1}A_{2}}\frac{\partial A_{2}}{\partial \alpha_{1}},\\ {\left({\bar{D}}_{\Omega }^{*\alpha_{\;1}}\right)}_{5}=-{\left({\bar{D}}_{\Omega }^{*\alpha_{\;3}}\right)}_{1}=\frac{1}{R_{1}},\;\;{\left({\bar{D}}_{\Omega }^{*\alpha_{\;2}}\right)}_{6}=-{\left({\bar{D}}_{\Omega }^{*\alpha_{\;3}}\right)}_{2}=\frac{1}{R_{2}},\;\;{\left({\bar{D}}_{\Omega }^{*\alpha_{\;1}}\right)}_{7}={\left({\bar{D}}_{\Omega }^{*\alpha_{\;2}}\right)}_{8}={\left({\bar{D}}_{\Omega }^{*\alpha_{\;3}}\right)}_{9}=-1,\\ {\left({\bar{D}}_{\Omega }^{*\alpha_{\;1}}\right)}_{6}={\left({\bar{D}}_{\Omega }^{*\alpha_{\;1}}\right)}_{8}={\left({\bar{D}}_{\Omega }^{*\alpha_{\;1}}\right)}_{9}={\left({\bar{D}}_{\Omega }^{*\alpha_{\;2}}\right)}_{5}={\left({\bar{D}}_{\Omega }^{*\alpha_{\;2}}\right)}_{7}={\left({\bar{D}}_{\Omega }^{*\alpha_{\;2}}\right)}_{9}={\left({\bar{D}}_{\Omega }^{*\alpha_{\;3}}\right)}_{3}={\left({\bar{D}}_{\Omega }^{*\alpha_{\;3}}\right)}_{4}={\left({\bar{D}}_{\Omega }^{*\alpha_{\;3}}\right)}_{7}={\left({\bar{D}}_{\Omega }^{*\alpha_{\;3}}\right)}_{8}=0 \end{array}$$

Introducing in Equation (40) the definition of $S^{\left(\tau \right)\alpha_{i}}$ in terms of $u^{\left(\tau \right)}$, as happens in Equation (31), the fundamental equations are derived in each point of the physical domain for the static analysis of doubly-curved shells with higher-order theories for each τ=0,…,N+1 [16]:(44)$$\sum_{\eta =0}^{N+1}L^{\left(\tau \eta \right)}u^{\left(\eta \right)}=q^{\left(\tau \right)}\;\;\;\;\;\;\;\;\;\mathrm{for}\;\;\;\tau =0,\ldots ,N+1$$

The fundamental matrix $L^{\left(\tau \eta \right)}$ referred to an arbitrary τ,η=0,…,N+1 occurring in Equation (44) turns out to be of size 3×3, reading as follows [16]:(45)$$L_{\;}^{\left(\tau \eta \right)}=\left[\begin{array}{ccc} L{\;}_{11}^{\left(\tau \eta \right)\alpha_{1}\alpha_{1}} & L{\;}_{12}^{\left(\tau \eta \right)\alpha_{1}\alpha_{2}} & L{\;}_{13}^{\left(\tau \eta \right)\alpha_{1}\alpha_{3}}\\ L{\;}_{21}^{\left(\tau \eta \right)\alpha_{2}\alpha_{1}} & L{\;}_{22}^{\left(\tau \eta \right)\alpha_{2}\alpha_{2}} & L{\;}_{23}^{\left(\tau \eta \right)\alpha_{2}\alpha_{3}}\\ L{\;}_{31}^{\left(\tau \eta \right)\alpha_{3}\alpha_{1}} & L{\;}_{32}^{\left(\tau \eta \right)\alpha_{3}\alpha_{2}} & L{\;}_{33}^{\left(\tau \eta \right)\alpha_{3}\alpha_{3}} \end{array}\right]=\sum_{i=1}^{3}\sum_{j=1}^{3}D_{\Omega }^{*\alpha_{i}}A^{\left(\tau \eta \right)\alpha_{i}\alpha_{j}}D_{\Omega }^{\alpha_{j}}\;\;\mathrm{for}\;\;\;\tau ,\eta =0,\ldots ,N+1$$

As shown in Equations (33) and (34), the application of the integration by parts rule with respect to the integration along $\alpha_{1},\alpha_{2}$ allows one to also derive the boundary conditions of the problem, which are applied at the edges of the rectangular physical domain $\left[\alpha_{1}^{0},\alpha_{1}^{1}\right]\times \left[\alpha_{2}^{0},\alpha_{2}^{1}\right]$ of extremes $\alpha_{\;i}^{0},\alpha_{\;i}^{1}$ with i=1,2. More specifically, the boundary conditions reported below are applied at the edges of the shell located at $\alpha_{1}=\alpha_{\;1}^{0}$ or $\alpha_{1}=\alpha_{\;1}^{1}$:(46)$$\begin{array}{l} N_{1}^{\left(\tau \right)\alpha_{1}}={\bar{N}}_{1}^{\left(\tau \right)\alpha_{1}}\;\;\;\;\;\mathrm{or}\;\;\;\;\;u_{1}^{\left(\tau \right)}={\bar{u}}_{1}^{\left(\tau \right)}\\ N_{12}^{\left(\tau \right)\alpha_{2}}={\bar{N}}_{12}^{\left(\tau \right)\alpha_{2}}\;\;\;\;\;\mathrm{or}\;\;\;\;\;u_{2}^{\left(\tau \right)}={\bar{u}}_{2}^{\left(\tau \right)}\\ T_{1}^{\left(\tau \right)\alpha_{3}}={\bar{T}}_{1}^{\left(\tau \right)\alpha_{3}}\;\;\;\;\;\;\;\mathrm{or}\;\;\;\;\;u_{3}^{\left(\tau \right)}={\bar{u}}_{3}^{\left(\tau \right)} \end{array}$$
where ${\bar{N}}_{\;1}^{\left(\tau \right)\alpha_{1}},{\bar{N}}_{\;12}^{\left(\tau \right)\alpha_{2}},{\bar{T}}_{\;1}^{\left(\tau \right)\alpha_{3}}$ and ${\bar{u}}_{\;1}^{\left(\tau \right)},{\bar{u}}_{\;2}^{\left(\tau \right)},{\bar{u}}_{\;3}^{\left(\tau \right)}$ are pre-determined values of the generalized stress resultants and the generalized displacement components, respectively, which can be assigned a-priori. In the same way, the following boundary conditions are derived for the physical domain edges with $\alpha_{2}=\alpha_{\;2}^{0}$ or $\alpha_{2}=\alpha_{\;2}^{1}$:(47)$$\begin{array}{l} N_{21}^{\left(\tau \right)\alpha_{1}}={\bar{N}}_{21}^{\left(\tau \right)\alpha_{1}}\;\;\;\;\;\mathrm{or}\;\;\;\;\;u_{1}^{\left(\tau \right)}={\bar{u}}_{1}^{\left(\tau \right)}\\ N_{2}^{\left(\tau \right)\alpha_{2}}={\bar{N}}_{2}^{\left(\tau \right)\alpha_{2}}\;\;\;\;\;\mathrm{or}\;\;\;\;\;u_{2}^{\left(\tau \right)}={\bar{u}}_{2}^{\left(\tau \right)}\\ T_{2}^{\left(\tau \right)\alpha_{3}}={\bar{T}}_{2}^{\left(\tau \right)\alpha_{3}}\;\;\;\;\;\;\;\mathrm{or}\;\;\;\;\;u_{3}^{\left(\tau \right)}={\bar{u}}_{3}^{\left(\tau \right)} \end{array}$$

In the previous relation, the fixed values of the generalized stress resultants and generalized displacement field components are denoted by ${\bar{N}}_{\;21}^{\left(\tau \right)\alpha_{1}},{\bar{N}}_{\;2}^{\left(\tau \right)\alpha_{2}},{\bar{T}}_{\;2}^{\left(\tau \right)\alpha_{3}}$ and ${\bar{u}}_{\;1}^{\left(\tau \right)},{\bar{u}}_{\;2}^{\left(\tau \right)},{\bar{u}}_{\;3}^{\left(\tau \right)}$, respectively.

Starting from Equations (46) and (47), the boundary conditions of physical interests are derived because they assign a null value to the kinematic and static quantities along the shell sides. In particular, the Simply-supported (S) boundary conditions are defined in order to neglect in-plane displacements of the lateral surfaces of the doubly-curved shell solid:(48)$$\begin{array}{l} N_{1}^{\left(\tau \right)\alpha_{1}}=0,\;\;\;\;\;\;u_{2}^{\left(\tau \right)}=u_{3}^{\left(\tau \right)}=0\;\;\;\;\;\;\mathrm{at}\;\;\;\;\;\;\alpha_{1}=\alpha_{1}^{0}\;\;\;\mathrm{or}\;\;\;\alpha_{1}=\alpha_{1}^{1}\\ N_{2}^{\left(\tau \right)\alpha_{2}}=0,\;\;\;\;\;\;u_{1}^{\left(\tau \right)}=u_{3}^{\left(\tau \right)}=0\;\;\;\;\;\;\mathrm{at}\;\;\;\;\;\;\alpha_{2}=\alpha_{2}^{0}\;\;\;\mathrm{or}\;\;\;\alpha_{2}=\alpha_{2}^{1} \end{array}$$

## 6. Semi-Analytical Navier Solution

In the present section, a semi-analytical solution is found for the higher-order differential problem of Equation (44). To this end, the Navier method is adopted; therefore, some geometric assumptions are made. More specifically, it is assumed that the geometry of the shell is characterized by constant values of the Lamè parameters $A_{\;1},A_{\;2}$ and of the principal radii of curvature $R_{\;1},R_{\;2}$. As a result, the relations reported in the following are considered:(49)$$\begin{array}{l} A_{1}=1\;\Rightarrow \;\frac{\partial^{n+m}A_{1}}{\partial s_{1}^{n}\partial s_{2}^{m}}=0,\;\;\;\;\;A_{2}=1\;\Rightarrow \;\frac{\partial^{n+m}A_{2}}{\partial s_{1}^{n}\partial s_{2}^{m}}=0,\\ R_{1}=\cos t\;\Rightarrow \;\frac{\partial^{n+m}R_{1}}{\partial s_{1}^{n}\partial s_{2}^{m}}=0,\;\;\;\;\;R_{2}=\cos t\;\Rightarrow \;\frac{\partial^{n+m}R_{2}}{\partial s_{1}^{n}\partial s_{2}^{m}}=0 \end{array}$$

As a consequence, the lengths $L_{\;1},L_{\;2}$ of the curvilinear parametric lines can be calculated in terms of the radii $R_{\;1},R_{\;2}$ as follows:(50)$$\begin{array}{l} L_{1}=s_{1}^{1}-s_{1}^{0}=\left(\alpha_{1}^{1}-\alpha_{1}^{0}\right)R_{1}\\ L_{2}=s_{2}^{1}-s_{2}^{0}=\left(\alpha_{2}^{1}-\alpha_{2}^{0}\right)R_{2} \end{array}$$

In the case of a cylindrical surface with $k_{\;n1}=0$ and $R_{\;2}=R$, the quantities $L_{\;1},L_{\;2}$ read as follows:(51)$$\begin{array}{l} L_{1}=s_{1}^{1}-s_{1}^{0}=\alpha_{1}^{1}-\alpha_{1}^{0}\\ L_{2}=s_{2}^{1}-s_{2}^{0}=\left(\alpha_{2}^{1}-\alpha_{2}^{0}\right)R \end{array}$$

When a rectangular plate is studied, the principal curvatures are null, namely $k_{\;n1}=k_{\;n2}=0$, therefore, the lengths of the parametric lines are calculated from the following expression:(52)$$\begin{array}{l} L_{1}=s_{1}^{1}-s_{1}^{0}=\alpha_{1}^{1}-\alpha_{1}^{0}\\ L_{2}=s_{2}^{1}-s_{2}^{0}=\alpha_{2}^{1}-\alpha_{2}^{0} \end{array}$$

The semi-analytical Navier solution accounts for the harmonic distribution of the unknown field variable within the physical domain $\left(s_{1},s_{2}\right)\in \left[0,L_{1}\right]\times \left[0,L_{2}\right]$. The displacement field components $u_{\;1}^{\left(\tau \right)},u_{\;2}^{\left(\tau \right)}$ and $u_{\;3}^{\left(\tau \right)}$ are thus expressed for each τ-th kinematic expansion order with τ=0,…,N+1 as follows:(53)$$\begin{array}{l} u_{1}^{\left(\tau \right)}\left(s_{1},s_{2}\right)=\sum_{n=1}^{\tilde{N}}\sum_{m=1}^{\tilde{M}}U_{1nm}^{\left(\tau \right)}\cos\left(\frac{n\pi }{L_{1}}s_{1}\right)\sin\left(\frac{m\pi }{L_{2}}s_{2}\right)\\ u_{2}^{\left(\tau \right)}\left(s_{1},s_{2}\right)=\sum_{n=1}^{\tilde{N}}\sum_{m=1}^{\tilde{M}}U_{2nm}^{\left(\tau \right)}\sin\left(\frac{n\pi }{L_{1}}s_{1}\right)\cos\left(\frac{m\pi }{L_{2}}s_{2}\right)\\ u_{3}^{\left(\tau \right)}\left(s_{1},s_{2}\right)=\sum_{n=1}^{\tilde{N}}\sum_{m=1}^{\tilde{M}}U_{3nm}^{\left(\tau \right)}\sin\left(\frac{n\pi }{L_{1}}s_{1}\right)\sin\left(\frac{m\pi }{L_{2}}s_{2}\right) \end{array}$$

In the previous equation, the quantities n and m denote the wave number of the solution along $s_{\;1}$ and $s_{\;2}$, respectively, whereas the quantities $U_{1nm}^{\left(\tau \right)},U_{2nm}^{\left(\tau \right)}$ and $U_{3nm}^{\left(\tau \right)}$ are the wave amplitudes associated to each wave number n,m. The total number of waves along $s_{\;1}$ and $s_{\;2}$ is denoted by $\tilde{N}$ and $\tilde{M}$, respectively. According to the Navier’s method, these quantities are assumed as $\tilde{N}=\tilde{M}=\infty$. It can be seen that the harmonic expansion of Equation (53) respects the following boundary conditions along the edges of the physical domain:(54)$$\begin{array}{l} u_{1}^{\left(\tau \right)}\left(s_{1}=0,s_{2}\right)\neq 0\;\;\;\;\;\;u_{1}^{\left(\tau \right)}\left(s_{1}=L_{1},s_{2}\right)\neq 0\;\;\;\;\;\;u_{1}^{\left(\tau \right)}\left(s_{1},s_{2}=0\right)=0\;\;\;\;\;\;u_{1}^{\left(\tau \right)}\left(s_{1},s_{2}=L_{2}\right)=0\\ u_{2}^{\left(\tau \right)}\left(s_{1}=0,s_{2}\right)=0\;\;\;\;\;\;u_{2}^{\left(\tau \right)}\left(s_{1}=L_{1},s_{2}\right)=0\;\;\;\;\;\;u_{2}^{\left(\tau \right)}\left(s_{1},s_{2}=0\right)\neq 0\;\;\;\;\;\;u_{2}^{\left(\tau \right)}\left(s_{1},s_{2}=L_{2}\right)\neq 0\\ u_{3}^{\left(\tau \right)}\left(s_{1}=0,s_{2}\right)=0\;\;\;\;\;\;u_{3}^{\left(\tau \right)}\left(s_{1}=L_{1},s_{2}\right)=0\;\;\;\;\;\;u_{3}^{\left(\tau \right)}\left(s_{1},s_{2}=0\right)=0\;\;\;\;\;\;u_{3}^{\left(\tau \right)}\left(s_{1},s_{2}=L_{2}\right)=0 \end{array}$$

As far as the external loads are concerned, the quantities $q_{1}^{\left(\pm \right)},q_{2}^{\left(\pm \right)}$ and $q_{3}^{\left(\pm \right)}$ of Equation (39), which are applied at the top and bottom surfaces of the shell, are also expanded in a harmonic form by means of the following expression:(55)$$\begin{array}{l} q_{1}^{\left(\pm \right)}\left(s_{1},s_{2}\right)=\sum_{n=1}^{\tilde{N}}\sum_{m=1}^{\tilde{M}}Q_{1snm}^{\left(\pm \right)}\cos\left(\frac{n\pi }{L_{1}}s_{1}\right)\sin\left(\frac{m\pi }{L_{2}}s_{2}\right)\\ q_{2}^{\left(\pm \right)}\left(s_{1},s_{2}\right)=\sum_{n=1}^{\tilde{N}}\sum_{m=1}^{\tilde{M}}Q_{2snm}^{\left(\pm \right)}\sin\left(\frac{n\pi }{L_{1}}s_{1}\right)\cos\left(\frac{m\pi }{L_{2}}s_{2}\right)\\ q_{3}^{\left(\pm \right)}\left(s_{1},s_{2}\right)=\sum_{n=1}^{\tilde{N}}\sum_{m=1}^{\tilde{M}}Q_{3snm}^{\left(\pm \right)}\sin\left(\frac{n\pi }{L_{1}}s_{1}\right)\sin\left(\frac{m\pi }{L_{2}}s_{2}\right) \end{array}$$
with $\tilde{N}=\tilde{M}=\infty$. The wave amplitudes of the external loads, defined for each wave number n,m, are denoted by $Q_{1snm}^{\left(\pm \right)},Q_{2snm}^{\left(\pm \right)}$ and $Q_{3snm}^{\left(\pm \right)}$. In Figure 1 and Figure 2 we report the expressions of the wave amplitudes of some load shapes of practical interest. 

The generalized external actions $q_{\;1}^{\left(\tau \right)},q_{\;2}^{\left(\tau \right)}$ and $q_{\;3}^{\left(\tau \right)}$ introduced in Equation (38) which are on the shell reference surface, are expanded for each τ=0,…,N+1 with trigonometric series:(56)$$\begin{array}{l} q_{1}^{\left(\tau \right)}\left(s_{1},s_{2}\right)=\sum_{n=1}^{\tilde{N}}\sum_{m=1}^{\tilde{M}}Q_{1snm}^{\left(\tau \right)}\cos\left(\frac{n\pi }{L_{1}}s_{1}\right)\sin\left(\frac{m\pi }{L_{2}}s_{2}\right)\\ q_{2}^{\left(\tau \right)}\left(s_{1},s_{2}\right)=\sum_{n=1}^{\tilde{N}}\sum_{m=1}^{\tilde{M}}Q_{2snm}^{\left(\tau \right)}\sin\left(\frac{n\pi }{L_{1}}s_{1}\right)\cos\left(\frac{m\pi }{L_{2}}s_{2}\right)\\ q_{3}^{\left(\tau \right)}\left(s_{1},s_{2}\right)=\sum_{n=1}^{\tilde{N}}\sum_{m=1}^{\tilde{M}}Q_{3snm}^{\left(\tau \right)}\sin\left(\frac{n\pi }{L_{1}}s_{1}\right)\sin\left(\frac{m\pi }{L_{2}}s_{2}\right) \end{array}$$
being $Q_{1snm}^{\left(\tau \right)},Q_{2snm}^{\left(\tau \right)}$ and $Q_{3snm}^{\left(\tau \right)}$ the amplitudes of the generalized external actions $q_{\;1}^{\left(\tau \right)},q_{\;2}^{\left(\tau \right)},q_{\;3}^{\left(\tau \right)}$ associated to the τ-th kinematic expansion order, with τ=0,…,N+1. Introducing Equations (55) and (56) in the static equivalence principle of Equation (37), the following definitions of the generalized amplitudes $Q_{1snm}^{\left(\tau \right)},Q_{2snm}^{\left(\tau \right)},Q_{3snm}^{\left(\tau \right)}$ of the actions $q_{\;1}^{\left(\tau \right)},q_{\;2}^{\left(\tau \right)},q_{\;3}^{\left(\tau \right)}$ are obtained in terms of the wave amplitudes $Q_{1snm}^{\left(\pm \right)},Q_{2snm}^{\left(\pm \right)},Q_{3snm}^{\left(\pm \right)}$ of the surface loads applied at the top and bottom surfaces of the shell:(57)$$\begin{array}{l} Q_{1snm}^{\left(\tau \right)}=Q_{1snm}^{\left(-\right)}F_{\tau }^{\left(1\right)\alpha_{1}\left(-\right)}H_{1}^{\left(-\right)}H_{2}^{\left(-\right)}+Q_{1snm}^{\left(+\right)}F_{\tau }^{\left(l\right)\alpha_{1}\left(+\right)}H_{1}^{\left(+\right)}H_{2}^{\left(+\right)}\\ Q_{2snm}^{\left(\tau \right)}=Q_{2snm}^{\left(-\right)}F_{\tau }^{\left(1\right)\alpha_{2}\left(-\right)}H_{1}^{\left(-\right)}H_{2}^{\left(-\right)}+Q_{2snm}^{\left(+\right)}F_{\tau }^{\left(l\right)\alpha_{2}\left(+\right)}H_{1}^{\left(+\right)}H_{2}^{\left(+\right)}\\ Q_{3snm}^{\left(\tau \right)}=Q_{3snm}^{\left(-\right)}F_{\tau }^{\left(1\right)\alpha_{3}\left(-\right)}H_{1}^{\left(-\right)}H_{2}^{\left(-\right)}+Q_{3snm}^{\left(+\right)}F_{\tau }^{\left(l\right)\alpha_{3}\left(+\right)}H_{1}^{\left(+\right)}H_{2}^{\left(+\right)} \end{array}$$

As it is well-known, the semi-analytical Navier solution can be found only for cross-ply lamination schemes. Therefore, the three-dimensional elastic constitutive relationship of Equation (17), written in the reference system of the problem for a generally anisotropic material, takes the following aspect:(58)$$\left[\begin{array}{c} \sigma_{\;1}^{\left(k\right)}\\ \sigma_{\;2}^{\left(k\right)}\\ \tau_{\;12}^{\left(k\right)}\\ \tau_{\;13}^{\left(k\right)}\\ \tau_{\;23}^{\left(k\right)}\\ \sigma_{\;3}^{\left(k\right)} \end{array}\right]=\left[\begin{array}{cccccc} {\bar{E}}_{\;11}^{\left(k\right)} & {\bar{E}}_{\;12}^{\left(k\right)} & 0 & 0 & 0 & {\bar{E}}_{\;13}^{\left(k\right)}\\ {\bar{E}}_{\;12}^{\left(k\right)} & {\bar{E}}_{\;22}^{\left(k\right)} & 0 & 0 & 0 & {\bar{E}}_{\;23}^{\left(k\right)}\\ 0 & 0 & {\bar{E}}_{\;66}^{\left(k\right)} & 0 & 0 & 0\\ 0 & 0 & 0 & {\bar{E}}_{\;44}^{\left(k\right)} & 0 & 0\\ 0 & 0 & 0 & 0 & {\bar{E}}_{\;55}^{\left(k\right)} & 0\\ {\bar{E}}_{\;13}^{\left(k\right)} & {\bar{E}}_{\;23}^{\left(k\right)} & 0 & 0 & 0 & {\bar{E}}_{\;33}^{\left(k\right)} \end{array}\right]\left[\begin{array}{c} \varepsilon_{\;1}^{\left(k\right)}\\ \varepsilon_{\;2}^{\left(k\right)}\\ \gamma_{\;12}^{\left(k\right)}\\ \gamma_{\;13}^{\left(k\right)}\\ \gamma_{\;23}^{\left(k\right)}\\ \varepsilon_{\;3}^{\left(k\right)} \end{array}\right]\;\;\;\;\;k=1,\ldots ,l$$

In addition, the material orientation angle $\vartheta^{\left(k\right)}$ occurring in Equation (19) is selected so that $\vartheta^{\left(k\right)}=\pm \pi /2$ or $\vartheta^{\left(k\right)}=0$. Introducing the harmonic expansion of the unknown field variables of Equation (53) and of the generalized external actions of Equation (56) in the fundamental relations of Equation (44), the vector $U_{nm}^{\left(\tau \right)}={\left[U_{1nm}^{\left(\tau \right)}\;\;\;U_{2nm}^{\left(\tau \right)}\;\;\;U_{3nm}^{\left(\tau \right)}\right]}^{T}$ of the wave amplitude of the displacement field components is derived for each τ=0,…,N+1 from the following expression:(59)$$\sum_{n=1}^{\tilde{N}}\sum_{m=1}^{\tilde{M}}\left(\sum_{\eta =0}^{N+1}{\tilde{L}}_{nm}^{\left(\tau \eta \right)}U_{nm}^{\left(\eta \right)}+Q_{snm}^{\left(\tau \right)}\right)=0$$
where the vector $Q_{snm}^{\left(\tau \right)}={\left[Q_{1snm}^{\left(\tau \right)}\;\;\;Q_{2snm}^{\left(\tau \right)}\;\;\;Q_{3snm}^{\left(\tau \right)}\right]}^{T}$ collects the amplitudes of the generalized external actions of the τ-th expansion order. Furthermore, the quantity ${\tilde{L}}_{nm}^{\left(\tau \eta \right)}$ is the fundamental matrix related to the wave numbers n,m, defined for each τ,η=0,…,N+1. In a more expanded form, the previous relation becomes:(60)$$\sum_{n=1}^{\tilde{N}}\sum_{m=1}^{\tilde{M}}\left(\sum_{\eta =0}^{N+1}\left[\begin{array}{ccccccc} {\tilde{L}}_{11nm}^{\left(\tau \eta \right)\alpha_{1}\alpha_{1}} & {\tilde{L}}_{12nm}^{\left(\tau \eta \right)\alpha_{1}\alpha_{2}} & {\tilde{L}}_{13nm}^{\left(\tau \eta \right)\alpha_{1}\alpha_{3}} & {\tilde{L}}_{14nm}^{\left(\tau \eta \right)\alpha_{1}\alpha_{4}} & {\tilde{L}}_{15nm}^{\left(\tau \eta \right)\alpha_{1}\alpha_{5}} & {\tilde{L}}_{16nm}^{\left(\tau \eta \right)\alpha_{1}\alpha_{6}} & {\tilde{L}}_{17nm}^{\left(\tau \eta \right)\alpha_{1}\alpha_{7}}\\ {\tilde{L}}_{21nm}^{\left(\tau \eta \right)\alpha_{2}\alpha_{1}} & {\tilde{L}}_{22nm}^{\left(\tau \eta \right)\alpha_{2}\alpha_{2}} & {\tilde{L}}_{23nm}^{\left(\tau \eta \right)\alpha_{2}\alpha_{3}} & {\tilde{L}}_{24nm}^{\left(\tau \eta \right)\alpha_{2}\alpha_{4}} & {\tilde{L}}_{25nm}^{\left(\tau \eta \right)\alpha_{2}\alpha_{5}} & {\tilde{L}}_{26nm}^{\left(\tau \eta \right)\alpha_{2}\alpha_{6}} & {\tilde{L}}_{27nm}^{\left(\tau \eta \right)\alpha_{2}\alpha_{7}}\\ {\tilde{L}}_{31nm}^{\left(\tau \eta \right)\alpha_{3}\alpha_{1}} & {\tilde{L}}_{32nm}^{\left(\tau \eta \right)\alpha_{3}\alpha_{2}} & {\tilde{L}}_{33nm}^{\left(\tau \eta \right)\alpha_{3}\alpha_{3}} & {\tilde{L}}_{34nm}^{\left(\tau \eta \right)\alpha_{3}\alpha_{4}} & {\tilde{L}}_{35nm}^{\left(\tau \eta \right)\alpha_{3}\alpha_{5}} & {\tilde{L}}_{36nm}^{\left(\tau \eta \right)\alpha_{3}\alpha_{6}} & {\tilde{L}}_{37nm}^{\left(\tau \eta \right)\alpha_{3}\alpha_{7}}\\ {\tilde{L}}_{41nm}^{\left(\tau \eta \right)\alpha_{4}\alpha_{1}} & {\tilde{L}}_{42nm}^{\left(\tau \eta \right)\alpha_{4}\alpha_{2}} & {\tilde{L}}_{43nm}^{\left(\tau \eta \right)\alpha_{4}\alpha_{3}} & {\tilde{L}}_{44nm}^{\left(\tau \eta \right)\alpha_{4}\alpha_{4}} & {\tilde{L}}_{45nm}^{\left(\tau \eta \right)\alpha_{4}\alpha_{5}} & {\tilde{L}}_{46nm}^{\left(\tau \eta \right)\alpha_{4}\alpha_{6}} & {\tilde{L}}_{47nm}^{\left(\tau \eta \right)\alpha_{4}\alpha_{7}}\\ {\tilde{L}}_{51nm}^{\left(\tau \eta \right)\alpha_{5}\alpha_{1}} & {\tilde{L}}_{52nm}^{\left(\tau \eta \right)\alpha_{5}\alpha_{2}} & {\tilde{L}}_{53nm}^{\left(\tau \eta \right)\alpha_{5}\alpha_{3}} & {\tilde{L}}_{54nm}^{\left(\tau \eta \right)\alpha_{5}\alpha_{4}} & {\tilde{L}}_{55nm}^{\left(\tau \eta \right)\alpha_{5}\alpha_{5}} & {\tilde{L}}_{56nm}^{\left(\tau \eta \right)\alpha_{5}\alpha_{6}} & {\tilde{L}}_{57nm}^{\left(\tau \eta \right)\alpha_{5}\alpha_{7}}\\ {\tilde{L}}_{61nm}^{\left(\tau \eta \right)\alpha_{6}\alpha_{1}} & {\tilde{L}}_{62nm}^{\left(\tau \eta \right)\alpha_{6}\alpha_{2}} & {\tilde{L}}_{63nm}^{\left(\tau \eta \right)\alpha_{6}\alpha_{3}} & {\tilde{L}}_{64nm}^{\left(\tau \eta \right)\alpha_{6}\alpha_{4}} & {\tilde{L}}_{65nm}^{\left(\tau \eta \right)\alpha_{6}\alpha_{5}} & {\tilde{L}}_{66nm}^{\left(\tau \eta \right)\alpha_{6}\alpha_{6}} & {\tilde{L}}_{67nm}^{\left(\tau \eta \right)\alpha_{6}\alpha_{7}}\\ {\tilde{L}}_{71nm}^{\left(\tau \eta \right)\alpha_{7}\alpha_{1}} & {\tilde{L}}_{72nm}^{\left(\tau \eta \right)\alpha_{7}\alpha_{2}} & {\tilde{L}}_{73nm}^{\left(\tau \eta \right)\alpha_{7}\alpha_{3}} & {\tilde{L}}_{74nm}^{\left(\tau \eta \right)\alpha_{7}\alpha_{4}} & {\tilde{L}}_{75nm}^{\left(\tau \eta \right)\alpha_{7}\alpha_{5}} & {\tilde{L}}_{76nm}^{\left(\tau \eta \right)\alpha_{7}\alpha_{6}} & {\tilde{L}}_{77nm}^{\left(\tau \eta \right)\alpha_{7}\alpha_{7}} \end{array}\right]\left[\begin{array}{c} U_{1nm}^{\left(\eta \right)}\\ U_{2nm}^{\left(\eta \right)}\\ U_{3nm}^{\left(\eta \right)}\\ \Phi_{nm}^{\left(\eta \right)}\\ \Psi_{nm}^{\left(\eta \right)}\\ \Xi_{nm}^{\left(\eta \right)}\\ {Κ}_{nm}^{\left(\eta \right)} \end{array}\right]+\left[\begin{array}{c} Q_{1snm}^{\left(\tau \right)}\\ Q_{2snm}^{\left(\tau \right)}\\ Q_{3snm}^{\left(\tau \right)}\\ Q_{Dsnm}^{\left(\tau \right)}\\ Q_{Bsnm}^{\left(\tau \right)}\\ Q_{Tsnm}^{\left(\tau \right)}\\ Q_{Csnm}^{\left(\tau \right)} \end{array}\right]\right)=\left[\begin{array}{c} 0\\ 0\\ 0\\ 0\\ 0\\ 0\\ 0 \end{array}\right]$$

The complete expression of the coefficients ${\tilde{L}}_{ijnm}^{\left(\tau \eta \right)\alpha_{i}\alpha_{j}}$ of the fundamental matrix ${\tilde{L}}_{nm}^{\left(\tau \eta \right)},$ with i,j=1,2,3, is reported below for each τ,η=0,…,N+1:(61)$$\begin{array}{l} {\tilde{L}}_{11nm}^{\left(\tau \eta \right)\alpha_{1}\alpha_{1}}=-A_{11\left(20\right)}^{\left(\tau \eta \right)\left[00\right]\alpha_{1}\alpha_{1}}{\left(\frac{n\pi }{L_{1}}\right)}^{2}-A_{66\left(02\right)}^{\left(\tau \eta \right)\left[00\right]\alpha_{1}\alpha_{1}}{\left(\frac{m\pi }{L_{2}}\right)}^{2}-\frac{A_{44\left(20\right)}^{\left(\tau \eta \right)\left[00\right]\alpha_{1}\alpha_{1}}}{R_{\;1}^{2}}+\frac{A_{44\left(10\right)}^{\left(\tau \eta \right)\left[01\right]\alpha_{1}\alpha_{1}}+A_{44\left(10\right)}^{\left(\tau \eta \right)\left[10\right]\alpha_{1}\alpha_{1}}}{R_{\;1}}-A_{44\left(00\right)}^{\left(\tau \eta \right)\left[11\right]\alpha_{1}\alpha_{1}}\\ {\tilde{L}}_{12nm}^{\left(\tau \eta \right)\alpha_{1}\alpha_{2}}=-\left(A_{12\left(11\right)}^{\left(\tau \eta \right)\left[00\right]\alpha_{1}\alpha_{2}}+A_{66\left(11\right)}^{\left(\tau \eta \right)\left[00\right]\alpha_{1}\alpha_{2}}\right)\left(\frac{n\pi }{L_{1}}\right)\left(\frac{m\pi }{L_{2}}\right)\\ {\tilde{L}}_{13nm}^{\left(\tau \eta \right)\alpha_{1}\alpha_{3}}=\left(A_{13\left(10\right)}^{\left(\tau \eta \right)\left[01\right]\alpha_{1}\alpha_{3}}-A_{44\left(10\right)}^{\left(\tau \eta \right)\left[10\right]\alpha_{1}\alpha_{3}}+\frac{A_{11\left(20\right)}^{\left(\tau \eta \right)\left[00\right]\alpha_{1}\alpha_{3}}+A_{44\left(20\right)}^{\left(\tau \eta \right)\left[00\right]\alpha_{1}\alpha_{3}}}{R_{\;1}}+\frac{A_{12\left(11\right)}^{\left(\tau \eta \right)\left[00\right]\alpha_{1}\alpha_{3}}}{R_{\;2}}\right)\left(\frac{n\pi }{L_{1}}\right)\\ {\tilde{L}}_{21nm}^{\left(\tau \eta \right)\alpha_{2}\alpha_{1}}=-\left(A_{12\left(11\right)}^{\left(\tau \eta \right)\left[00\right]\alpha_{2}\alpha_{1}}+A_{66\left(11\right)}^{\left(\tau \eta \right)\left[00\right]\alpha_{2}\alpha_{1}}\right)\left(\frac{n\pi }{L_{1}}\right)\left(\frac{m\pi }{L_{2}}\right)\\ {\tilde{L}}_{22nm}^{\left(\tau \eta \right)\alpha_{2}\alpha_{2}}=-A_{66\left(20\right)}^{\left(\tau \eta \right)\left[00\right]\alpha_{2}\alpha_{2}}{\left(\frac{n\pi }{L_{1}}\right)}^{2}-A_{22\left(02\right)}^{\left(\tau \eta \right)\left[00\right]\alpha_{2}\alpha_{2}}{\left(\frac{m\pi }{L_{2}}\right)}^{2}-\frac{A_{55\left(02\right)}^{\left(\tau \eta \right)\left[00\right]\alpha_{2}\alpha_{2}}}{R_{\;2}^{2}}+\frac{A_{55\left(01\right)}^{\left(\tau \eta \right)\left[01\right]\alpha_{2}\alpha_{2}}+A_{55\left(01\right)}^{\left(\tau \eta \right)\left[10\right]\alpha_{2}\alpha_{2}}}{R_{\;2}}-A_{55\left(00\right)}^{\left(\tau \eta \right)\left[11\right]\alpha_{2}\alpha_{2}}\\ {\tilde{L}}_{31nm}^{\left(\tau \eta \right)\alpha_{3}\alpha_{1}}=\left(A_{13\left(10\right)}^{\left(\tau \eta \right)\left[10\right]\alpha_{3}\alpha_{1}}-A_{44\left(10\right)}^{\left(\tau \eta \right)\left[01\right]\alpha_{3}\alpha_{1}}+\frac{A_{11\left(20\right)}^{\left(\tau \eta \right)\left[00\right]\alpha_{3}\alpha_{1}}+A_{44\left(20\right)}^{\left(\tau \eta \right)\left[00\right]\alpha_{3}\alpha_{1}}}{R_{\;1}}+\frac{A_{12\left(11\right)}^{\left(\tau \eta \right)\left[00\right]\alpha_{3}\alpha_{1}}}{R_{\;2}}\right)\left(\frac{n\pi }{L_{1}}\right)\\ {\tilde{L}}_{32nm}^{\left(\tau \eta \right)\alpha_{3}\alpha_{2}}=\left(A_{23\left(01\right)}^{\left(\tau \eta \right)\left[10\right]\alpha_{3}\alpha_{2}}-A_{55\left(01\right)}^{\left(\tau \eta \right)\left[01\right]\alpha_{3}\alpha_{2}}+\frac{A_{12\left(11\right)}^{\left(\tau \eta \right)\left[00\right]\alpha_{3}\alpha_{2}}}{R_{\;1}}+\frac{A_{22\left(02\right)}^{\left(\tau \eta \right)\left[00\right]\alpha_{3}\alpha_{2}}+A_{55\left(02\right)}^{\left(\tau \eta \right)\left[00\right]\alpha_{3}\alpha_{2}}}{R_{\;2}}\right)\left(\frac{m\pi }{L_{2}}\right)\\ {\tilde{L}}_{33nm}^{\left(\tau \eta \right)\alpha_{3}\alpha_{3}}=-A_{44\left(20\right)}^{\left(\tau \eta \right)\left[00\right]\alpha_{3}\alpha_{3}}{\left(\frac{n\pi }{L_{1}}\right)}^{2}-A_{55\left(02\right)}^{\left(\tau \eta \right)\left[00\right]\alpha_{3}\alpha_{3}}{\left(\frac{m\pi }{L_{2}}\right)}^{2}-\frac{A_{11\left(20\right)}^{\left(\tau \eta \right)\left[00\right]\alpha_{3}\alpha_{3}}}{R_{\;1}^{2}}-\frac{A_{22\left(02\right)}^{\left(\tau \eta \right)\left[00\right]\alpha_{3}\alpha_{3}}}{R_{\;2}^{2}}-\frac{2A_{12\left(11\right)}^{\left(\tau \eta \right)\left[00\right]\alpha_{3}\alpha_{3}}}{R_{\;1}R_{\;2}}\\ \;\;\;\;\;\;\;\;\;\;\;-\frac{A_{13\left(10\right)}^{\left(\tau \eta \right)\left[01\right]\alpha_{3}\alpha_{3}}+A_{13\left(10\right)}^{\left(\tau \eta \right)\left[10\right]\alpha_{3}\alpha_{3}}}{R_{\;1}}-\frac{A_{23\left(01\right)}^{\left(\tau \eta \right)\left[01\right]\alpha_{3}\alpha_{3}}+A_{23\left(01\right)}^{\left(\tau \eta \right)\left[10\right]\alpha_{3}\alpha_{3}}}{R_{\;2}}-A_{33\left(00\right)}^{\left(\tau \eta \right)\left[11\right]\alpha_{3}\alpha_{3}} \end{array}$$

It should be noted that in the case of a cylindrical surface, the fundamental governing equations are modified, remembering the geometric relations of Equation (51). More specifically, when $R_{1}=\infty$, the generalized coefficients $A_{\;nm\;\;\left(pq\right)}^{\left(\tau \eta \right)\left[fg\right]\alpha_{i}\alpha_{j}}$ of Equation (29) are calculated for each τ,η=0,…,N+1 from the following relation:(62)$$A_{\;nm\;\;\left(pq\right)}^{\left(\tau \eta \right)\left[fg\right]\alpha_{i}\alpha_{j}}=\sum_{k=1}^{l}\int_{\zeta_{k}}^{\zeta_{k+1}}{\bar{B}}_{nm}^{\left(k\right)}\frac{\partial^{f}F^{\left(k\eta \right)\alpha_{\;j}}}{\partial \zeta^{f}}\frac{\partial^{g}F^{\left(k\tau \right)\alpha_{\;i}}}{\partial \zeta^{g}}\frac{H_{\;2}}{H_{\;2}^{q}}d\zeta$$

In this case, the fundamental coefficients ${\tilde{L}}_{ijnm}^{\left(\tau \eta \right)\alpha_{i}\alpha_{j}}$, which have been defined in Equation (61), are simplified because one principal direction is characterized by a null value of the principal curvature. The interested reader can find in Appendix A the extended expression of ${\tilde{L}}_{ijnm}^{\left(\tau \eta \right)\alpha_{i}\alpha_{j}}$ for the case of a cylindrical surface.

Finally, in the case of a rectangular plate with $R_{1}=R_{2}=\infty$, the generalized coefficients $A_{\;nm\;\;\left(pq\right)}^{\left(\tau \eta \right)\left[fg\right]\alpha_{i}\alpha_{j}}$ are derived as follows:(63)$$A_{\;nm\;\;\left(pq\right)}^{\left(\tau \eta \right)\left[fg\right]\alpha_{i}\alpha_{j}}=\sum_{k=1}^{l}\int_{\zeta_{k}}^{\zeta_{k+1}}{\bar{B}}_{nm}^{\left(k\right)}\frac{\partial^{f}F^{\left(k\eta \right)\alpha_{\;j}}}{\partial \zeta^{f}}\frac{\partial^{g}F^{\left(k\tau \right)\alpha_{\;i}}}{\partial \zeta^{g}}d\zeta$$

The expression of the coefficients ${\tilde{L}}_{ijnm}^{\left(\tau \eta \right)\alpha_{i}\alpha_{j}}$ of the fundamental matrix ${\tilde{L}}_{nm}^{\left(\tau \eta \right)}$ can be found for the case of a rectangular plate in Appendix B.

## 7. Generalized Differential Quadrature Method

In the present section, the main features of the GDQ method are presented for the derivation of the numerical solution of the three-dimensional equilibrium equations along the thickness direction, as shown previously.

A computational grid made of a discrete number of points is defined in the thickness direction, following a symmetric non-uniform distribution. In this context, the Chebyshev–Gauss–Lobatto (CGL) distribution [16] is adopted. Referring to the interval $\left[-1,1\right]$, the location of the generic element ${\bar{x}}_{\;i}$ of the grid at issue is derived as follows:(64)$${\bar{x}}_{\;i}=-\cos\left(\frac{i-1}{I_{Q}-1}\pi \right),\;\;\;\;\;i=1,\ldots ,I_{\;Q},\;\;\;\;\;\mathrm{for}\;\;\;x_{\;i}\in \left[-1,1\right]$$
where $I_{\;Q}$ is the total number of discrete points. As stated previously, the GDQ method allows one to evaluate the derivative of a generic n-th order of a smooth function from a weighted sum of the values assumed by the function itself in its definition domain. Referring to an arbitrary univariate function $f=f\left(x\right)$ defined in the closed interval $\left[a,b\right]$ with a,b∈ℝ, its n-th order derivative in a point $x_{\;i}\in \left[a,b\right]$ with $i=1,\ldots ,I_{\;Q}$ is thus calculated with the expression reported in the following [16]:(65)$${\left.f^{\left(n\right)}\left(x_{\;i}\right)=\frac{\partial^{n}f\left(x\right)}{\partial x^{n}}\right|}_{x=x_{i}}≅\sum_{j=1}^{I_{\;Q}}\varsigma_{ij}^{\left(n\right)}f\left(x_{j}\right)\;\;\;i=1,\;2,\ldots ,\;I_{Q}$$

In the previous relation, the quantity $f\left(x_{j}\right)$ with $j=1,\ldots ,I_{\;Q}$ denotes the values assumed by the function in the discrete grid, whereas $\varsigma_{ij}^{\left(n\right)}$ are the weighting coefficients of the numerical method. As shown in other works, the present numerical approach provides a high level of accuracy for a sufficient number of discrete points, namely $I_{\;Q}>n$. The GDQ weighting coefficients $\varsigma_{ij}^{\left(n\right)}$ are calculated with a recursive procedure [16] based on the adoption of the Lagrange polynomials, defined on the computational discrete grid, for the interpolation of the solution:(66)$$\begin{array}{cc} \varsigma_{ij}^{\left(1\right)}=\frac{L^{\;\left(1\right)}\left(x_{\;i}\right)}{\left(x_{\;i}-x_{\;j}\right)L^{\;\left(1\right)}\left(x_{\;j}\right)},\;\;\;\;\;\;\varsigma_{ij}^{\left(n\right)}=n\left(\varsigma_{ij}^{\left(1\right)}\varsigma_{ii}^{\left(n-1\right)}-\frac{\varsigma_{ij}^{\left(n-1\right)}}{x_{\;i}-x_{\;j}}\right) & i\neq j\\ \varsigma_{ii}^{\left(n\right)}=-\sum_{j=1\;j\neq i}^{I_{\;N}}\varsigma_{ij}^{\left(n\right)} & i=j \end{array}$$

In the previous relation, the quantities $L^{\;\left(1\right)}\left(x_{i}\right)$ and $L^{\;\left(1\right)}\left(x_{j}\right)$ denote the first order derivatives of the Lagrange polynomials at the points $x_{\;i}$ and $x_{\;j}$, respectively. On the other hand, the definition $\varsigma_{\;ij}^{\left(0\right)}=\delta_{\;ij}$ with $i,j=1,\ldots ,I_{\;Q}$ should be introduced, being $\delta_{ij}$ the Kronecker delta operator.

The GDQ method can also be applied for the numerical computation of integrals within the GIQ numerical method. According to the GIQ, the integration, restricted to the closed interval $\left[x_{i},x_{j}\right]$ with $x_{\;i},x_{\;j}\in \left[a,b\right]$, of a smooth function $f=f\left(x\right)$ with $x\in \left[a,b\right]$ can be evaluated as follows [16]:(67)$$\int_{x_{i}}^{x_{j}}f\left(x\right)dx=\sum_{k=1}^{I_{\;Q}}{\tilde{w}}_{k}^{ij}f\left(x_{k}\right)=\sum_{k=1}^{I_{\;Q}}\left(w_{jk}-w_{ik}\right)f\left(x_{k}\right)$$

As can be seen, the definition interval of f is discretized with a grid of $I_{\;Q}$ points. The GIQ coefficients $w_{ik},w_{jk}$ with $i,j,k=1,\ldots ,I_{\;Q}$ are collected in the matrix W of size $I_{\;Q}\times I_{\;Q}$. The coefficients at issue are derived from the GDQ shifted coefficients of the first order derivative, denoted by ${\bar{\varsigma }}_{\;ij}^{\left(1\right)}$ with $i,j=1,\ldots I_{\;Q}$, which are defined as follows, setting $\varepsilon =1\times 10^{-10}$:(68)$${\bar{\varsigma }}_{\;ij}^{\left(1\right)}=\left\{\begin{array}{l} \frac{r_{\;i}-\varepsilon }{r_{\;j}-\varepsilon }{\tilde{\varsigma }}_{ij}^{\left(1\right)}\;\;\;\;\;\;\;\;\;i\neq j\\ \frac{1}{r_{\;i}-\varepsilon }+{\tilde{\varsigma }}_{ij}^{\left(1\right)}\;\;\;\;\;\;i=j \end{array}\right.$$

The shifted coefficients of Equation (68) are collected in the matrix ${\bar{\varsigma }}^{\left(1\right)}$, whose size is $I_{\;Q}\times I_{\;Q}$. It can be shown that the matrix of the GIQ coefficients is the inverse of the matrix ${\bar{\varsigma }}^{\left(1\right)}$:(69)$$W={\left({\bar{\varsigma }}^{\left(1\right)}\right)}^{-1}$$

When the numerical integration is restricted to an arbitrary interval $\left[a,b\right]$, the domain $\left[-1,1\right]$ becomes a parent interval. For this reason, the coefficients ${\tilde{w}}_{\;k}^{1I_{Q}}$ of Equation (67) are transformed into those $w_{\;k}^{1I_{Q}}$ by means of the following GIQ mapping technique:(70)$$w_{\;k}^{1I_{Q}}=\frac{b-a}{2}{\tilde{w}}_{\;k}^{1I_{Q}}$$

In this way, the integral of $f=f\left(x\right)$ restricted to the interval $\left[a,b\right]$ are evaluated as follows [16]:(71)$$\int_{a}^{b}f(x)dx=\sum_{k=1}^{I_{\;Q}}w_{k}^{1I_{Q}}f(x_{k})$$

## 8. Stress and Strain Recovery Procedure

In the previous section, the two-dimensional Navier closed-form solution of the fundamental relations reported in Equation (44) was derived. Therefore, the actual response of the three-dimensional doubly-curved solid is now derived. The reconstruction of stress and strain profiles requires the adoption of three-dimensional equilibrium equations because only the adoption of the ESL kinematic and constitutive relations may lead to erroneous results. Referring to a doubly-curved shell solid with constant principal radii of curvature $R_{\;1},R_{\;2}$ along the physical domain and $A_{\;1},A_{\;2}=1$, the three-dimensional equilibrium equations assume the following aspect [16]:(72)$$\begin{array}{l} \frac{\partial \tau_{13}^{\left(k\right)}}{\partial \zeta }+\left(\frac{2}{R_{\;1}+\zeta }+\frac{1}{R_{\;2}+\zeta }\right)\tau_{13}^{\left(k\right)}=a^{\left(k\right)}\\ \frac{\partial \tau_{23}^{\left(k\right)}}{\partial \zeta }+\left(\frac{1}{R_{\;1}+\zeta }+\frac{2}{R_{\;2}+\zeta }\right)\tau_{23}^{\left(k\right)}=b^{\left(k\right)}\\ \frac{\partial \sigma_{3}^{\left(k\right)}}{\partial \zeta }+\left(\frac{1}{R_{\;1}+\zeta }+\frac{1}{R_{\;2}+\zeta }\right)\sigma_{3}^{\left(k\right)}=c^{\left(k\right)} \end{array}$$
where the parameters $a^{\left(k\right)},b^{\left(k\right)}$ and $c^{\left(k\right)}$ are written in an extended form as follows:(73)$$\begin{array}{l} a^{\left(k\right)}=-\frac{1}{\left(1+\zeta /R_{\;1}\right)}\frac{\partial \sigma_{1}^{\left(k\right)}}{\partial s_{1}}-\frac{1}{\left(1+\zeta /R_{\;2}\right)}\frac{\partial \tau_{12}^{\left(k\right)}}{\partial s_{2}}\\ b^{\left(k\right)}=-\frac{1}{\left(1+\zeta /R_{\;1}\right)}\frac{\partial \tau_{12}^{\left(k\right)}}{\partial s_{1}}-\frac{1}{\left(1+\zeta /R_{\;2}\right)}\frac{\partial \sigma_{2}^{\left(k\right)}}{\partial s_{2}}\\ c^{\left(k\right)}=\frac{\sigma_{1}^{\left(k\right)}}{R_{1}+\zeta }+\frac{\sigma_{2}^{\left(k\right)}}{R_{2}+\zeta }-\frac{1}{\left(1+\zeta /R_{\;1}\right)}\frac{\partial \tau_{13}^{\left(k\right)}}{\partial s_{1}}-\frac{1}{\left(1+\zeta /R_{\;2}\right)}\frac{\partial \tau_{23}^{\left(k\right)}}{\partial s_{2}} \end{array}$$

At this point, a two-dimensional grid is extracted from the physical domain $\left[s_{1}^{0},s_{1}^{1}\right]\times \left[s_{2}^{0},s_{2}^{1}\right]$ made of $I_{\;N}\times I_{\;M}$ nodes, starting from the non-uniform one-dimensional CGL distribution of Equation (64).

As far as the thickness direction is concerned, a discrete grid of $I_{\;T}$ points is defined in each interval $\left[\zeta_{k},\zeta_{k+1}\right]$ of length $h_{\;k}=\zeta_{k+1}-\zeta_{k}$ related to an arbitrary k-th layer of the stacking sequence, setting k=1,…,l. The generic element of this grid is denoted by $\zeta_{\;\tilde{m}}^{\left(k\right)}$ for $\tilde{m}=1,\ldots ,I_{\;T}$, with $\zeta_{\;\tilde{m}}^{\left(k\right)}\in \left[\zeta_{k},\zeta_{k+1}\right]$. The quantity $\zeta_{\;\tilde{m}}^{\left(k\right)}$ is defined from an arbitrary distribution within the dimensionless interval $\xi_{\;\tilde{m}}\in \left[-1,1\right]$ as follows:(74)$$\zeta_{\;\tilde{m}}^{\left(k\right)}=\frac{h_{k}}{2}\xi_{\;\tilde{m}}+\frac{\zeta_{k+1}+\zeta_{k}}{2}$$

Finally, the elements $\zeta_{\;\tilde{m}}^{\left(k\right)}$ are arranged in the vector $\zeta^{\left(k\right)}={\left[\zeta_{1}^{\left(k\right)}\;\;\;\cdots \;\;\;\zeta_{\tilde{m}}^{\left(k\right)}\;\;\;\cdots \;\;\;\zeta_{I_{T}}^{\left(k\right)}\right]}^{T}$ of size $I_{\;T}\times 1$, defined in each k-th lamina. At this point, a new vector of size $l\;I_{T}\times 1$ is introduced, which contains all the discrete points $\zeta_{m}$ belonging to the interval $\left[-h/2,h/2\right]$ that are located in the thickness direction. To this end, the index $m=1,\ldots ,l\;I_{T}$ is introduced, defined as $m=\left(k-1\right)I_{T}+\tilde{m}$. As a consequence, the vector $\zeta^{\left(k\right)}$ is arranged in the global vector ${\left[\zeta_{1}\;\;\;\cdots \;\;\;\zeta_{m}\;\;\;\cdots \;\;\;\zeta_{l\;I_{T}}\right]}^{T}=\left[\zeta^{\left(1\right)T}\;\;\;\cdots \;\;\;\zeta^{\left(k\right)T}\;\;\;\cdots \;\;\;\zeta^{\left(l\right)T}\right]$ of size $l\;I_{\;T}\times 1$.

The through-the-thickness displacement field profile can be evaluated for each point $\left(s_{1i},s_{2j}\right)$ of the reference surface, remembering the kinematic assumption of Equation (8). The relation reported in the following is thus considered for each $m=1,\ldots ,l\;I_{T}$:(75)$$U_{\left(ijm\right)}^{\left(k\right)}=\sum_{\tau =0}^{N+1}F_{\tau \left(ijm\right)}^{\left(k\right)}u^{\left(\tau \right)}\left(s_{1i},s_{2j}\right)$$

In the same way, from Equation (14) the profiles are derived of the three-dimensional strain components of the vector $\varepsilon_{\left(ijm\right)}^{\left(k\right)}={\left[\varepsilon_{1\left(ijm\right)}^{\left(k\right)}\;\;\varepsilon_{2\left(ijm\right)}^{\left(k\right)}\;\;\gamma_{12\left(ijm\right)}^{\left(k\right)}\;\;\gamma_{13\left(ijm\right)}^{\left(k\right)}\;\;\gamma_{23\left(ijm\right)}^{\left(k\right)}\;\;\varepsilon_{3\left(ijm\right)}^{\left(k\right)}\right]}^{T}$ for each point $\left(s_{1i},s_{2j}\right)$ of the reference surface of the shell:(76)$$\varepsilon_{\left(ijm\right)}^{\left(k\right)}=\sum_{\tau =\;0}^{N+1}\sum_{i=1}^{3}Z{\;}_{\left(ijm\right)}^{\left(k\tau \right)\alpha_{\;i}}\varepsilon_{\left(ij\right)}^{\left(\tau \right)\alpha_{\;i}}$$

The quantity $\varepsilon_{\left(ij\right)}^{\left(\tau \right)\alpha_{i}}=\varepsilon^{\left(\tau \right)\alpha_{i}}\left(s_{1i},s_{2j}\right)$ with i=1,2,3 denotes the generalized strain vector, defined for each τ-th kinematic expansion order, evaluated in each point of the reference surface. Starting from the three-dimensional constitutive relation of Equation (17) with ${\bar{E}}_{\;ij}^{\left(k\right)}={\bar{C}}_{\;ij}^{\left(k\right)}$ for i,j=1,…,6, the distribution of the membrane stresses $\sigma_{\;1}^{\left(k\right)},\sigma_{\;2}^{\left(k\right)}$ and $\tau_{\;12}^{\left(k\right)}$ is derived introducing in each through-the-thickness discrete point of the computational grid the three-dimensional strain components of Equation (76), remembering the hypotheses made for the derivation of the Navier closed-form solution:(77)$$\left[\begin{array}{c} \sigma_{1\left(ijm\right)}^{\left(k\right)}\\ \sigma_{2\left(ijm\right)}^{\left(k\right)}\\ \tau_{12\left(ijm\right)}^{\left(k\right)} \end{array}\right]=\left[\begin{array}{cccccc} {\bar{C}}_{11\left(m\right)}^{\left(k\right)} & {\bar{C}}_{12\left(m\right)}^{\left(k\right)} & 0 & 0 & 0 & {\bar{C}}_{13\left(m\right)}^{\left(k\right)}\\ {\bar{C}}_{12\left(m\right)}^{\left(k\right)} & {\bar{C}}_{22\left(m\right)}^{\left(k\right)} & 0 & 0 & 0 & {\bar{C}}_{23\left(m\right)}^{\left(k\right)}\\ 0 & 0 & {\bar{C}}_{66\left(m\right)}^{\left(k\right)} & 0 & 0 & 0 \end{array}\right]\left[\begin{array}{c} \varepsilon_{\;1\left(ijm\right)}^{\left(k\right)}\\ \varepsilon_{\;2\left(ijm\right)}^{\left(k\right)}\\ \gamma_{\;12\left(ijm\right)}^{\left(k\right)}\\ \gamma_{\;13\left(ijm\right)}^{\left(k\right)}\\ \gamma_{\;23\left(ijm\right)}^{\left(k\right)}\\ \varepsilon_{\;3\left(ijm\right)}^{\left(k\right)} \end{array}\right]$$

Once the discrete distribution of membrane stresses is derived along the shell thickness according to Equation (77), it is useful to compute their first order derivative with respect to $s_{1},s_{2}$ parametric lines. These derivations are performed numerically by means of the GDQ method of Equation (65). One gets:(78)$$\begin{array}{l} {\left.\sigma_{1,1}^{\left(k\right)}=\frac{\partial \sigma_{1}^{\left(k\right)}}{\partial s_{1}}\right|}_{\left(ijm\right)}≅\sum_{r=1}^{I_{N}}\varsigma_{ir}^{s_{1}\left(1\right)}\sigma_{1\left(rjm\right)}^{\left(k\right)},\;\;\sigma_{2,2}^{\left(k\right)}={\left.\frac{\partial \sigma_{2}^{\left(k\right)}}{\partial s_{2}}\right|}_{\left(ijm\right)}≅\sum_{r=1}^{I_{M}}\varsigma_{jr}^{s_{2}\left(1\right)}\sigma_{2\left(irm\right)}^{\left(k\right)},\\ {\left.\tau_{12,1}^{\left(k\right)}=\frac{\partial \tau_{12}^{\left(k\right)}}{\partial s_{1}}\right|}_{\left(ijm\right)}≅\sum_{r=1}^{I_{N}}\varsigma_{ir}^{s_{1}\left(1\right)}\tau_{12\left(rjm\right)}^{\left(k\right)},\;\;\tau_{12,2}^{\left(k\right)}={\left.\frac{\partial \tau_{12}^{\left(k\right)}}{\partial s_{2}}\right|}_{\left(ijm\right)}≅\sum_{r=1}^{I_{M}}\varsigma_{jr}^{s_{2}\left(1\right)}\tau_{12\left(irm\right)}^{\left(k\right)} \end{array}$$
where the notations $\varsigma_{ir}^{\left(1\right)}=\varsigma_{ir}^{s_{1}\left(1\right)}$ and $\varsigma_{jr}^{\left(1\right)}=\varsigma_{jr}^{s_{2}\left(1\right)}$ mean that the GDQ rule is applied along $s_{1}$ and $s_{2}$ parametric line, respectively. Note that the partial derivatives of the stresses reported in Equation (78) are not solved according to the semi-analytical procedure, because in this case they should be evaluated for each n,m of the trigonometric expansion of Equation (53), and the post-processing recovery procedure should be applied many times. In contrast, the adoption of the GDQ numerical method allows one to apply directly the procedure to the expanded solution. The out-of-plane stress components $\tau_{\;13}^{\left(k\right)}$ and $\tau_{\;23}^{\left(k\right)}$ are evaluated from the first two three-dimensional equilibrium Equation (72) of a doubly-curved shell, which are written in each k-th layer of the shell in a discrete form as follows:(79)$$\begin{array}{l} \sum_{r=1}^{I_{T}}\varsigma_{\tilde{m}r}^{\zeta \left(1\right)}{\bar{\tau }}_{13\left(ij\left(\left(k-1\right)I_{T}+r\right)\right)}^{\left(k\right)}+\left(\frac{2}{R_{\;1}+\zeta_{m}}+\frac{1}{R_{\;2}+\zeta_{m}}\right){\bar{\tau }}_{13\left(ijm\right)}^{\left(k\right)}=a_{\left(ijm\right)}^{\left(k\right)}\\ \sum_{r=1}^{I_{T}}\varsigma_{\tilde{m}r}^{\zeta \left(1\right)}{\bar{\tau }}_{23\left(ij\left(\left(k-1\right)I_{T}+r\right)\right)}^{\left(k\right)}+\left(\frac{1}{R_{\;1}+\zeta_{m}}+\frac{2}{R_{\;2}+\zeta_{m}}\right){\bar{\tau }}_{23\left(ijm\right)}^{\left(k\right)}=b_{\left(ijm\right)}^{\left(k\right)} \end{array}$$

The parameters $a^{\left(k\right)}$ and $b^{\left(k\right)}$, dependent on the membrane stresses $\sigma_{\;1}^{\left(k\right)},\sigma_{\;2}^{\left(k\right)}$ and $\tau_{\;12}^{\left(k\right)}$, are written in a discrete form as follows:(80)$$\begin{array}{l} a_{\left(ijm\right)}^{\left(k\right)}=-\frac{1}{1+\zeta_{m}/R_{\;1}}{\left.\frac{\partial \sigma_{\;1}^{\left(k\right)}}{\partial s_{1}}\right|}_{\left(ijm\right)}-\frac{1}{1+\zeta_{m}/R_{\;2}}{\left.\frac{\partial \tau_{\;12}^{\left(k\right)}}{\partial s_{2}}\right|}_{\left(ijm\right)}\\ b_{\left(ijm\right)}^{\left(k\right)}=-\frac{1}{1+\zeta_{m}/R_{\;1}}{\left.\frac{\partial \tau_{12}^{\left(k\right)}}{\partial s_{1}}\right|}_{\left(ijm\right)}-\frac{1}{1+\zeta_{m}/R_{\;2}}{\left.\frac{\partial \sigma_{2}^{\left(k\right)}}{\partial s_{2}}\right|}_{\left(ijm\right)} \end{array}$$

The solution of Equation (79) is derived numerically by means of the GDQ method. On the other hand, the boundary conditions are enforced at the bottom surface of the shell, remembering the reciprocity principle of stress components. In other words, the out-of-plane shear stresses must be at the bottom of the shell, equal to the in-plane loads $q_{1}^{\left(-\right)}$ and $q_{2}^{\left(-\right)}$. In the same way, at the top surface of the shell, the shear stress profile must guarantee equilibrium with the external loads $q_{1}^{\left(+\right)}$ and $q_{2}^{\left(+\right)}$:(81)$$\begin{array}{l} \tau_{13}^{\left(1\right)}\left(\zeta =-h/2\right)=q_{1}^{\left(-\right)}\;\;\;\;\mathrm{or}\;\;\;\;\;\tau_{13}^{\left(l\right)}\left(\zeta =h/2\right)=q_{1}^{\left(+\right)}\\ \tau_{23}^{\left(1\right)}\left(\zeta =-h/2\right)=q_{2}^{\left(-\right)}\;\;\;\;\;\mathrm{or}\;\;\;\;\;\tau_{23}^{\left(l\right)}\left(\zeta =h/2\right)=q_{2}^{\left(+\right)} \end{array}$$

The boundary conditions introduced in the previous equation are written below in discrete form:(82)$${\bar{\tau }}_{13\left(ij1\right)}^{\left(1\right)}=q_{1\left(ij\right)}^{\left(-\right)},\;\;\;\;\;{\bar{\tau }}_{23\left(ij1\right)}^{\left(1\right)}=q_{2\left(ij\right)}^{\left(-\right)}$$
(83)$$\tau_{13\left(ij\left(lI_{T}\right)\right)}^{\left(l\right)}=q_{1\left(ij\right)}^{\left(+\right)},\;\;\;\;\;\tau_{23\left(ij\left(lI_{T}\right)\right)}^{\left(l\right)}=q_{2\left(ij\right)}^{\left(+\right)}$$

At this point, a numerical solution is found for k=1, taking into account the external constraints of Equation (82). Once the equilibrium Equation (79) is solved in the first layer, the boundary conditions for the generic k-th layer with k=2,…,l are assessed starting from the results obtained for k−1. Note that the discrete points associated to the indexes $m=\left(k-1\right)I_{T}$ and $m=\left(k-1\right)I_{T}+1$ are located at the same height in the interface region along the thickness direction, therefore, the relations reported below can be enforced at the interface between two adjacent layers:(84)$$\begin{array}{l} {\bar{\tau }}_{13\left(ij\left(\left(k-1\right)I_{T}+1\right)\right)}^{\left(k\right)}={\bar{\tau }}_{13\left(ij\left(\left(k-1\right)I_{T}\right)\right)}^{\left(k-1\right)}\\ {\bar{\tau }}_{23\left(ij\left(\left(k-1\right)I_{T}+1\right)\right)}^{\left(k\right)}={\bar{\tau }}_{23\left(ij\left(\left(k-1\right)I_{T}\right)\right)}^{\left(k-1\right)} \end{array}$$

Finally, the boundary conditions of Equation (83) are enforced at the top surface of the solid, remembering that the solution of the differential system of Equation (79) is defined as less than a linear transformation. For this reason, the through-the-thickness profiles of the stresses ${\bar{\tau }}_{\;13}^{\left(k\right)}$ and ${\bar{\tau }}_{\;23}^{\left(k\right)}$, derived numerically, are corrected by adding a linear term dependent on the thickness coordinate $\zeta_{m}$ [16], as shown in Figure 3:
(85)$$\begin{array}{cc} \begin{array}{l} \tau_{13\left(ijm\right)}^{\left(k\right)}={\bar{\tau }}_{13\left(ijm\right)}^{\left(k\right)}+\frac{q_{1\left(ij\right)}^{\left(+\right)}-{\bar{\tau }}_{13\left(ij\left(l\;I_{T}\right)\right)}^{\left(l\right)}}{h}\left(\zeta_{m}+\frac{h}{2}\right)\\ \tau_{23\left(ijm\right)}^{\left(k\right)}={\bar{\tau }}_{23\left(ijm\right)}^{\left(k\right)}+\frac{q_{2\left(ij\right)}^{\left(+\right)}-{\bar{\tau }}_{23\left(ij\left(l\;I_{T}\right)\right)}^{\left(l\right)}}{h}\left(\zeta_{m}+\frac{h}{2}\right) \end{array} & \;\;\;\;\;\;m=1,\ldots ,l\;I_{T} \end{array}$$

At this point, the derivative of $\tau_{\;13}^{\left(k\right)}$ and $\tau_{\;23}^{\left(k\right)}$ shear stresses with respect to $s_{1},s_{2}$ can be evaluated with the GDQ method as follows:(86)$$\begin{array}{cc} \begin{array}{l} {\left.\tau_{13,1}^{\left(k\right)}=\frac{\partial \tau_{13}^{\left(k\right)}}{\partial s_{1}}\right|}_{\left(ijm\right)}≅\sum_{r=1}^{I_{N}}\varsigma_{ir}^{s_{1}\left(1\right)}\tau_{13\left(rjm\right)}^{\left(k\right)}\\ {\left.\tau_{23,2}^{\left(k\right)}=\frac{\partial \tau_{23}^{\left(k\right)}}{\partial s_{2}}\right|}_{\left(ijm\right)}≅\sum_{r=1}^{I_{M}}\varsigma_{jr}^{s_{2}\left(1\right)}\tau_{23\left(irm\right)}^{\left(k\right)} \end{array} & \;\;\;\;\;\;m=1,\ldots ,l\;I_{T} \end{array}$$

The third equilibrium equation of Equation (72), reported below in discrete form, is adopted for the derivation of the actual profile of the normal stress $\sigma_{3}^{\left(k\right)}$:(87)$$\sum_{r=1}^{I_{\;T}}\varsigma_{\tilde{m}r}^{\zeta \left(1\right)}{\bar{\sigma }}_{3\left(ij\left(\left(k-1\right)I_{T}+r\right)\right)}^{\left(k\right)}+\left(\frac{1}{R_{1}+\zeta_{m}}+\frac{1}{R_{2}+\zeta_{m}}\right){\bar{\sigma }}_{3\left(ijm\right)}^{\left(k\right)}=c_{\left(ijm\right)}^{\left(k\right)}$$
where the parameter $c_{\left(ijm\right)}^{\left(k\right)}$ accounts for the following expression:(88)$$c_{\left(ijm\right)}^{\left(k\right)}=\frac{\sigma_{1\left(ijm\right)}^{\left(k\right)}}{R_{1}+\zeta_{m}}+\frac{\sigma_{2\left(ijm\right)}^{\left(k\right)}}{R_{2}+\zeta_{m}}-\frac{1}{1+\zeta_{m}/R_{1}}{\left.\frac{\partial \tau_{13}^{\left(k\right)}}{\partial s_{1}}\right|}_{\left(ijm\right)}-\frac{1}{1+\zeta_{m}/R_{2}}{\left.\frac{\partial \tau_{23}^{\left(k\right)}}{\partial s_{2}}\right|}_{\left(ijm\right)}$$

Two possible sets of boundary conditions are considered for the derivation of the numerical solution of Equation (87), setting $q_{3}^{\left(+\right)}$ and $q_{3}^{\left(-\right)}$ the value of the external actions oriented along the normal direction applied at the top and bottom surfaces of the shell, respectively:(89)$$\begin{array}{l} {\bar{\sigma }}_{3}^{\left(1\right)}\left(\zeta =-h/2\right)=q_{3}^{\left(-\right)}\\ {\bar{\sigma }}_{3}^{\left(l\right)}\left(\zeta =h/2\right)=q_{3}^{\left(+\right)} \end{array}$$

In discrete form, the last two relations become:(90)$$\begin{array}{l} {\bar{\sigma }}_{3\left(ij1\right)}^{\left(1\right)}=q_{3\left(ij\right)}^{\left(-\right)}\\ {\bar{\sigma }}_{3\left(ij\left(l\;I_{T}\right)\right)}^{\left(l\right)}=q_{3\left(ij\right)}^{\left(+\right)} \end{array}$$

Following the same approach as Equation (84), once the boundary condition in Equation (89) is employed for the derivation of the numerical solution of Equation (87) of the first layer $\left(k=1\right)$, the following boundary condition is considered for the numerical assessment of the equilibrium Equation (87) in the remaining laminae, namely for k=2,…,l:(91)$${\bar{\sigma }}_{3\left(ij\left(\left(k-1\right)I_{T}+1\right)\right)}^{\left(k\right)}={\bar{\sigma }}_{3\left(ij\left(\left(k-1\right)I_{T}\right)\right)}^{\left(k-1\right)}$$

The second boundary condition of Equation (90) is applied by means of a linear correction [16] of the profile of the normal stress ${\bar{\sigma }}_{3}^{\left(k\right)}$ derived from Equation (87):(92)$$\sigma_{3\left(ijm\right)}^{\left(k\right)}={\bar{\sigma }}_{3\left(ijm\right)}^{\left(k\right)}+\frac{q_{3\left(ij\right)}^{\left(+\right)}-{\bar{\sigma }}_{3\left(ij\left(l\;I_{T}\right)\right)}^{\left(l\right)}}{h}\left(\zeta_{m}+\frac{h}{2}\right)$$

The correction of the through-the-thickness profile of the normal stress has been represented in Figure 3. Once the three-dimensional stresses are derived from the present recovery procedure, the constitutive relationship of Equation (17) is used for the derivation of the updated profile of the out-of-plane strain components $\gamma_{13}^{\left(k\right)},\gamma_{23}^{\left(k\right)}$ and $\varepsilon_{3}^{\left(k\right)}$. To this end, the following relation is considered at each point $\left(s_{1i},s_{2j},\zeta_{m}\right)$ of the three-dimensional solid:(93)$$\begin{array}{l} {\bar{C}}_{44\left(m\right)}^{\left(k\right)}\gamma_{13\left(ijm\right)}^{\left(k\right)}=\tau_{13\left(ijm\right)}^{\left(k\right)}\\ {\bar{C}}_{55\left(m\right)}^{\left(k\right)}\gamma_{23\left(ijm\right)}^{\left(k\right)}=\tau_{23\left(ijm\right)}^{\left(k\right)}\\ {\bar{C}}_{33\left(m\right)}^{\left(k\right)}\varepsilon_{3\left(ijm\right)}^{\left(k\right)}=\sigma_{3\left(ijm\right)}^{\left(k\right)}-{\bar{C}}_{13\left(m\right)}^{\left(k\right)}\varepsilon_{1\left(ijm\right)}^{\left(k\right)}-{\bar{C}}_{23\left(m\right)}^{\left(k\right)}\varepsilon_{2\left(ijm\right)}^{\left(k\right)} \end{array}$$

Starting from the previous equation, the expression of the quantities $\gamma_{\;13\left(ijm\right)}^{\left(k\right)},\gamma_{\;23\left(ijm\right)}^{\left(k\right)}$ and $\varepsilon_{\;3\left(ijm\right)}^{\left(k\right)}$ is easily derived.

Finally, the out-of-plane strain components are introduced in the three-dimensional constitutive relation of Equation (77). In this way, the membrane stresses $\sigma_{\;1\left(ijm\right)}^{\left(k\right)},\sigma_{\;2\left(ijm\right)}^{\left(k\right)}$ and $\tau_{\;12\left(ijm\right)}^{\left(k\right)}$ are corrected because, in the previous step, they were calculated without considering the three-dimensional equilibrium equations.

## 9. Application and Results

In the present section, some examples of investigation are shown, where the present semi-analytical theory is adopted for the derivation of the static response of structures of different curvatures and lamination schemes for different load cases. The accuracy of the solution is validated with success with respect to highly computationally demanding three-dimensional finite element models, shown in Figure 4, as implemented in the ABAQUS code. In all examples, it is shown that the present semi-analytical solution, together with the recovery of stresses and strains, can accurately predict the three-dimensional response of curved and laminated structures with a reduced computational cost. Furthermore, the convergence of the method is studied when load shapes of practical interest are considered, like uniform, concentrated, line, and hydrostatic loads. Finally, some examples are presented where curved and layered structures are subjected to generally-shaped pressures. In all the examples, lamination schemes are made starting from layers of graphite-epoxy $\left(\rho^{\left(k\right)}=\;1450\;\;\mathrm{kg}/m^{3}\right)$, whose engineering constants, obtained from Ref. [101], are reported in the following:(94)$$\begin{array}{ccc} E_{1}^{\left(k\right)}=137.90\;\;\mathrm{GPa} & G_{23}^{\left(k\right)}=6.21\;\;\mathrm{GPa} & \nu_{23}^{\left(k\right)}=0.49\\ E_{2}^{\left(k\right)}=E_{3}^{\left(k\right)}=8.96\;\;\mathrm{GPa} & G_{12}^{\left(k\right)}=G_{13}^{\left(k\right)}=7.10\;\;\mathrm{GPa} & \nu_{12}^{\left(k\right)}=\nu_{13}^{\left(k\right)}=0.30 \end{array}$$

In the previous relation, the quantities $\nu_{\;12}^{\left(k\right)},\nu_{\;13}^{\left(k\right)},\nu_{\;23}^{\left(k\right)}$ denote the Poisson’s coefficients of the orthotropic materials, whereas $E_{\;1}^{\left(k\right)},E_{\;2}^{\left(k\right)},E_{\;3}^{\left(k\right)}$ and $G_{\;12}^{\left(k\right)},G_{\;13}^{\left(k\right)},G_{\;23}^{\left(k\right)}$ are the elastic moduli and the shear moduli, respectively. They are related to the three-dimensional stiffness constants $C_{\;ij}^{\left(k\right)}=E_{\;ij}^{\left(k\right)}$ with i,j=1,…,6 of Equation (18) according to the procedure extensively detailed in Ref. [16]. The stiffness constants $C_{\;ij}^{\left(k\right)}$ of soft orthotropic layers, denoted by graphite-epoxy soft5 and graphite-epoxy soft10, are five and ten times softer, respectively, than those of the graphite-epoxy of Equation (94). As a consequence, the engineering constants of the graphite-epoxy soft5, whose density is thus $\rho^{\left(k\right)}=\;290\;\;\mathrm{kg}/m^{3}$, the following aspect:(95)$$\begin{array}{ccc} E_{1}^{\left(k\right)}=27.58\;\;\mathrm{GPa} & G_{23}^{\left(k\right)}=1.242\;\;\mathrm{GPa} & \nu_{23}^{\left(k\right)}=0.49\\ E_{2}^{\left(k\right)}=E_{3}^{\left(k\right)}=1.792\;\;\mathrm{GPa} & G_{12}^{\left(k\right)}=G_{13}^{\left(k\right)}=1.42\;\;\mathrm{GPa} & \nu_{12}^{\left(k\right)}=\nu_{13}^{\left(k\right)}=0.30 \end{array}$$

In the same way, the mechanical properties of the graphite-epoxy soft10 $\left(\rho^{\left(k\right)}=\;145\;\;\mathrm{kg}/m^{3}\right)$ can be written as:(96)$$\begin{array}{ccc} E_{1}^{\left(k\right)}=13.790\;\;\mathrm{GPa} & G_{23}^{\left(k\right)}=0.621\;\;\mathrm{GPa} & \nu_{23}^{\left(k\right)}=0.49\\ E_{2}^{\left(k\right)}=E_{3}^{\left(k\right)}=0.896\;\;\mathrm{GPa} & G_{12}^{\left(k\right)}=G_{13}^{\left(k\right)}=0.710\;\;\mathrm{GPa} & \nu_{12}^{\left(k\right)}=\nu_{13}^{\left(k\right)}=0.30 \end{array}$$

In each simulation, the lamination scheme consists of three layers of thicknesses $h_{\;1}=h_{\;3}=0.03\;\;m$ and $h_{\;2}=0.04\;\;m$ with material orientation angles equal to $\vartheta^{\left(1\right)}=\vartheta^{\left(3\right)}=0$ and $\vartheta^{\left(2\right)}=\pi /2$. Finally, the recovery procedure has been applied setting $I_{T}=31$ discrete points in the thickness direction for each k-th layer. Note that the computational cost of the present semi-analytical formulation is here intended as the number of terms, denoted by $\tilde{N}$ and $\tilde{M}$, occurring in the harmonic expansion of the solution according to Equation (53). No details are given on the computational time, whose aspect depends on the machine properties and on the numerical implementation of the linear system (59), which is not the main focus of this work.

The first example consists of a simply-supported rectangular plate of dimensions $L_{\;1}=2\;\;m$ and $L_{\;2}=1\;\;m$ made of three layers of graphite-epoxy with properties as in Equation (94) of thicknesses $h_{\;1}=h_{\;3}=0.03\;\;m$ and $h_{\;2}=0.04\;\;m$ subjected to two different patch loads, one at the top surface and the other at the bottom surface with magnitudes ${\bar{q}}_{\;3}^{\left(+\right)}=-7\times 10^{5}\;N/m^{2}$ and ${\bar{q}}_{\;3}^{\left(-\right)}=-3\times 10^{5}\;N/m^{2}$, respectively. The position and shape parameters of the load distribution in hand have been selected so that the external pressure is applied in a quarter of the physical domain. On the other hand, the through-the-thickness distribution of the three-dimensional kinematic and mechanical quantities have been evaluated in a region where the external load is not applied, namely $\left(0.25L_{1},0.25L_{2}\right)$. In Figure 5 the reader can find some information regarding the convergence rate of the present semi-analytical solution, which is computed by the following percentage error $e_{\%}$:(97)$$e_{\%}=\left|\frac{w_{\tilde{N}}-w_{\mathrm{FEM}}}{w_{\mathrm{FEM}}}\right|\times 100\;\%$$
where $w_{\tilde{N}}$ denotes the vertical deflection of the central point of the three-dimensional solid derived from Equation (75), while $w_{\mathrm{FEM}}=0.00150236\;\;m$ is the corresponding value derived from the finite element simulation.

As visible in Table 1, a very rapid convergence rate is found for a limited number of terms within the harmonic expansion (53). In this way, the results of the simulation are shown to be very stable for the selected wave numbers. Note that the selected value of $\tilde{N}=\tilde{M}$ not only depends on the convergence of the vertical deflection to the reference value, but also on the fulfillment of the loading conditions at the top and bottom surfaces of the plate under consideration.

The distributions of the displacement field components, the three-dimensional strain, and the stresses have been reported in Figure 5, Figure 6 and Figure 7, respectively. For each quantity, a 3D FEM model of parabolic C3D20 elements, consisting of 582327 DOFs, has been adopted for the derivation of a reference solution. Furthermore, a two-dimensional numerical solution is derived with the GDQ method, accounting for the FSDT and TSDT theories as well as the EDZ4. The physical domain is discretized with a two-dimensional grid made of $I_{N}=I_{M}=31$ discrete points, following the CGL distribution. The present semi-analytical theory has been used with the EDZ4 displacement field assumption, setting $\tilde{N}=\tilde{M}=150$. As can be seen from Figure 5, very accurate results are provided in terms of the displacement field components $U_{1},U_{2},U_{3}$. On the other hand, in Figure 6, it is shown that the recovery procedure is determining the correct prediction of the out-of-plane strain components $\gamma_{13},\gamma_{23},\varepsilon_{3}$, especially in the central layer. Furthermore, very accurate results are obtained for in-plane quantities $\varepsilon_{1},\varepsilon_{2},\gamma_{12}$. As far as the three-dimensional stress components are concerned, in Figure 7, it is shown that the reconstruction of out-of-plane stresses from the semi-analytical Navier solution by means of the three-dimensional constitutive relationship of Equation (17) leads to erroneous results, especially for the $\sigma_{3}$ normal stress. On the other hand, when the equilibrium-based recovery procedure is applied, the results coming from the two-dimensional semi-analytical solution, denoted by (E), perfectly match those coming from the 3D FEM simulation.

The next example takes into account the same rectangular panel subjected to hydrostatic loads. More specifically, the first load of magnitude ${\bar{q}}_{\;3}^{\left(+\right)}=-7\times 10^{5}\;N/m^{2}$ and directed along $\alpha_{1}$ principal direction is applied at the top surface, whereas the second load of magnitude ${\bar{q}}_{\;3}^{\left(-\right)}=-3\times 10^{5}\;N/m^{2}$, applied at the bottom surface, is directed along $\alpha_{2}$ principal direction. Three different load cases have been considered, denoted by H1, H2, and H3. In the first one, only the top surface is loaded, whereas in the second one, the external load is applied only to the bottom surface. Finally, in the H3 load case, the two hydrostatic loads have been considered together. Thickness plots calculated at the point located at $\left(0.25\;L_{1},0.25\;L_{2}\right)$ have been collected in Figure 8, Figure 9 and Figure 10.

For each load configuration, the semi-analytical solution has been calculated with $\tilde{N}=\tilde{M}=150$. The results are compared to those of a 3D FEM simulation and to those coming from a GDQ numerical model with classical and higher-order theories, showing a very good agreement between different approaches. When lower order theories like FSDT and TSDT are employed, a slight discrepancy in results can be seen in the thickness plot of $U_{3}$, reported in Figure 8, whereas the EDZ4 theory perfectly matches the three-dimensional predictions of all quantities in each load case. With particular reference to both in-plane and out-of-plane strain and stress profiles in Figure 9 and Figure 10, it should be noted that for each case, the loading conditions are perfectly respected. Furthermore, the solution coming from the H3 load case can be seen as the sum of H1 and H2, due to the additivity property of the linear solutions calculated previously.

At this point, a laminated cylindrical panel of radius $R_{\;2}=1.2\;\;m$ is considered with $L_{\;1}=2\;\;m$ and $\left(\alpha_{2}^{0},\alpha_{2}^{1}\right)=\left(-\pi /3,\pi /3\right)$. Two externally concentrated loads of magnitude ${\bar{q}}_{\;3}^{\left(+\right)}=-2000\;N$ and ${\bar{q}}_{\;3}^{\left(-\right)}=-2000\;N$ are applied at the top and bottom surfaces, respectively. The position parameters are $s_{10}=0.5\;L_{1}$ and $s_{20}=0.5\;L_{2}$, selected so that the structure is loaded in the center of the physical domain. The lamination scheme consists of two external layers of graphite-epoxy, as defined in Equation (94), whereas the central core is made of graphite-epoxy soft10, as defined in Equation (96).

Thickness plots are provided at the point located at $\left(0.25\;L_{1},0.25\;L_{2}\right)$, and they are collected in Figure 11, Figure 12 and Figure 13. A 3D FEM model with 741975 DOFs made of parabolic C3D20 brick elements has been developed, and a three-dimensional reference solution has been provided.

In addition, a numerical solution based on the GDQ numerical technique has been derived, taking into account both the FSDT and the TSDT displacement field assumptions. The zigzag effects are clearly visible in the deflection of the panel because an abrupt variation of the material stiffness occurs between two adjacent laminae. For this reason, a parametric investigation has been conducted with the semi-analytical approach with $\tilde{N}=\tilde{M}=500$, and the static response of the cylindrical panel has been evaluated employing various higher-order theories. As can be seen from Figure 11, the exact solutions accounting for the EDZ3 and the EDZ4 higher-order theories perfectly match the predictions of the three-dimensional FEM reference model. Similar considerations can be made for the plots of the three-dimensional strain components in Figure 12. The adoption of a higher-order displacement field is key for the prediction of both in-plane and out-of-plane kinematic quantities. In the same way, the three-dimensional stress components of Figure 13 are well described by the EDZ4 model, but with a significantly reduced computational cost if compared to the 3D-FEM simulation.

The EDZ4 model has been taken as a reference also in the next example, where the same simply-supported cylindrical panel, made of three layers of graphite-epoxy, as defined in Equation (94), has been loaded with a line load of magnitude ${\bar{q}}_{\;3}^{\left(+\right)}=-4.58\times 10^{3}\;N/m$ distributed along $\alpha_{1}$ principal direction, located at $s_{20}=0.75\;L_{2}$. The thickness plots are evaluated at $\left(0.25\;L_{1},0.25\;L_{2}\right)$. The GDQ numerical solution has been calculated with $I_{N}=I_{M}=31$, accounting for FSDT, TSDT, and EDZ4 theories, whereas the semi-analytical Navier solution is derived with the EDZ4 kinematic assumption setting $\tilde{N}=\tilde{M}=500$. Thickness plots of displacement field, strain, and stress components are reported in Figure 14, Figure 15 and Figure 16, respectively. As shown in Figure 14, classical approaches like the FSDT and the TSDT provide a uniform value of $U_{3}$, and the stretching effect predicted by the 3D-FEM model cannot be evaluated. On the other hand, when the EDZ4 theory is adopted, the parabolic distribution of the displacement field components is predicted with a sufficient level of accuracy. The distribution of the three-dimensional strain components, reported in Figure 15, provides refined results with respect to 3D FEM if a higher-order displacement field is considered. It is important to underline that the recovery procedure provides very accurate results because of the external load cases, which are the boundary conditions of the problem; therefore, an accurate distribution of the stress components is derived. Finally, from the results reported in Figure 16, it can be said that the three-dimensional in-plane stress profiles derived from the 3D FEM model are in line with those provided by both the GDQ numerical solution and the present approach.

Next example points out the high convergence rate of the present semi-analytical method in the case of general loads. Let us consider a cylindrical panel of graphite epoxy with an internal region made of graphite-epoxy soft10 subjected to different external pressures applied at the extrados of the shell. More specifically, two hydrostatic loads of magnitudes ${\bar{q}}_{\;3}^{\left(+\right)}=-7\times 10^{5}\;N/m^{2}$ and ${\bar{q}}_{\;3}^{\left(-\right)}=-4\times 10^{5}\;N/m^{2}$ are considered, which are denoted by H1 and H2, respectively. In addition, a uniform pressure (U) of magnitude ${\bar{q}}_{\;3}^{\left(+\right)}=-2\times 10^{5}\;N/m^{2}$ acting along the thickness direction is applied. Five different load cases are considered, accounting for different combinations of the external pressures introduced previously. They are summarized as following:(98)$$\begin{array}{ccc} \mathrm{Case}\;01\;\;\left(C1\right) & \to & H1\\ \mathrm{Case}\;02\;\;\left(C2\right) & \to & H2\\ \mathrm{Case}\;03\;\;\left(C3\right) & \to & H1+H2\\ \mathrm{Case}\;04\;\;\left(C4\right) & \to & U\\ \mathrm{Case}\;05\;\;\left(C5\right) & \to & H1+H2+U \end{array}$$

For each load case, the thickness plots are evaluated $\left(0.25\;L_{1},0.25\;L_{2}\right)$ and collected in Figure 17, Figure 18 and Figure 19, and a 3D finite element reference solution has been derived with commercial software. In addition, a GDQ-based numerical solution has been evaluated with the EDZ4 theory, which matches the 3D FEM predictions. The semi-analytical solution is evaluated with the EDZ4 displacement field assumption, set $\tilde{N}=\tilde{M}=150$ in each load case. The results are in line with the reference solution in terms of displacement field components, as can be seen in Figure 17. It should be noted that the selected lamination scheme presents some zigzag effects, especially for case C5. The adoption of Murakami’s zigzag function in the present model allows one to consider the piecewise inclination of the profile of the displacement field components. Similar considerations are made for the three-dimensional strain components of Figure 18, where the 3D FEM reference solution is perfectly predicted in a reduced amount of time by the higher-order semi-analytical model, and a good level of accuracy is also reached in the central region of the structure. In order to reduce the computational effort of the simulation, the results of more complicated load cases like C3 are obtained from the algebraic sum of C1 and C2 simulations. In the same way, load case C5 is derived from the sum of C3 and C4. As a consequence, the results of the simulations referred to in C3 and C5 can be efficiently obtained as an algebraic sum of the profiles of all kinematic and mechanical quantities recovered previously in the corresponding two-dimensional semi-analytical solutions, as shown in Figure 19 in the case of the three-dimensional stress components. The recovery procedure is not applied in C3 and C5 because the equilibrium equations in the thickness direction have already been solved independently in C1, C2, and C4. The 3D FEM numerical predictions are perfectly matched by the semi-analytical model, with a significantly reduced computational cost and time.

The next example presents a shallow spherical panel of radius R=3  m, whose physical domain is defined so that $\left(\alpha_{1}^{0},\alpha_{1}^{1}\right)=\left(5\pi /12,7\pi /12\right)$ and $\left(\alpha_{2}^{0},\alpha_{2}^{1}\right)=\left(-\pi /9,\pi /9\right)$. Three different lamination schemes are considered made of two external layers of graphite-epoxy (94), whereas the central core consists of graphite-epoxy (94), graphite-epoxy soft5 (95) and graphite-epoxy soft10 (96) for cases C1, C2, and C3, respectively. In the first load case, the panel under consideration is subjected to sinusoidal loads of magnitude ${\bar{q}}_{\;3}^{\left(+\right)}=-7\times 10^{5}\;N/m^{2}$ and ${\bar{q}}_{\;3}^{\left(-\right)}=-3\times 10^{5}\;N/m^{2}$ with $\tilde{N}=\tilde{M}=1$. Two reference solutions are provided, developed with 3D finite element solution and a two-dimensional GDQ-based formulation, accounting for the EDZ4 displacement field theory. 

The 3D FEM model is made of parabolic C3D20 brick elements with a total number of 293847 DOFs, whereas the 2D-GDQ model is built starting from a two-dimensional CGL grid with $I_{N}=I_{M}=31$. The solution obtained from the semi-analytical simulation is based on the EDZ3 and EDZ4 theories. The thickness plots, reported in Figure 20, Figure 21 and Figure 22, are provided for the point located at $\left(0.25\;L_{1},0.25\;L_{2}\right)$ within the physical domain, where the quantities $L_{\;1}$ and $L_{\;2}$ have been defined in Equation (50).

As shown in Figure 20, the reduction of the stiffness of the central core of the panel leads to a typical zigzag profile of the in-plane displacement field components $U_{\;1}$ and $U_{\;2}$. Furthermore, when the central layer stiffness is reduced, the vertical deflection $U_{\;3}$ increases. Similar considerations can be made for the strain components of Figure 21, where $\varepsilon_{\;1},\varepsilon_{\;2}$ and $\varepsilon_{\;3}$ assume in the central lamina a non-linear profile in the case of C2 and C3, whereas the value of $\gamma_{\;12},\gamma_{\;13}$ and $\gamma_{\;23}$ is increased. Furthermore, for all strain components, a good agreement can be seen between the predictions of various numerical approaches and the semi-analytical results.

Referring to the results of Figure 22, the values of both in-plane and out-of-plane stress components derived from both the 3D FEM and the GDQ are predicted with success by the present semi-analytical formulation. Furthermore, the boundary conditions are respected at the top and bottom surfaces.

Once the semi-analytical model of the spherical panel has been validated for the case of sinusoidal loads ($\tilde{N}=\tilde{M}=1$), the linear static response of the same structure is derived for the case of uniform transverse loads ${\bar{q}}_{\;3}^{\left(+\right)}=-7\times 10^{5}\;N/m^{2}$ and ${\bar{q}}_{\;3}^{\left(-\right)}=-3\times 10^{5}\;N/m^{2}$ applied at the top and bottom surfaces, respectively. In this case, the results obtained with the present semi-analytical theory have been derived setting $\tilde{N}=\tilde{M}=150$, taking into account the EDZ4 higher-order theory. The thickness plots are provided for the point $\left(0.25\;L_{1},0.25\;L_{2}\right)$ and collected in Figure 23, Figure 24 and Figure 25.

The reference solution has been calculated with a 3D-FEM model and some GDQ numerical simulations, based on the FSDT and the TSDT kinematic field assumptions. The profiles of the displacement field components in Figure 23 show that the predictions of the reference models can be obtained only when a higher-order displacement field is considered in the semi-analytical model. In fact, in the case of softcore lamination schemes, classical approaches like the FSDT and the TSDT are not consistent, whereas the results provided with the EDZ4 theory match the 3D-FEM predictions. In Figure 24, the through-the-thickness profiles of the three-dimensional strain components are provided. As can be seen, for both hardcore and softcore configurations of the stacking sequence, the present higher-order semi-analytical solution predicts with success the strain profiles provided by the three-dimensional model, even in the central softcore lamina. Furthermore, the presence of the zigzag function allows one to see what happens in the interface region between two adjacent laminae. 

As far as the three-dimensional stress components are concerned, Figure 25 shows that for each lamination scheme that has been investigated, the results of the three-dimensional model are well predicted when higher-order thickness functions are considered in the two-dimensional semi-analytical solution.

## 10. Conclusions

In the present manuscript, an efficient two-dimensional semi-analytical model has been presented for the evaluation of the static response of curved laminated panels subjected to arbitrary loads. The fundamental governing equations have been written according to the ESL approach with a generalized description of the kinematic field variable along the thickness direction. The solution to the semi-analytical problem has been provided with the Navier method, coupled with a post-processing recovery procedure based on the GDQ numerical technique. The model has been employed in some examples of investigation, where the three-dimensional linear static response of panels with different curvatures, lamination schemes, and load shapes has been derived and successfully compared to the numerical predictions coming from refined 3D FEM simulations. It has been shown that when the semi-analytical approach is used for the derivation of the solution, the GDQ-based recovery procedure allows the model to perfectly fulfil the load conditions in each point of the panel. In addition, the adoption of higher-order theories, together with stress and strain recovery, allows for a good level of accuracy in evaluating the three-dimensional behavior of structures with more cross-ply lamination schemes. Finally, a high level of accuracy is also seen in the case of softcore layers when a higher-order two-dimensional theory is considered, thus reducing the computational effort of each simulation. The numerical examples show that the EDZ4 theory is a valid tool for many lamination schemes, and the semi-analytical solution perfectly matches the 3D finite element predictions, especially in the case of rectangular plates and cylindrical panels. It has been shown that when uniformly-distributed patch and hydrostatic loads are applied to the panel, the convergence of results is seen for $\tilde{N}=\tilde{M}=150$, while further terms are required in the case of concentrated and line loads, namely $\tilde{N}=\tilde{M}=500$. It is shown that this issue does not depend on the geometry or lamination scheme, but only on the applied load shape. The present semi-analytical solution can be a valid alternative to well-established finite element simulations. In addition, among two-dimensional theories, it allows for a rapid and accurate solution of the problem for structures of constant curvatures with cross-ply whose lamination schemes are made of orthotropic materials. For this reason, it can be applied to fiber-reinforced composite materials as well as lattice and honeycomb panels and structures reinforced with dispersed short fibers without significant computational effort if compared to trustworthy numerical models.

## Figures and Tables

**Figure 1 materials-17-00588-f001:**
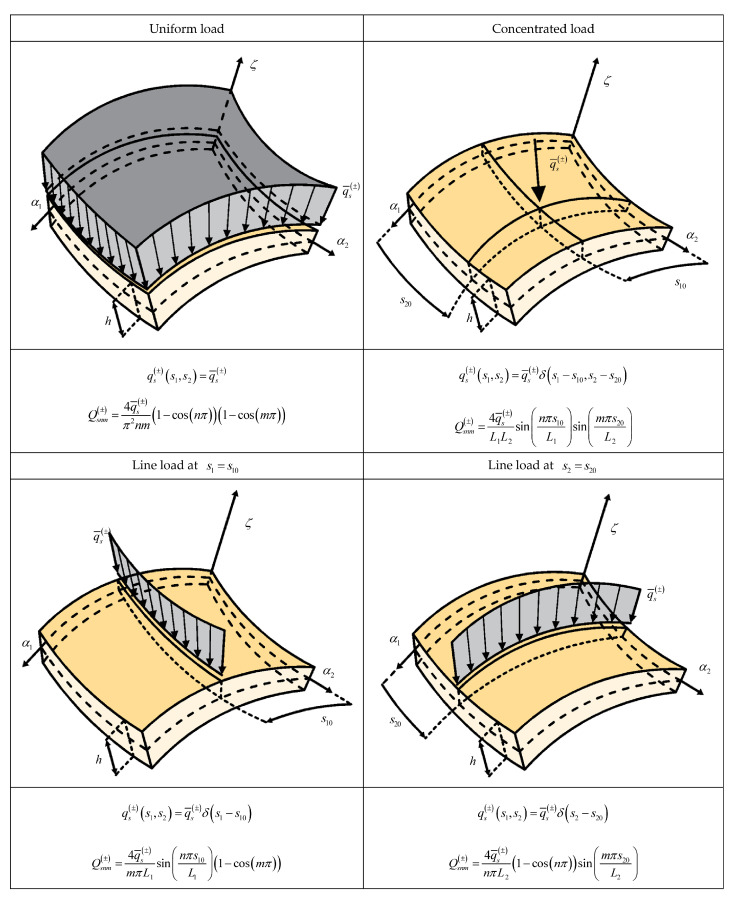
Coefficients for the harmonic expansion of the wave amplitudes $Q_{snm}^{\left(\pm \right)}=Q_{1nm}^{\left(\pm \right)},Q_{2nm}^{\left(\pm \right)},Q_{3nm}^{\left(\pm \right)}$ of external surface loads $q_{s}^{\left(\pm \right)}=q_{1}^{\left(\pm \right)},q_{2}^{\left(\pm \right)},q_{3}^{\left(\pm \right)}$ of reference magnitudes ${\bar{q}}_{s}^{\left(\pm \right)}={\bar{q}}_{1}^{\left(\pm \right)},{\bar{q}}_{2}^{\left(\pm \right)},{\bar{q}}_{3}^{\left(\pm \right)}$ applied on a doubly-curved shell panel.

**Figure 2 materials-17-00588-f002:**
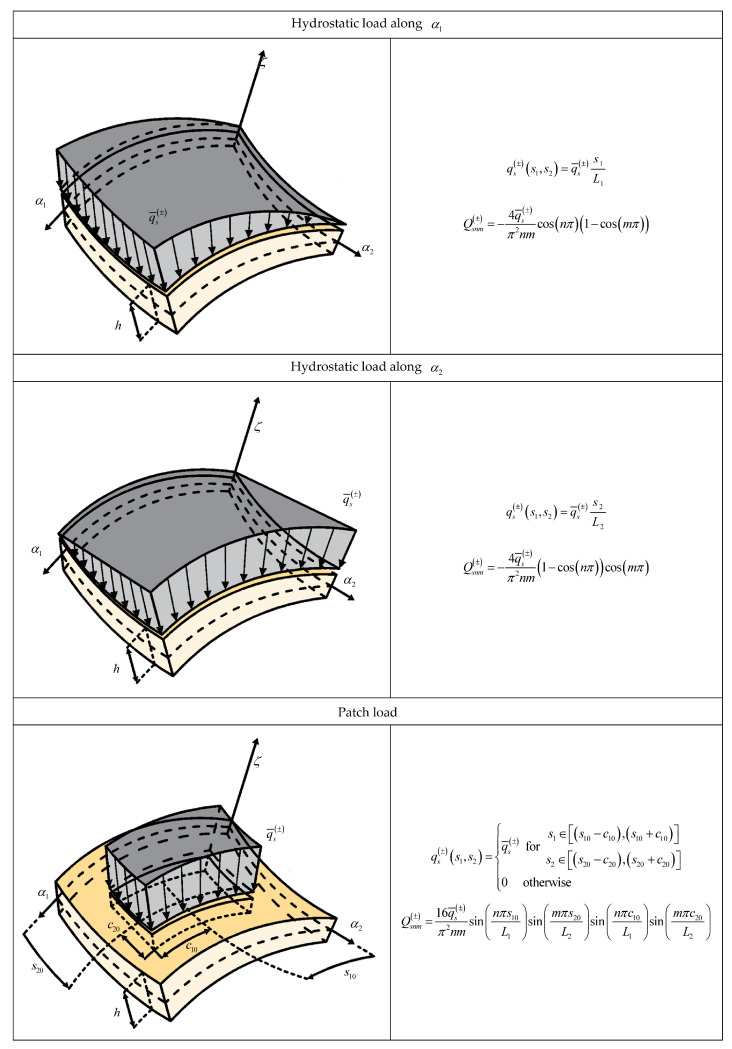
Further coefficients for the harmonic expansion of the wave amplitudes $Q_{snm}^{\left(\pm \right)}=Q_{1nm}^{\left(\pm \right)},Q_{2nm}^{\left(\pm \right)},Q_{3nm}^{\left(\pm \right)}$ of external surface loads $q_{s}^{\left(\pm \right)}=q_{1}^{\left(\pm \right)},q_{2}^{\left(\pm \right)},q_{3}^{\left(\pm \right)}$ of reference magnitudes ${\bar{q}}_{s}^{\left(\pm \right)}={\bar{q}}_{1}^{\left(\pm \right)},{\bar{q}}_{2}^{\left(\pm \right)},{\bar{q}}_{3}^{\left(\pm \right)}$ applied on a doubly-curved shell panel.

**Figure 3 materials-17-00588-f003:**
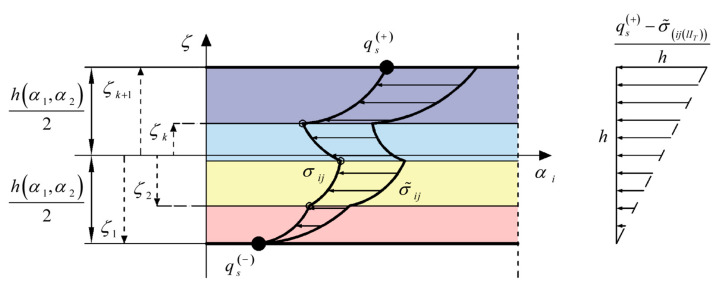
Linear correction $\sigma_{ij}=\tau_{13},\tau_{23},\sigma_{3}$ of the profile of the out-of-plane stress components by means of the external load $q_{s}^{\left(+\right)}=q_{1}^{\left(+\right)},q_{2}^{\left(+\right)},q_{3}^{\left(+\right)}$ acting at the top surface of the shell. The external pressure at the bottom surface, denoted by $q_{s}^{\left(-\right)}=q_{1}^{\left(-\right)},q_{2}^{\left(-\right)},q_{3}^{\left(-\right)}$, has been previously modeled as the boundary condition of the three-dimensional elasticity equation for the derivation of ${\tilde{\sigma }}_{ij}={\bar{\tau }}_{13},{\bar{\tau }}_{23},{\bar{\sigma }}_{3}$.

**Figure 4 materials-17-00588-f004:**
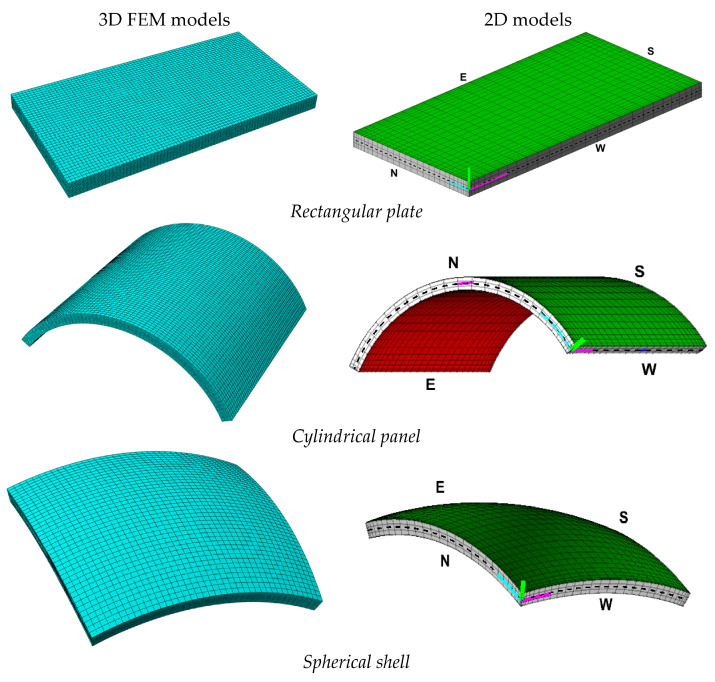
Representation of 3D FEM models of structures of various curvatures, and 2D GDQ-based models for the present semi-analytical formulation.

**Figure 5 materials-17-00588-f005:**
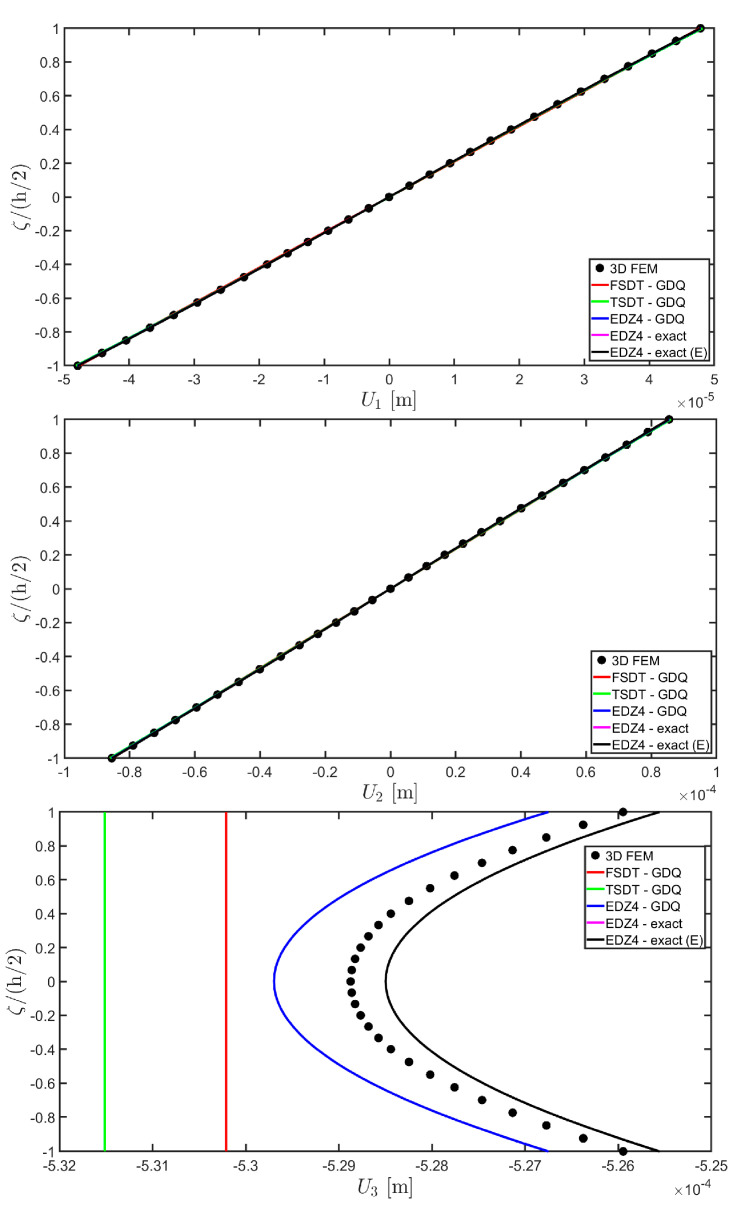
Reconstruction of the profiles of the components of the three-dimensional displacement field vector $U\left(\alpha_{1},\alpha_{2},\zeta \right)$ along the thickness direction of a laminated rectangular plate subjected to patch loads of magnitudes ${\bar{q}}_{3}^{\left(+\right)}=-7\times 10^{5}\;N/m^{2}$ and ${\bar{q}}_{3}^{\left(-\right)}=-3\times 10^{5}\;N/m^{2}$ with $c_{10}=0.25\;L_{1}$ and $c_{20}=0.25\;L_{2}$ applied at $\left(s_{10},s_{20}\right)=\left(0.75\;L_{1},0.75\;L_{2}\right)$. Thickness plots have been provided for the point located at $\left(0.25\;L_{1},0.25\;L_{2}\right)$.

**Figure 6 materials-17-00588-f006:**
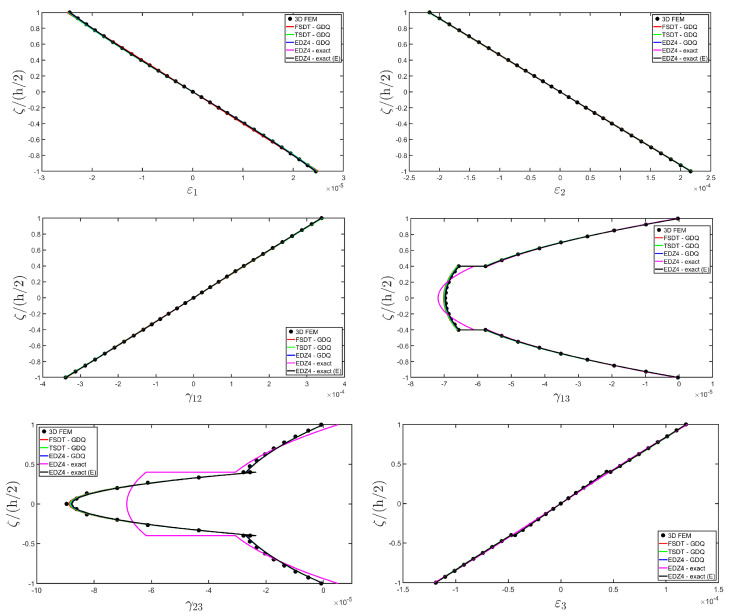
Reconstruction of the profiles of the components of the three-dimensional strain vector $\varepsilon \left(\alpha_{1},\alpha_{2},\zeta \right)$ along the thickness direction of a laminated rectangular plate subjected to patch loads of magnitudes ${\bar{q}}_{3}^{\left(+\right)}=-7\times 10^{5}\;N/m^{2}$ and ${\bar{q}}_{3}^{\left(-\right)}=-3\times 10^{5}\;N/m^{2}$ with $c_{10}=0.25\;L_{1}$ and $c_{20}=0.25\;L_{2}$ applied at $\left(s_{10},s_{20}\right)=\left(0.75\;L_{1},0.75\;L_{2}\right)$. Thickness plots have been provided for the point located at $\left(0.25\;L_{1},0.25\;L_{2}\right)$.

**Figure 7 materials-17-00588-f007:**
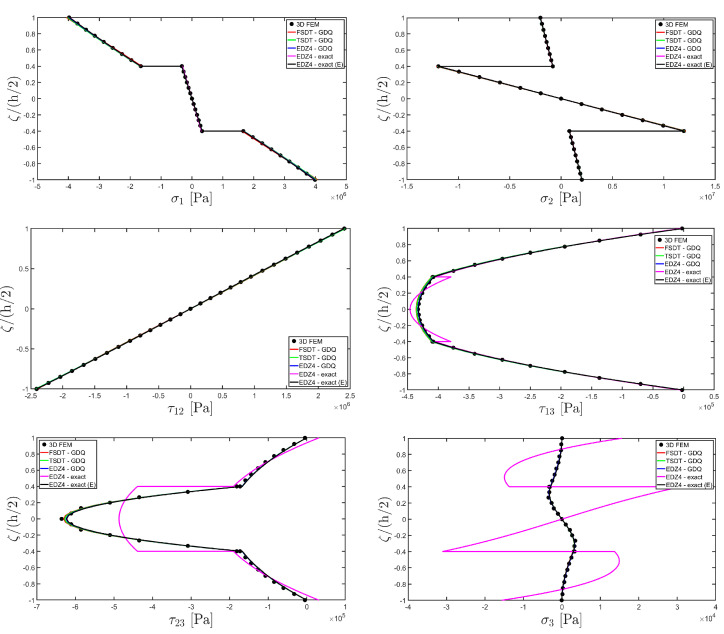
Reconstruction of the profiles of the components of the three-dimensional stress vector $\sigma \left(\alpha_{1},\alpha_{2},\zeta \right)$ along the thickness direction of a laminated rectangular plate subjected to patch loads of magnitudes ${\bar{q}}_{3}^{\left(+\right)}=-7\times 10^{5}\;N/m^{2}$ and ${\bar{q}}_{3}^{\left(-\right)}=-3\times 10^{5}\;N/m^{2}$ with $c_{10}=0.25\;L_{1}$ and $c_{20}=0.25\;L_{2}$ applied at $\left(s_{10},s_{20}\right)=\left(0.75\;L_{1},0.75\;L_{2}\right)$. Thickness plots have been provided for the point located at $\left(0.25\;L_{1},0.25\;L_{2}\right)$.

**Figure 8 materials-17-00588-f008:**
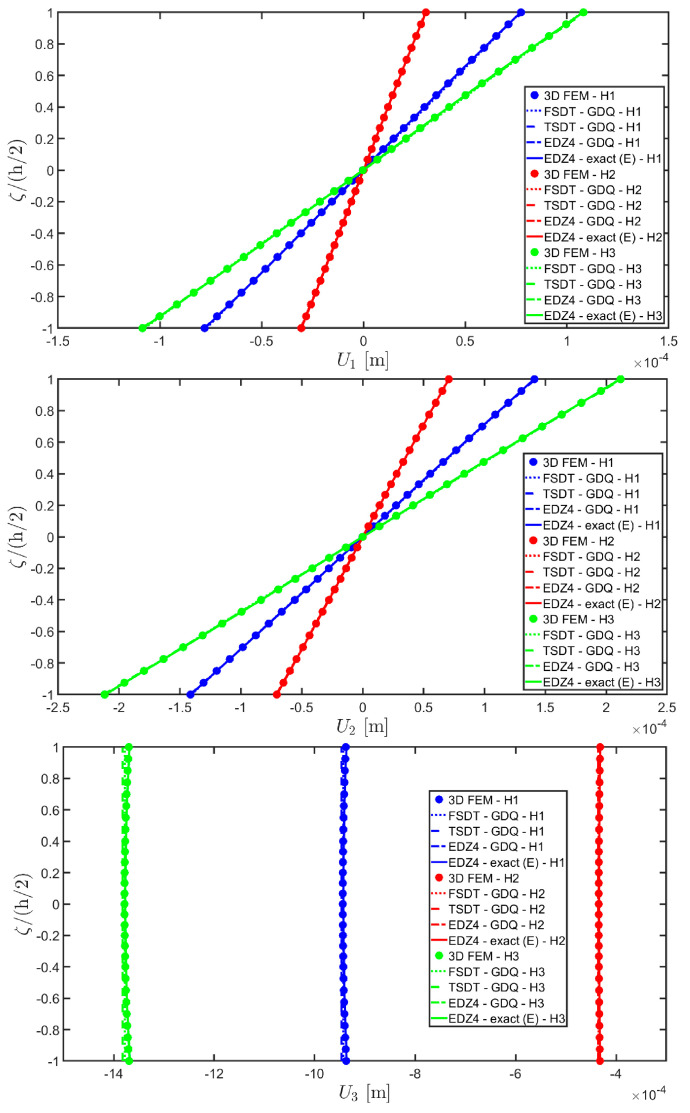
Reconstruction of the profiles of the components of the three-dimensional displacement field vector $U\left(\alpha_{1},\alpha_{2},\zeta \right)$ along the thickness direction of a laminated rectangular plate subjected to hydrostatic loads along $\alpha_{1}$ and $\alpha_{2}$ principal directions of magnitudes ${\bar{q}}_{3}^{\left(+\right)}=-7\times 10^{5}\;N/m^{2}$ and ${\bar{q}}_{3}^{\left(-\right)}=-3\times 10^{5}\;N/m^{2}$, respectively. Thickness plots have been provided for the point located at $\left(0.25\;L_{1},0.25\;L_{2}\right)$.

**Figure 9 materials-17-00588-f009:**
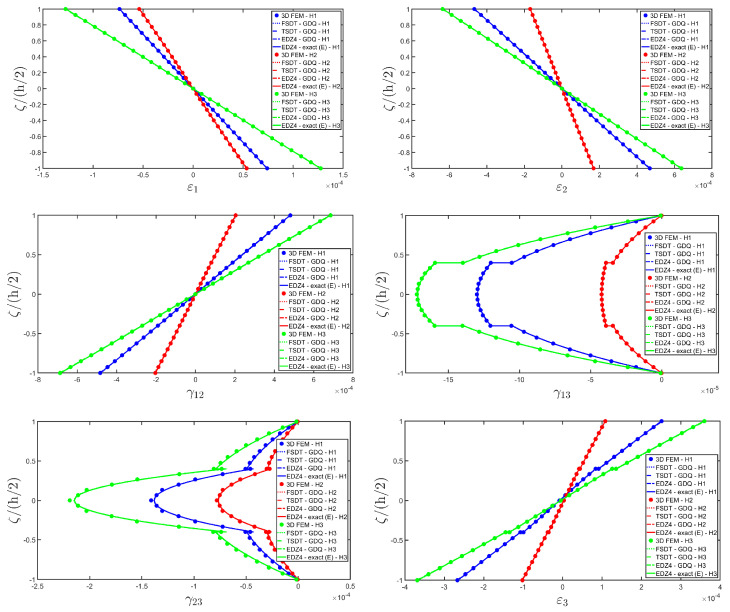
Reconstruction of the profiles of the components of the three-dimensional strain vector $\varepsilon \left(\alpha_{1},\alpha_{2},\zeta \right)$ along the thickness direction of a laminated rectangular plate subjected to hydrostatic loads along $\alpha_{1}$ and $\alpha_{2}$ principal directions of magnitudes ${\bar{q}}_{3}^{\left(+\right)}=-7\times 10^{5}\;N/m^{2}$ and ${\bar{q}}_{3}^{\left(-\right)}=-3\times 10^{5}\;N/m^{2}$, respectively. Thickness plots have been provided for the point located at $\left(0.25\;L_{1},0.25\;L_{2}\right)$.

**Figure 10 materials-17-00588-f010:**
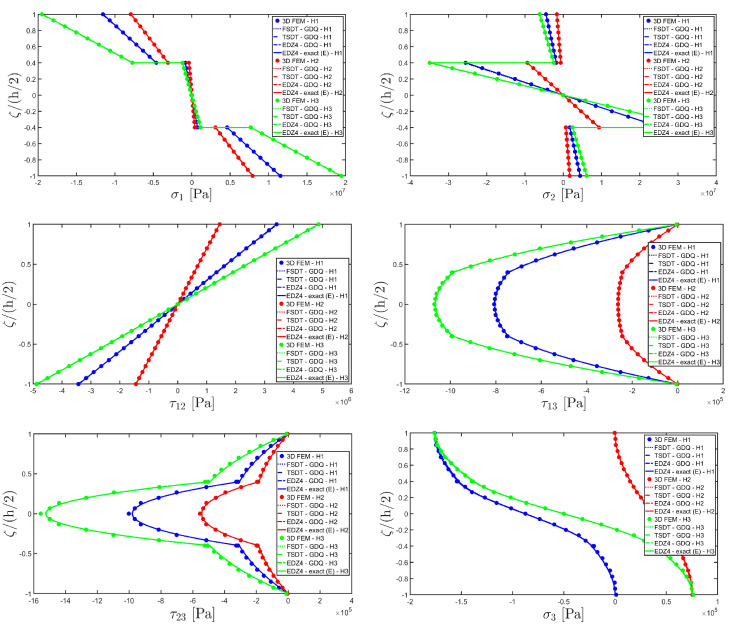
Reconstruction of the profiles of the components of the three-dimensional stress vector $\sigma \left(\alpha_{1},\alpha_{2},\zeta \right)$ along the thickness direction of a laminated rectangular plate subjected to hydrostatic loads along $\alpha_{1}$ and $\alpha_{2}$ principal directions of magnitudes ${\bar{q}}_{3}^{\left(+\right)}=-7\times 10^{5}\;N/m^{2}$ and ${\bar{q}}_{3}^{\left(-\right)}=-3\times 10^{5}\;N/m^{2}$, respectively. Thickness plots have been provided for the point located at $\left(0.25\;L_{1},0.25\;L_{2}\right)$.

**Figure 11 materials-17-00588-f011:**
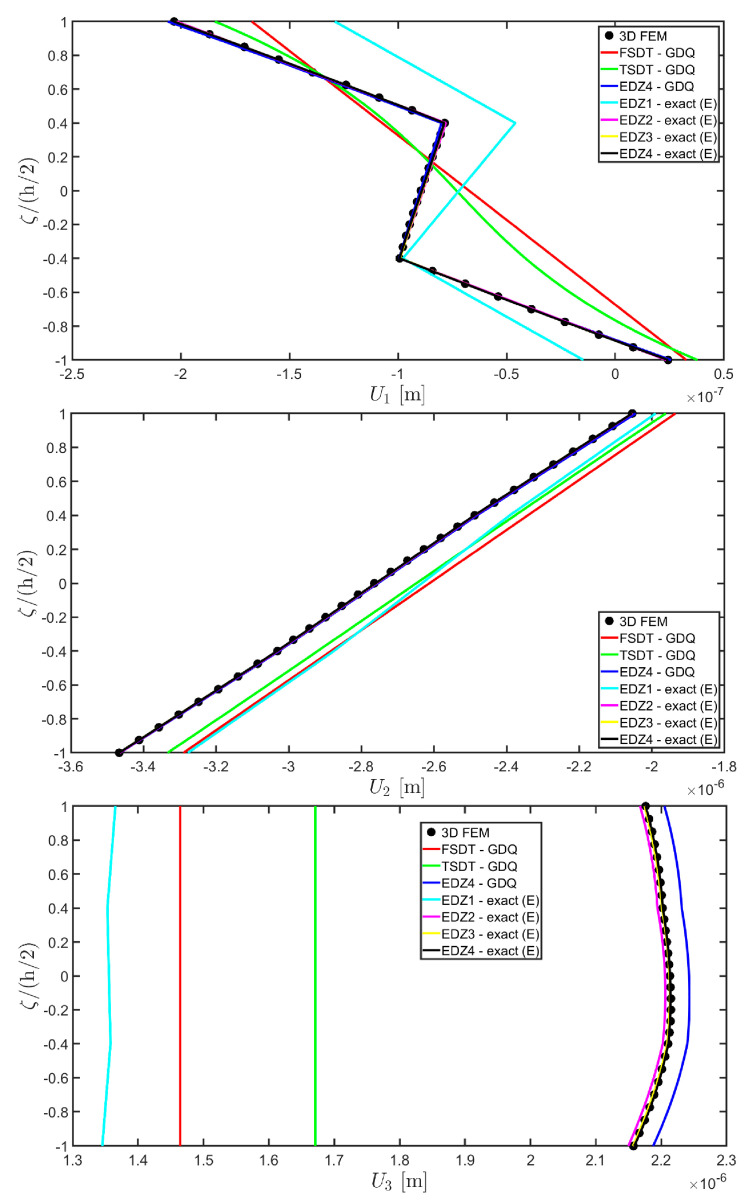
Reconstruction of the profiles of the components of the three-dimensional displacement field vector $U\left(\alpha_{1},\alpha_{2},\zeta \right)$ along the thickness direction of a laminated cylindrical panel subjected to concentrated loads of magnitudes ${\bar{q}}_{3}^{\left(+\right)}=-2000\;N$ and ${\bar{q}}_{3}^{\left(-\right)}=-2000\;N$ with $s_{10}=0.5\;L_{1}$ and $s_{20}=0.5\;L_{2}$. Thickness plots have been provided for the point located at $\left(0.25\;L_{1},0.25\;L_{2}\right)$.

**Figure 12 materials-17-00588-f012:**
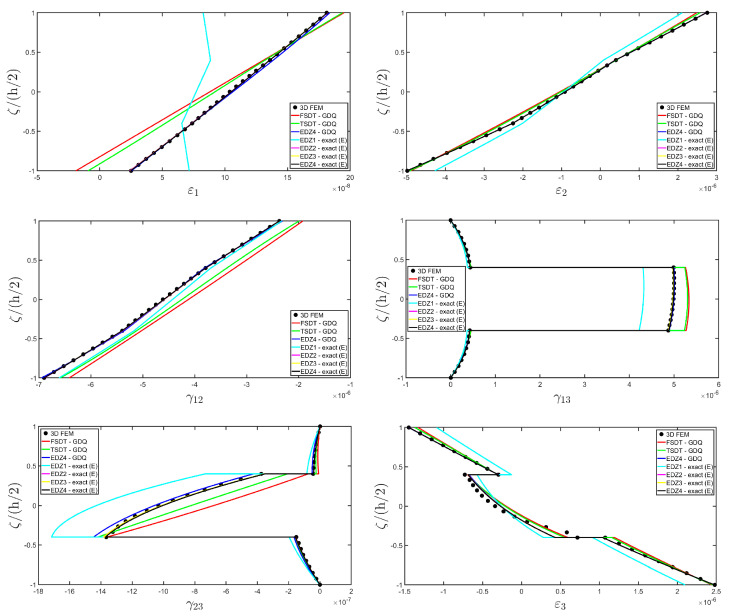
Reconstruction of the profiles of the components of the three-dimensional strain vector $\varepsilon \left(\alpha_{1},\alpha_{2},\zeta \right)$ along the thickness direction of a laminated cylindrical panel subjected to concentrated loads of magnitudes ${\bar{q}}_{3}^{\left(+\right)}=-2000\;N$ and ${\bar{q}}_{3}^{\left(-\right)}=-2000\;N$ with $s_{10}=0.5\;L_{1}$ and $s_{20}=0.5\;L_{2}$. Thickness plots have been provided for the point located at $\left(0.25\;L_{1},0.25\;L_{2}\right)$.

**Figure 13 materials-17-00588-f013:**
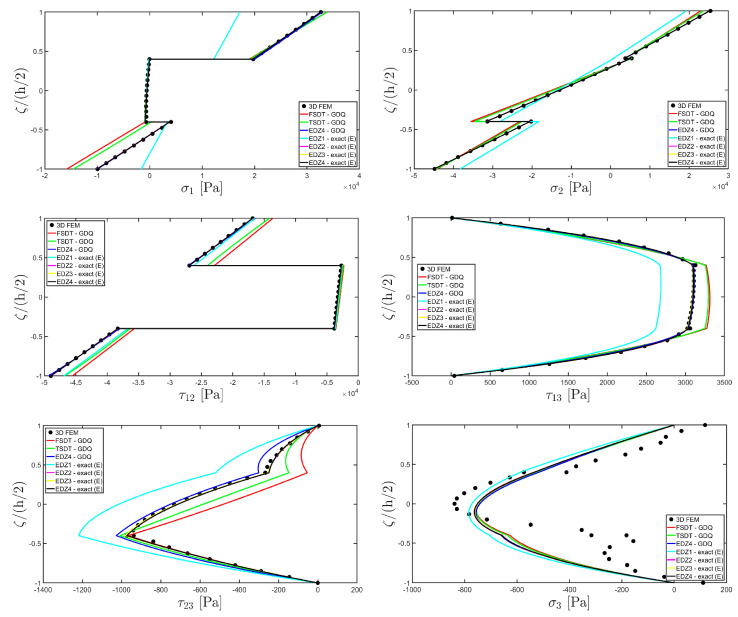
Reconstruction of the profiles of the components of the three-dimensional stress vector $\sigma \left(\alpha_{1},\alpha_{2},\zeta \right)$ along the thickness direction of a laminated cylindrical panel subjected to concentrated loads of magnitudes ${\bar{q}}_{3}^{\left(+\right)}=-2000\;N$ and ${\bar{q}}_{3}^{\left(-\right)}=-2000\;N$ with $s_{10}=0.5\;L_{1}$ and $s_{20}=0.5\;L_{2}$. Thickness plots have been provided for the point located at $\left(0.25\;L_{1},0.25\;L_{2}\right)$.

**Figure 14 materials-17-00588-f014:**
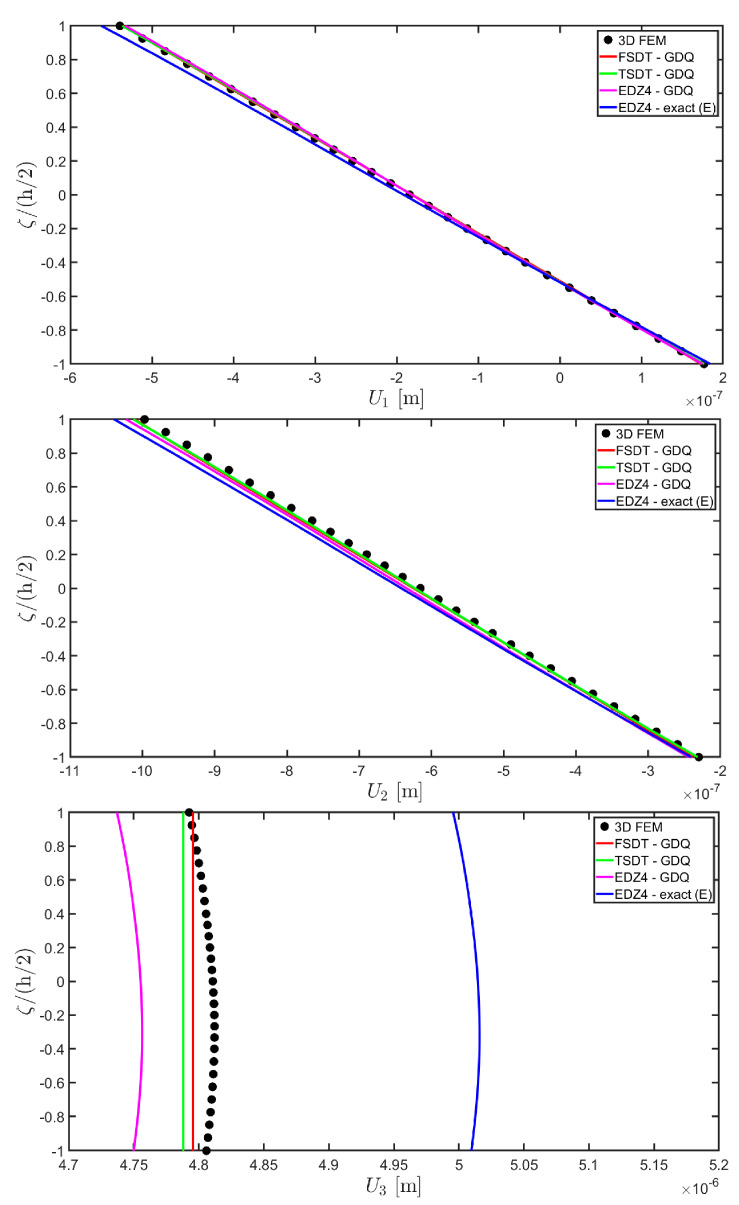
Reconstruction of the profiles of the components of the three-dimensional displacement field vector $U\left(\alpha_{1},\alpha_{2},\zeta \right)$ along the thickness direction of a laminated cylindrical panel subjected to a line load distributed along $\alpha_{1}$ principal direction of magnitude ${\bar{q}}_{3}^{\left(+\right)}=-4.58\times 10^{3}\;N/m$. The Navier solution has been calculated setting n=m=500. Thickness plots have been provided for the point located at $\left(0.25\;L_{1},0.25\;L_{2}\right)$.

**Figure 15 materials-17-00588-f015:**
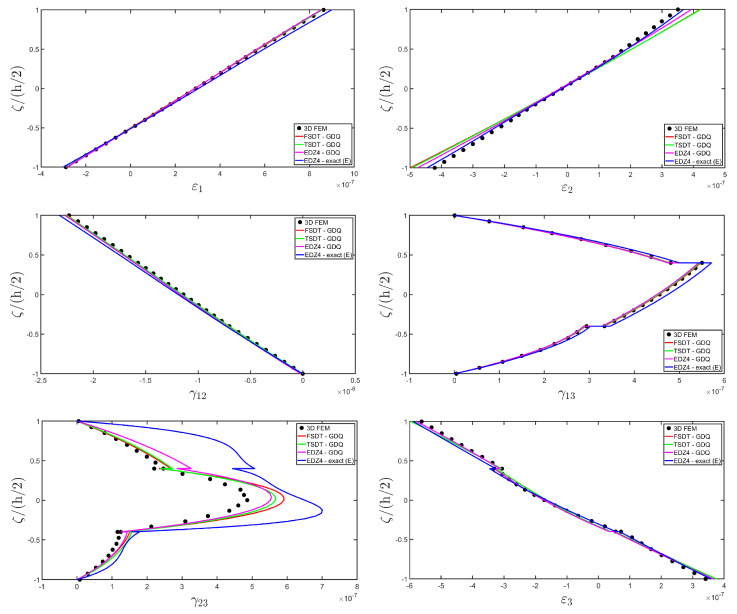
Reconstruction of the profiles of the components of the three-dimensional strain vector $\varepsilon \left(\alpha_{1},\alpha_{2},\zeta \right)$ along the thickness direction of a laminated cylindrical panel subjected to a line load distributed along $\alpha_{1}$ principal direction of magnitude ${\bar{q}}_{3}^{\left(+\right)}=-4.58\times 10^{3}\;N/m$. The Navier solution has been calculated setting n=m=500. Thickness plots have been provided for the point located at $\left(0.25\;L_{1},0.25\;L_{2}\right)$.

**Figure 16 materials-17-00588-f016:**
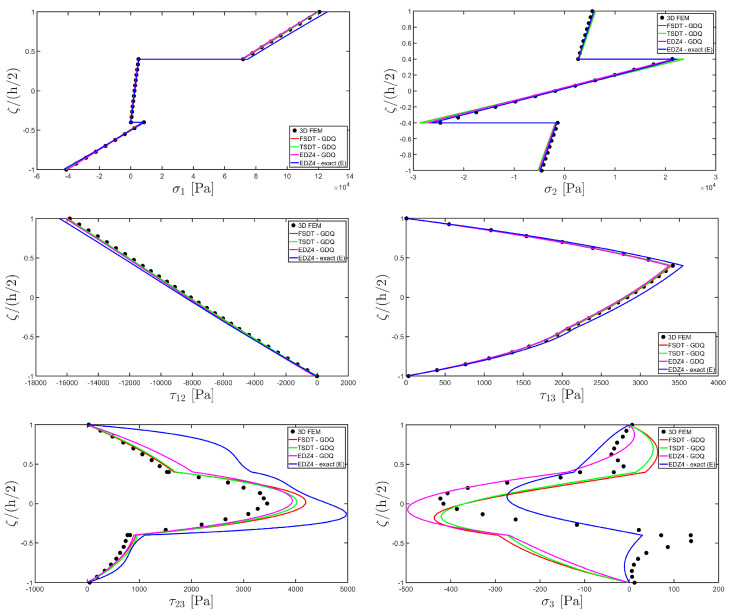
Reconstruction of the profiles of the components of the three-dimensional stress vector $\sigma \left(\alpha_{1},\alpha_{2},\zeta \right)$ along the thickness direction of a laminated cylindrical panel subjected to a line load distributed along $\alpha_{1}$ principal direction of magnitude ${\bar{q}}_{3}^{\left(+\right)}=-4.58\times 10^{3}\;N/m$. The Navier solution has been calculated setting n=m=500. Thickness plots have been provided for the point located at $\left(0.25\;L_{1},0.25\;L_{2}\right)$.

**Figure 17 materials-17-00588-f017:**
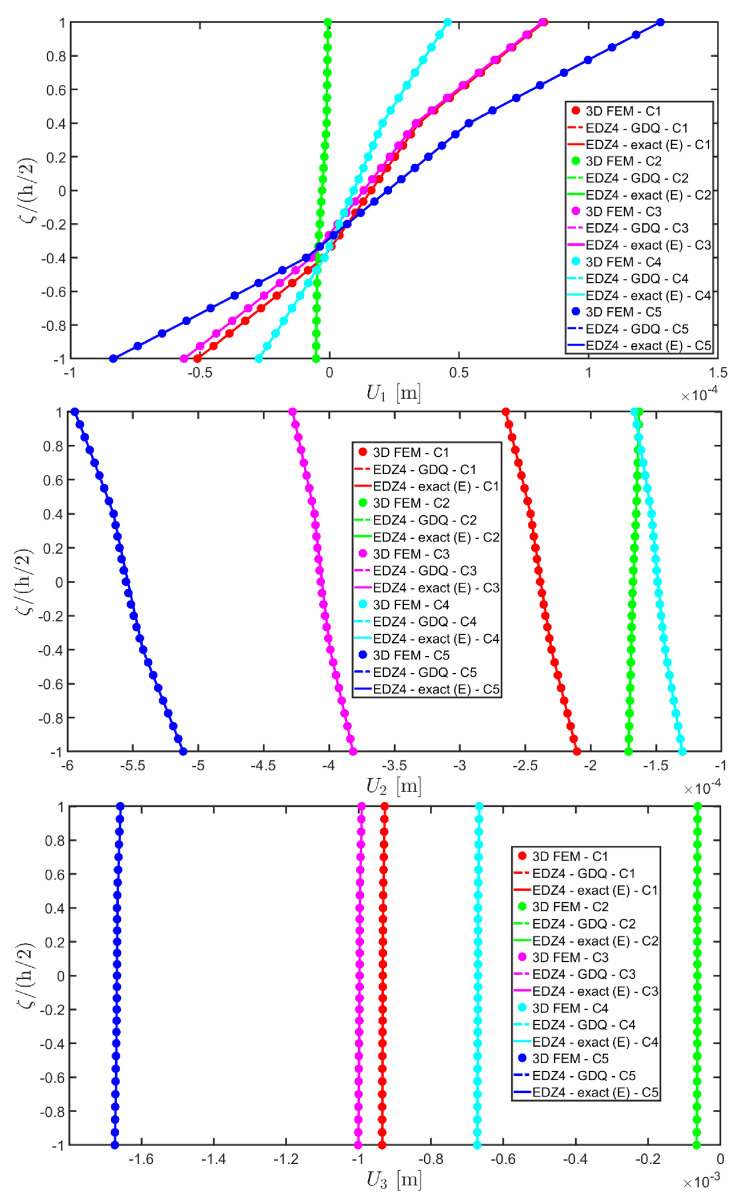
Reconstruction of the profiles of the components of the three-dimensional displacement field vector $U\left(\alpha_{1},\alpha_{2},\zeta \right)$ along the thickness direction of a laminated cylindrical panel subjected to a combination of uniform and hydrostatic loads along $\alpha_{1}$ and $\alpha_{2}$ principal directions of magnitudes ${\bar{q}}_{3}^{\left(+\right)}=-2\times 10^{5}\;N/m^{2}$, ${\bar{q}}_{3}^{\left(+\right)}=-7\times 10^{5}\;N/m^{2}$ and ${\bar{q}}_{3}^{\left(+\right)}=-4\times 10^{5}\;N/m^{2}$, respectively. Thickness plots have been provided for the point located at $\left(0.25\;L_{1},0.25\;L_{2}\right)$.

**Figure 18 materials-17-00588-f018:**
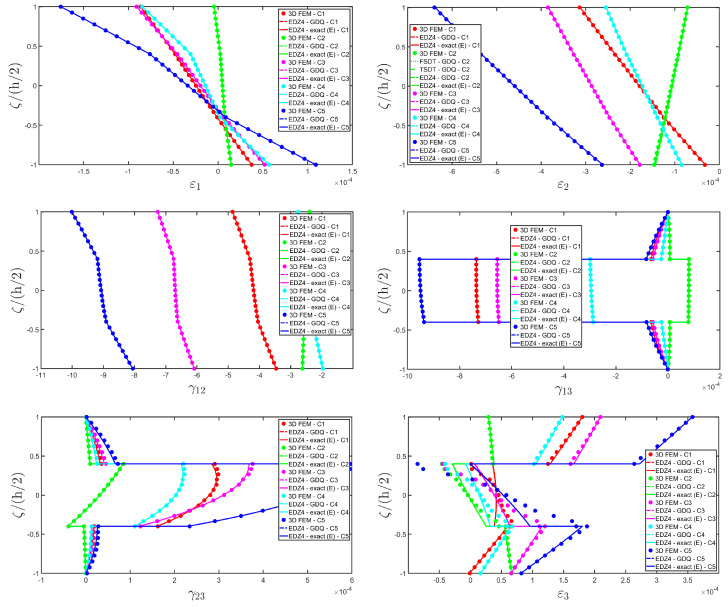
Reconstruction of the profiles of the components of the three-dimensional strain vector $\varepsilon \left(\alpha_{1},\alpha_{2},\zeta \right)$ along the thickness direction of a laminated rectangular plate subjected to a combination of uniform and hydrostatic loads along $\alpha_{1}$ and $\alpha_{2}$ principal directions of magnitudes ${\bar{q}}_{3}^{\left(+\right)}=-2\times 10^{5}\;N/m^{2}$, ${\bar{q}}_{3}^{\left(+\right)}=-7\times 10^{5}\;N/m^{2}$ and ${\bar{q}}_{3}^{\left(+\right)}=-4\times 10^{5}\;N/m^{2}$, respectively. Thickness plots have been provided for the point located at $\left(0.25\;L_{1},0.25\;L_{2}\right)$.

**Figure 19 materials-17-00588-f019:**
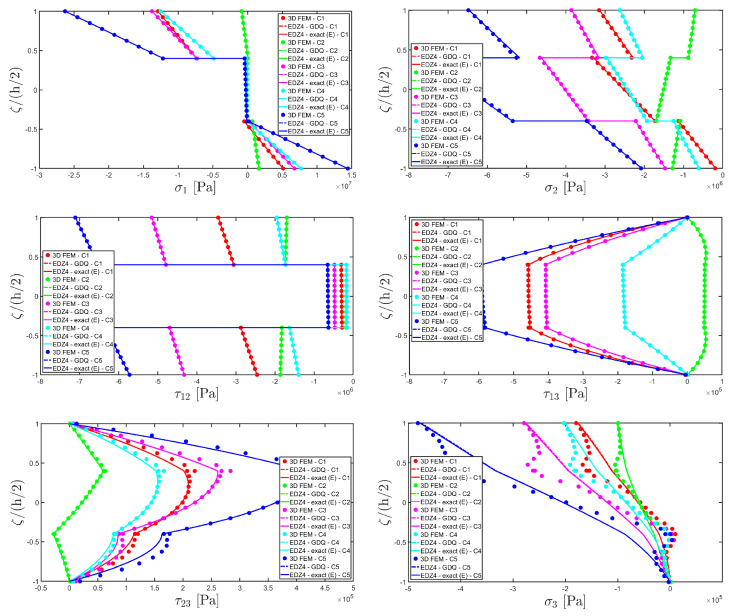
Reconstruction of the profiles of the components of the three-dimensional stress vector $\sigma \left(\alpha_{1},\alpha_{2},\zeta \right)$ along the thickness direction of a laminated rectangular plate subjected to a combination of uniform and hydrostatic loads along $\alpha_{1}$ and $\alpha_{2}$ principal directions of magnitudes ${\bar{q}}_{3}^{\left(+\right)}=-2\times 10^{5}\;N/m^{2}$, ${\bar{q}}_{3}^{\left(+\right)}=-7\times 10^{5}\;N/m^{2}$ and ${\bar{q}}_{3}^{\left(+\right)}=-4\times 10^{5}\;N/m^{2}$, respectively. Thickness plots have been provided for the point located at $\left(0.25\;L_{1},0.25\;L_{2}\right)$.

**Figure 20 materials-17-00588-f020:**
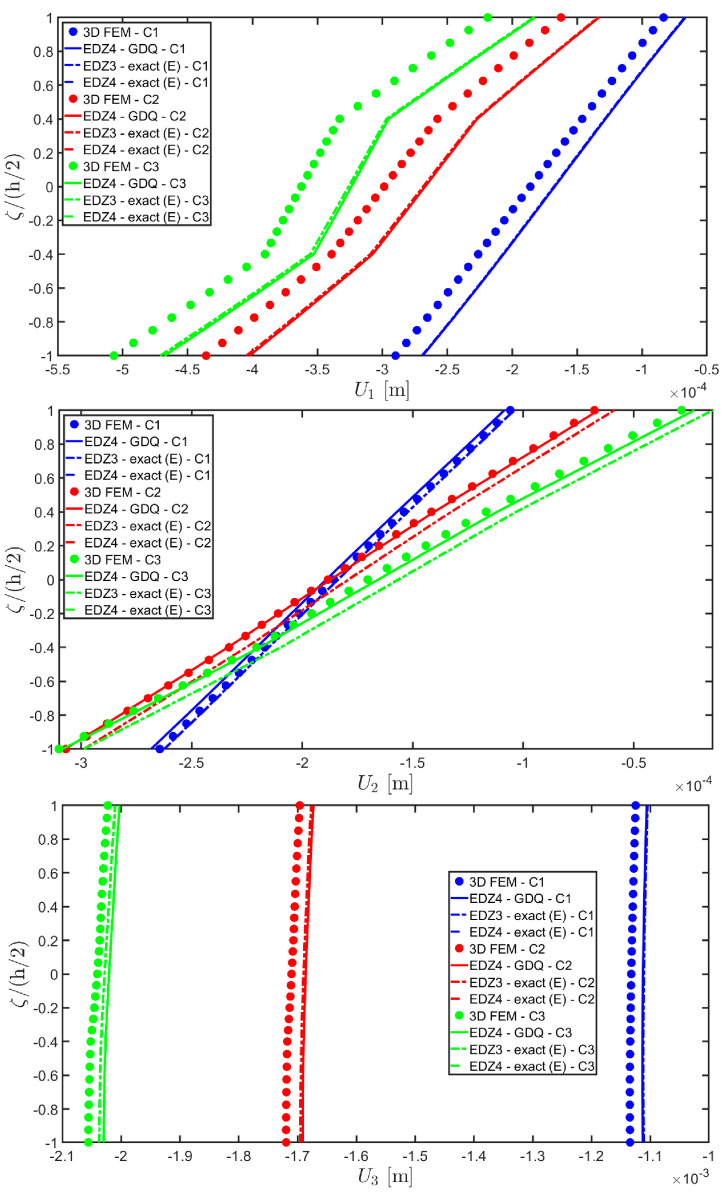
Reconstruction of the profiles of the components of the three-dimensional displacement field vector $U\left(\alpha_{1},\alpha_{2},\zeta \right)$ along the thickness direction of a laminated spherical panel subjected to sinusoidal loads of magnitudes ${\bar{q}}_{3}^{\left(+\right)}=-7\times 10^{5}\;N/m^{2}$ and ${\bar{q}}_{3}^{\left(-\right)}=-3\times 10^{5}\;N/m^{2}$ with n=m=1. Thickness plots have been provided for the point located at $\left(0.25\;L_{1},0.25\;L_{2}\right)$.

**Figure 21 materials-17-00588-f021:**
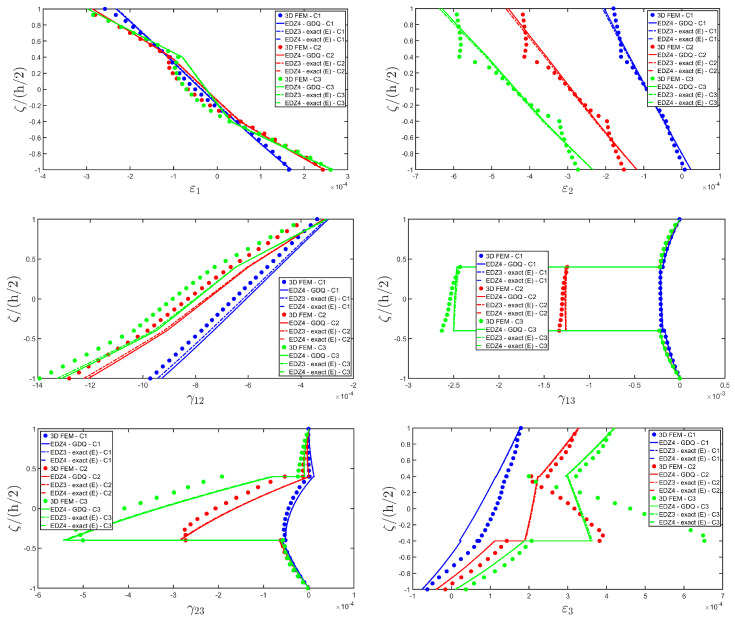
Reconstruction of the profiles of the components of the three-dimensional strain vector $\varepsilon \left(\alpha_{1},\alpha_{2},\zeta \right)$ along the thickness direction of a laminated spherical panel subjected to sinusoidal loads of magnitudes ${\bar{q}}_{3}^{\left(+\right)}=-7\times 10^{5}\;N/m^{2}$ and ${\bar{q}}_{3}^{\left(-\right)}=-3\times 10^{5}\;N/m^{2}$ with $\tilde{N}=\tilde{M}=1$. Thickness plots have been provided for the point located at $\left(0.25\;L_{1},0.25\;L_{2}\right)$.

**Figure 22 materials-17-00588-f022:**
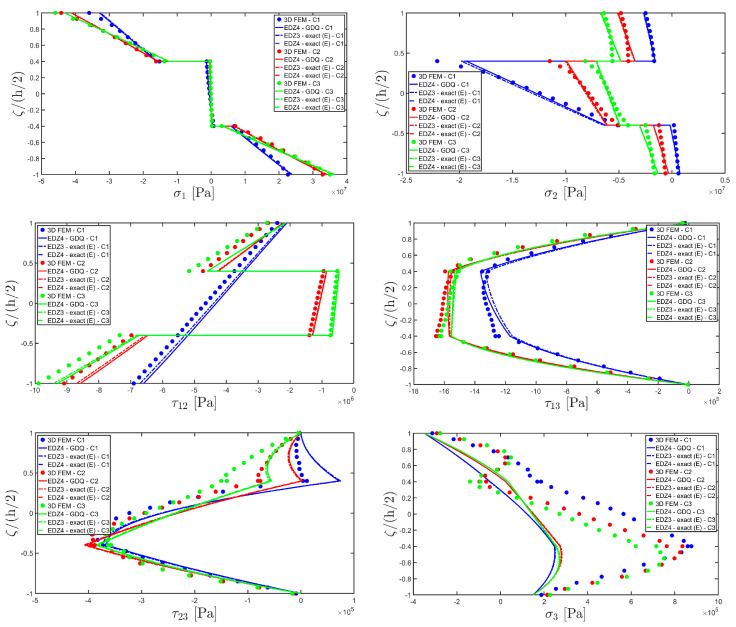
Reconstruction of the profiles of the components of the three-dimensional stress vector $\sigma \left(\alpha_{1},\alpha_{2},\zeta \right)$ along the thickness direction of a laminated spherical panel subjected to sinusoidal loads of magnitudes ${\bar{q}}_{3}^{\left(+\right)}=-7\times 10^{5}\;N/m^{2}$ and ${\bar{q}}_{3}^{\left(-\right)}=-3\times 10^{5}\;N/m^{2}$ with $\tilde{N}=\tilde{M}=1$. Thickness plots have been provided for the point located at $\left(0.25\;L_{1},0.25\;L_{2}\right)$.

**Figure 23 materials-17-00588-f023:**
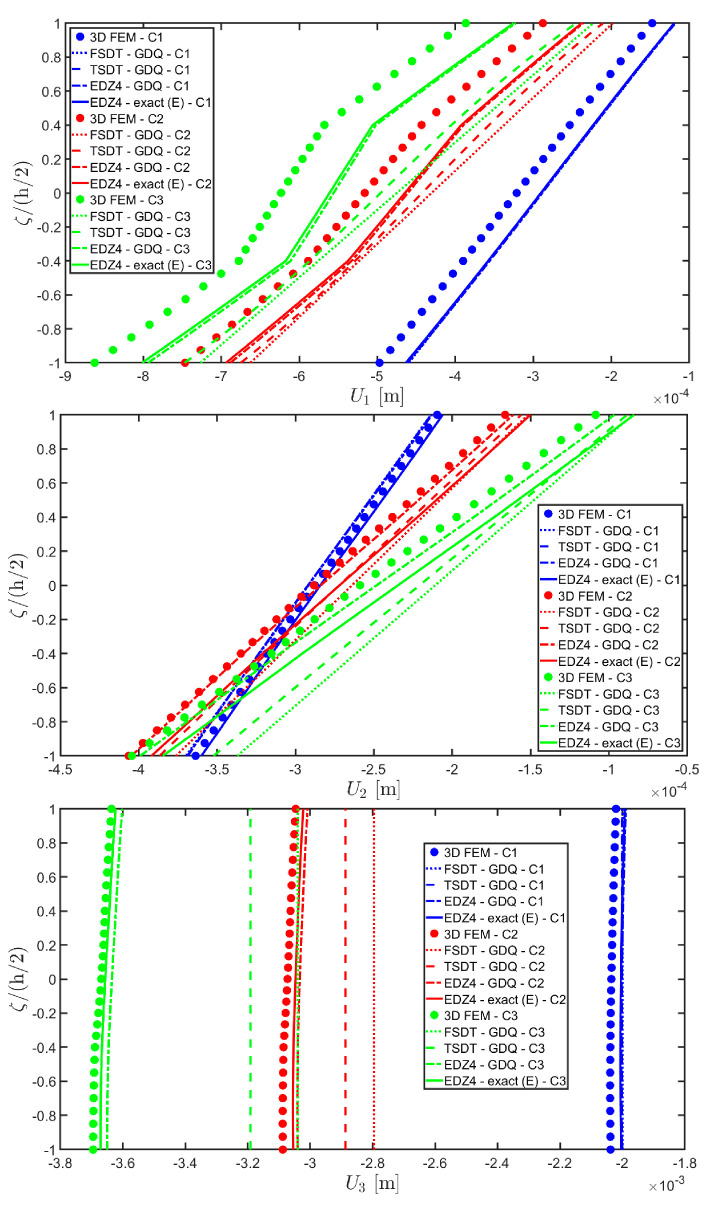
Reconstruction of the profiles of the components of the three-dimensional displacement field vector $U\left(\alpha_{1},\alpha_{2},\zeta \right)$ along the thickness direction of a laminated spherical panel subjected to uniform loads of magnitudes ${\bar{q}}_{3}^{\left(+\right)}=-7\times 10^{5}\;N/m^{2}$ and ${\bar{q}}_{3}^{\left(-\right)}=-3\times 10^{5}\;N/m^{2}$. Thickness plots have been provided for the point located at $\left(0.25\;L_{1},0.25\;L_{2}\right)$.

**Figure 24 materials-17-00588-f024:**
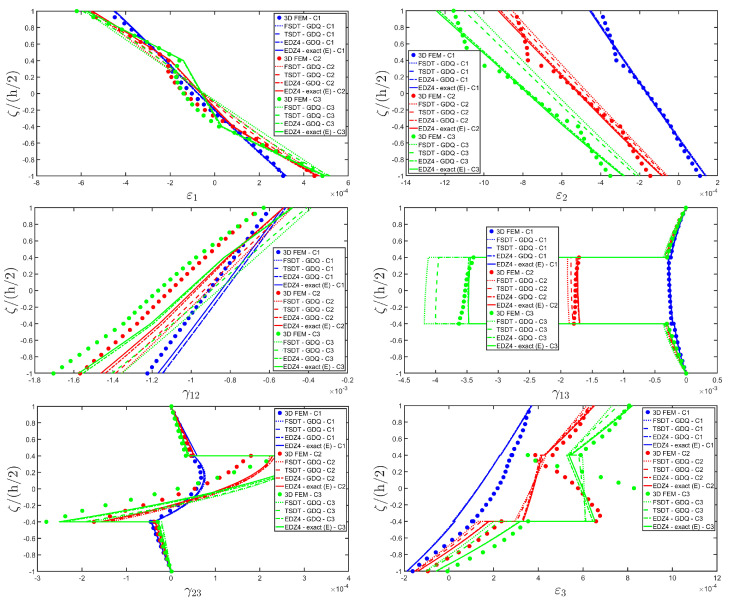
Reconstruction of the profiles of the components of the three-dimensional strain vector $\varepsilon \left(\alpha_{1},\alpha_{2},\zeta \right)$ along the thickness direction of a laminated spherical panel subjected to uniform loads of magnitudes ${\bar{q}}_{3}^{\left(+\right)}=-7\times 10^{5}\;N/m^{2}$ and ${\bar{q}}_{3}^{\left(-\right)}=-3\times 10^{5}\;N/m^{2}$. Thickness plots have been provided for the point located at $\left(0.25\;L_{1},0.25\;L_{2}\right)$.

**Figure 25 materials-17-00588-f025:**
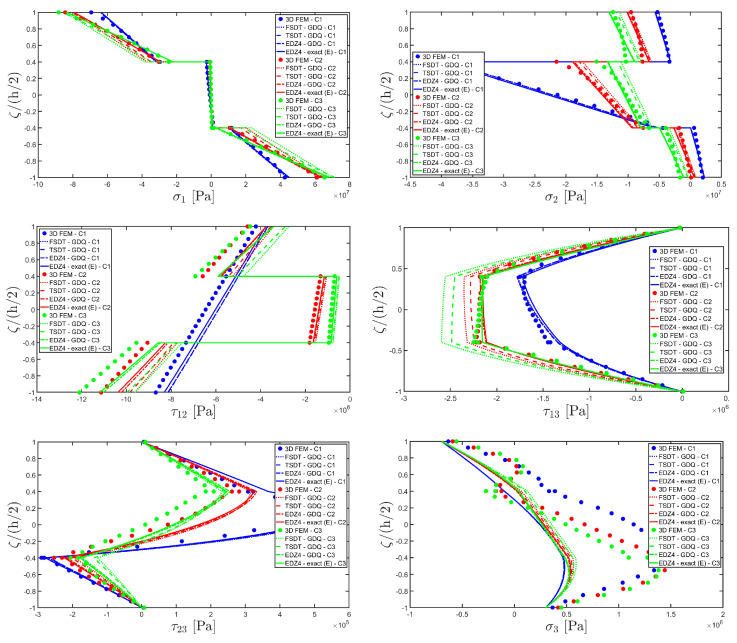
Reconstruction of the profiles of the components of the three-dimensional stress vector $\sigma \left(\alpha_{1},\alpha_{2},\zeta \right)$ along the thickness direction of a laminated spherical panel subjected to uniform loads of magnitudes ${\bar{q}}_{3}^{\left(+\right)}=-7\times 10^{5}\;N/m^{2}$ and ${\bar{q}}_{3}^{\left(-\right)}=-3\times 10^{5}\;N/m^{2}$. Thickness plots have been provided for the point located at $\left(0.25\;L_{1},0.25\;L_{2}\right)$.

**Table 1 materials-17-00588-t001:** Static analysis of a rectangular plate subjected to a patch load. The percentage error in Equation (97) defines the convergence rate of the semi-analytical solution with respect to the reference simulation derived from a three-dimensional finite element simulation. The quantity $w_{\tilde{N}}$ is expressed in $10^{-4}\;\;m$.

$\tilde{N}\;=\;\tilde{M}$	$w_{\;\tilde{N}}$	$e_{\;\%}$
5	15.0476	0.16%
10	15.0420	0.12%
12	15.0077	0.11%
20	15.0084	0.10%
50	15.0085	0.10%
100	15.0085	0.10%
120	15.0085	0.10%
150	15.0085	0.10%
200	15.0085	0.10%

## Data Availability

No new data were created or analyzed in this study. Data sharing is not applicable to this article.

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
