# Peer review of "On the Importance of the Recovery Procedure in the Semi-Analytical Solution for the Static Analysis of Curved Laminated Panels: Comparison with 3D Finite Elements"

_materials, 2024, doi:10.3390/ma17030588_

Round 1

Reviewer 1 Report

Comments and Suggestions for Authors

The manuscript is devoted to the modelling of static response of laminated panels to general external load. A semi-analytical simulation procedure is proposed and tested comparing simulation results with the results obtained by the finite element method. The general idea of the procedure is the simulation using a 2D model with subsequent application of a so-called "recovery procedure" to obtain 3D solutions numerically solving additional differential equations using a finite-difference scheme. Despite the general approach is not new, as it is mentioned in the paper, the combination of techniques and the presented simulation results form a sufficient novelty.

In general, the manuscript is well-written and contains a very detailed and thorough description of both the background and results.

Sections 2-5,7,8 seems to repeat or at least extensively use the results from [16]. Regarding the overall size of the paper they could be shortened.

The issue of computational cost is not analysed in the Results section despite the claim that proposed simulation procedure lowers it. Specifically, computation times are not given and discussed. Linear solver used for obtaining finite-difference solutions is not specified. Given grid sizes are different for different cases and this issue is not discussed.

As different studied techniques show different behaviour of accuracy according to the presented simulation results, it would be beneficial to make some summary of these results in Conclusions to give some recommendations about what technique to use in what situation.

One minor issue should also be checked:

L175-188: please check whether commas are needed in the subscripts of r.

Author Response

We thank the Reviewers for their valuable comments, which helped the authors to improve the quality of the manuscript. The detailed responses are reported here below.

Reviewer #1

The manuscript is devoted to the modelling of static response of laminated panels to general external load. A semi-analytical simulation procedure is proposed and tested comparing simulation results with the results obtained by the finite element method. The general idea of the procedure is the simulation using a 2D model with subsequent application of a so-called "recovery procedure" to obtain 3D solutions numerically solving additional differential equations using a finite-difference scheme. Despite the general approach is not new, as it is mentioned in the paper, the combination of techniques and the presented simulation results form a sufficient novelty.

Comment: Sections 2-5,7,8 seems to repeat or at least extensively use the results from [16]. Regarding the overall size of the paper, they could be shortened.

Response: We thank the Reviewer for his valuable comment. The sections mentioned by the Reviewer summarize the main aspects of a more general theory which is here solved with a semi-analytical procedure. The reader is addressed to Ref. [16] for a more comprehensive presentation of the general topic. In this way, we aim to provide a self-consistent paper which does not strictly require the adoption of further references for a complete understanding of the theory. In the revised version of the manuscript, the sections at issue have been revised so that the overall size of the paper is shortened.

Comment: The issue of computational cost is not analyzed in the Results section despite the claim that proposed simulation procedure lowers it. Specifically, computation times are not given and discussed. Linear solver used for obtaining finite-difference solutions is not specified. Given grid sizes are different for different cases and this issue is not discussed.

Response: We thank the Reviewer for his valuable comment. As stated throughout the paper, a semi-analytical solution of the fundamental relations is provided following the Navier’s approach. In this context, the results are obtained with a reduced computational cost since no numerical methods are applied at this stage. The issue of the computational cost is thus expressed in terms of the number of waves (and consequently of the linear systems to be solved using MATLAB software) required for the convergence of results with respect of the predictions of 3D FEM models. Specifically, regarding the computational cost in terms of time is not important for an analytical solution. Obviously, the solution has a negligible cost in terms of time, if compared with 3D solutions; specifically, we have a solution in few seconds against a 3D solution in hours. This could be not of interest for the reader.

Furthermore, the derivatives of the three-dimensional stress components occurring in the recovery procedure are performed numerically through the Generalized Differential Quadrature (GDQ), since otherwise the semi-analytical methodology would have required the computation of these derivatives for each wave number. The main features of the GDQ method are briefly presented in a proper section, and further details are provided in Ref. [16] for the interested reader. Regarding the selection of the size of the computational grid, the Authors have decided not to report a preliminary check because the main focus of the work is the semi-analytical solution and the recovery procedure. The size of the computational grid has already been focused with more details in several previous papers by the Authors (refers also to [16]). In the revised version of the manuscript, this aspect has been more clearly pointed out.

Comment: As different studied techniques show different behaviour of accuracy according to the presented simulation results, it would be beneficial to make some summary of these results in Conclusions to give some recommendations about what technique to use in what situation.

Response: We thank the Reviewer for his valuable comment. The Conclusion section has been revised in its current form in order to include further details in brief of the main results obtained in the present study. Furthermore, some recommendations are given about the applicability of the formulation in some specific situations like different materials and load shapes.

Comment: L175-188: please check whether commas are needed in the subscripts of r.

Response: We thank the Reviewer for his valuable comment. The presence of commas in lines 175-188 in the symbol  indicate the partial derivative of the reference surface position vector  with respect to  principal direction. In this context, this is not a typo, but a specific choice by the authors.

Reviewer 2 Report

Comments and Suggestions for Authors

The paper is very interesting, it deals with mathematical approach to two-dimensional semi-analytical modeling for the evaluation of the static response of curved laminated panels subjected to arbitrary loads. The paper is carefully prepared, however it is quite difficult to read because of many long equations presented.

In my opinion it needs a revision as follows:

1. The equations should be commented - after each part of them please state comments about the meaning of equations and application

2. The conclusions should be widen with the quantitative and qualitative of the results (some comparison), future plans and possibilities of future application (which materials could be modeled this way?)

3. The most important thing - how the analyses were verified?

Author Response

We thank the Reviewers for their valuable comments, which helped the authors to improve the quality of the manuscript. The detailed responses are reported here below.

Reviewer #2

The paper is very interesting, it deals with mathematical approach to two-dimensional semi-analytical modeling for the evaluation of the static response of curved laminated panels subjected to arbitrary loads. The paper is carefully prepared, however it is quite difficult to read because of many long equations presented.

Comment: The equations should be commented - after each part of them please state comments about the meaning of equations and application.

Response: We thank the Reviewer for his valuable comment. The theoretical aspects of the paper have been summarized in order to give the reader a general understanding of the formulation. Further details have been reported in Ref. [16]. In the revised version of the manuscript, the theoretical aspects have been explained, and some mathematical manipulations that are not strictly required to understand the whole meaning of the research have been deleted. In this way, the overall length of the manuscript has been shortened.

Comment: The conclusions should be widened with the quantitative and qualitative of the results (some comparison), future plans and possibilities of future application (which materials could be modeled this way?)

Response: We are grateful to the Reviewer for his comment. The Conclusion section has been modified and widened, pointing out the quantitative and qualitative main findings of the research. Furthermore, some brief remarks of future and applications on various materials and lamination schemes have been added.

Comment: The most important thing - how the analyses were verified?

Response: As it has been remarked throughout the manuscript, the results of the simulations derived from the present two-dimensional theory have been checked with respect to three-dimensional simulations developed with commercial software, which are based on the Finite Element Method (FEM). In a preliminary step that is not cited in the work for the sake of brevity, the convergence and the accuracy of the 3D FEM simulations were verified. Then, these results are taken as reference solutions for the outcomes of the results presented in this paper. Note that in all the thickness plots provided in the paper, black and colored dots refer to the results coming from these simulations. For the sake of clarity, in the revised version of the manuscript a figure is added showing the 3D FEM models adopted in the simulations.

Reviewer 3 Report

Comments and Suggestions for Authors

It is acceptable for publication after revision.

See attached file for the reviewer's comments

Comments on the Quality of English Language

Moderate English editing is required.

Author Response

We thank the Reviewers for their valuable comments, which helped the authors to improve the quality of the manuscript. The detailed responses are reported here below.

Reviewer #3

This paper deals with the semi-analytical solutions with 3D capabilities for the static responses of laminated curved structures subjected to general external loads. The numerical examples are shown in the rectangular plate, cylindrical panel and spherical shell. This paper is well-written and interesting for the structural engineers. It is acceptable to publish in the journal after revision according to the comments.

Comment: Ultimately, the solution is obtained using commercial finite element methods, so the title should reflect this fact. For example, .... using Finite Element Method.

Response: - We thank the Reviewer for his valuable comment. As extensively stated throughout the paper, the solution of the mechanical problem is derived employing the Navier’s method since the generalized displacement field variables are expanded in harmonic form. In this way, a refined two-dimensional solution of the problem under consideration is obtained. Then, in the post-processing, an equilibrium-based procedure employing the GDQ method is adopted to derive the three-dimensional response of the laminated panel. On the other hand, a commercial software has been adopted here to conduct a three-dimensional Finite Element analysis in order to derive a reference solution to check the validity of the semi-analytical solution. In any case, the title of the paper has been varied to “On the Importance of the Recovery Procedure in the Semi-Analytical Solution for the Static Analysis of Curved Laminated Panels: Comparison with 3D Finite Elements”. We hope that this explanation may specify better the main aim of the present study, so that the meaning of the proposed title is clear.

Comment: Line 325: In this study, the cross section is rectangular so that shear correction factor is . Make clear this in the text.

Response: We thank the Reviewer for his valuable comment. The selection of the value  for the shear correction factor moves from the classical theories for beams with rectangular cross section. In the revised version of the manuscript, some comments have been added so that this aspect is made clear within the text.

Comment: In 456: No definition of parameters of  are found. Define these parameters before use. 4. In Figure 1, the thickness of the composite layer should be indicated for clarity.

Response: - We thank the Reviewer for his valuable comment. The quantities under consideration are introduced for the first time in Eqn. 55 of the submitted manuscript. According to the Navier’s method, they account for an infinite number of terms for the harmonic expansion of the solution and external loads. However, results of practical interests can already be derived by considering a truncated series. As a result, the quantities in hand assume an integer value. In the revised paper, further details on the quantities in hand have been added, and they are introduced in Eqn. 54 for the first time. On the other hand, Figure 1 and Figure 2 have been modified according to the Reviewer’s suggestions.

Comment: Line 708: Provide the name of the commercial FEM software used as reference, such as ADINA, ABAQUS, etc.

Response: - We thank the Reviewer for his valuable comment. The reference solutions, derived from commercial FEM software, have been derived employing the ABAQUS code. In the revised version of the paper, this detail has been added.

Comment: In this study , but this cannot adopt in FEM solution. Convergence analysis curve must be displayed in order to the suitable  values through comparing the solutions between this study and EDZ4(Exact).

Response: We thank the Reviewer for his valuable comment. The present formulation, based on a harmonic expansion of the unknown variables, is intended to converge to the real solution for a sufficient number of waves, namely . In the revised version of the manuscript, the convergence analysis curve is displayed. In this way, the convergence of EDZ4 solution is pointed out with respect to 3D FEM reference simulation.

Round 2

Reviewer 1 Report

Comments and Suggestions for Authors

Regarding the corrections, I still adhere to an opinion that the manuscript could have been shortened and that the arguments about computational time could have been included into the text. But, in general, the manuscript was sufficiently improved, I agree with the answers to the comments, and recommend the paper to be accepted in present form.

Author Response

We thank the Reviewers for their valuable comments, which helped the authors to improve the quality of the manuscript. The detailed responses are reported here below.

Reviewer #1

Regarding the corrections, I still adhere to an opinion that the manuscript could have been shortened and that the arguments about computational time could have been included into the text. But, in general, the manuscript was sufficiently improved, I agree with the answers to the comments, and recommend the paper to be accepted in present form.

Response: We are grateful to the Reviewer for his work that helped us to improve the manuscript. For the sake of completeness, some comments have been added in the revised version of the paper addressing the issue of computational time, according to the comments of the previous review step.

Reviewer 2 Report

Comments and Suggestions for Authors

The authors considered all my comments. The paper can be accepted in present form.

Author Response

We thank the Reviewers for their valuable comments, which helped the authors to improve the quality of the manuscript. The detailed responses are reported here below.

Reviewer #2

The authors considered all my comments. The paper can be accepted in present form.

Response: We are grateful to the Reviewer for the appreciation of our research work. We hope that the scientific community will take advantage of our paper.

Reviewer 3 Report

Comments and Suggestions for Authors

The revised manuscript is acceptable for publishing in the journal.

Comments on the Quality of English Language

Minor editing of English is required.